# Win-win opportunities combining high yields with high multi-taxa biodiversity in tropical agroforestry

Annemarie Wurz [1,2 ✉], Teja Tscharntke[1,3], Dominic Andreas Martin [4,5], Kristina Osen[6],
Anjaharinony A. N. A. Rakotomalala[1,7], Estelle Raveloaritiana [1,8], Fanilo Andrianisaina[9], Saskia Dröge[4,10],
Thio Rosin Fulgence[4,11,12], Marie Rolande Soazafy[12,13], Rouvah Andriafanomezantsoa[11], Aristide Andrianarimisa[11],
Fenohaja Soavita Babarezoto[14], Jan Barkmann [15], Hendrik Hänke [15], Dirk Hölscher [3,6], Holger Kreft [3,4],
Bakolimalala Rakouth[8], Nathaly R. Guerrero-Ramírez[4], Hery Lisy Tiana Ranarijaona[13],
Romual Randriamanantena[12], Fanomezana Mihaja Ratsoavina[11], Lala Harivelo Raveloson Ravaomanarivo[7] &
Ingo Grass [16]

Resolving ecological-economic trade-offs between biodiversity and yields is a key challenge when addressing the biodiversity crisis in tropical agricultural landscapes. Here, we focused on the relation between seven different taxa (trees, herbaceous plants, birds, amphibians, reptiles, butterflies, and ants) and yields in vanilla agroforests in Madagascar. Agroforests established in forests supported overall 23% fewer species and 47% fewer endemic species than old-growth forests, and 14% fewer endemic species than forest fragments. In contrast, agroforests established on fallows had overall 12% more species and 38% more endemic species than fallows. While yields increased with vanilla vine density and length, non-yield related variables largely determined biodiversity. Nonetheless, trade-offs existed between yields and butterflies as well as reptiles. Vanilla yields were generally unrelated to richness of trees, herbaceous plants, birds, amphibians, reptiles, and ants, opening up possibilities for conservation outside of protected areas and restoring degraded land to benefit farmers and biodiversity alike.

[1] Agroecology, University of Göttingen, Grisebachstr. 6, 37077 Göttingen, Germany. [2] Conservation Ecology, Department of Biology, Philipps-University Marburg, Marburg, Germany. [3] Centre for Biodiversity and Sustainable Land Use (CBL), University of Göttingen, Büsgenweg 1, 37077 Göttingen, Germany. [4] Biodiversity, Macroecology and Biogeography, University of Göttingen, Büsgenweg 1, 37077 Göttingen, Germany. [5] Department of Geography, University of Zurich, Winterthurerstrasse 190, 8057 Zürich, Switzerland. [6] Tropical Silviculture and Forest Ecology, University of Göttingen, Büsgenweg 1, 37077 Göttingen, Germany. [7] Entomology Department Faculty of Science, University of Antananarivo, PO Box 906 Antananarivo 101, Madagascar. [8] Plant Biology and Ecology Department, University of Antananarivo, University of Antananarivo, Antananarivo, Madagascar. [9] Department of Tropical Agriculture and Sustainable Development, Higher School of Agronomic Science,University of Antananarivo, Antananarivo, Madagascar. [10] Division of Forest, Nature and Landscape, KU Leuven, Leuven, Belgium. [11] Zoology and Animal Biodiversity, Faculty of Sciences, University of Antananarivo, Antananarivo, Madagascar. [12] Natural and Environmental Sciences, Regional University Centre of the SAVA Region (CURSA), Antalaha, Madagascar. [13] Doctoral School of Natural Ecosystems (EDEN), University of Mahajanga, Mahajanga, Madagascar. [14] Diversity Turn in Land Use Science, coordination office, Sambava, Madagascar. [15] Department of Agricultural Economics and Rural Development, Research Unit Environmental- and Resource Economics, University of Göttingen, Göttingen, Germany. [16] Ecology of Tropical Agricultural Systems, University of Hohenheim, Garbenstrasse 13, 70599 Stuttgart, Germany. ✉email: annemarie.wurz@uni-goettingen.de

Agricultural expansion and intensification are the main drivers of today's biodiversity crisis[1]. Increases in agricultural productivity are typically achieved at the cost of biodiversity[2,3]. Solutions to the resulting ecological-economic trade-offs are urgently needed, especially in tropical landscapes that undergo rapid transformation[4]. In order to prevent, halt and reverse the degradation of ecosystems, the United Nations has declared the years 2021–2030 as the Decade on Ecosystem Restoration[5]. Restoration is an approach, that can at least partially restore levels of biodiversity and ecosystem services[6]. In this context, degradation represents the decline in biodiversity and ecosystem services, and restoration aims to prevent, halt and reverse the degradation of ecosystems[6,7]. Agroforestry opens up promising opportunities for ecosystem restoration[8,9], but more system-specific knowledge is needed for an even wider implementation of win–win solutions. This is of particular importance for degraded land that has much reduced biodiversity and services provisioning[10], and which makes up large shares of tropical landscapes characterized by subsistence agriculture and shifting cultivation[11]. However, agroforestry may also result in biodiversity losses if established at the expense of forests[12,13]. Whether tropical agroforests contribute to halting deforestation or accelerate biodiversity declines thus depends on their land-use history, meaning whether they are established on open land (i.e. cropland, pastures, fallow, or degraded land) or by thinning of forest[14,15]. Surprisingly, despite decades of research in agroforestry, land-use history is not considered in most studies on tropical agroforestry[15]. Furthermore, the productivity of agroforestry systems is decisive for their overall biodiversity value, because low-yielding agroforestry systems need more land to meet the same demands as provided by high-yielding monocultures, possibly leading to more forest conversion and biodiversity loss on a landscape-level[16]. Here, we focus on vanilla agroforestry in Madagascar (Supplementary Fig. 1), a tropical biodiversity hotspot[17]. Madagascar has exceptionally high rates of endemism[18] but faces great challenges in biodiversity conservation and human development in the face of extreme poverty[19,20].

Madagascar is the biggest producer of vanilla worldwide[21], with a majority produced by smallholders[22]. The high world market price of vanilla over the last years brought great socioeconomic benefits for Malagasy smallholders, incentivizing the expansion of vanilla cultivation[22,23]. The hemi-epiphytic vanilla orchid is typically grown in agroforests on support trees in combination with shade trees[24,25]. Vanilla agroforests are either established through conversion of forest or on fallow land[15]. In contrast to the degradation of forests caused by forest-derived vanilla agroforestry, conversion of fallow land to vanilla agroforests can partially restore biodiversity and important ecosystem functions such as pest predation[26,27]. Fallow land, forming part of the shifting cultivation cycle for hill rice production, has increased in Madagascar[28] and is widespread in northeastern Madagascar[23,29]. As a consequence, Madagascar has lost 44% of its old-growth forest within the past six decades[30], generating a need for land-use solutions aligning conservation with agricultural production[31]. We quantified the effect of vanilla cultivation on multiple taxa including trees, herbaceous plants, birds, amphibians, reptiles, butterflies, and ants, and used yield data from 30 vanilla agroforests, to identify yield-biodiversity trade-offs. We assessed the biodiversity value of forest- and fallow-derived vanilla agroforest and compared it with old-growth forest, forest fragments, and fallow land (Supplementary Fig. 2). Here, we focus on the restoration of species richness while addressing also differences in species composition. We differentiate between overall species richness and endemic species richness to account for Madagascar's high share of endemic species and their

vulnerability to land use[32,33]. To identify biodiversity-friendly as well as profitable strategies of vanilla cultivation, we assessed environmental and management-related variables as drivers of yields and species richness.

## Results

**Biodiversity and vanilla yield.** Comparing species richness and vanilla yield (kg/ha) in forest- and fallow-derived vanilla agroforests, we found that higher vanilla yields were not associated with a decrease in the overall species richness, nor the richness of endemic species, for trees, herbaceous plants, birds, amphibians, and ants. Notably, the analyzed relationship of yield with biodiversity is likely mediated by underlying variables such as management and land-use history. Moreover, the overall and endemic diversity of all taxa combined (i.e., their mean normalized richness) was also not related to vanilla yields at the plot level (Fig. 1; Supplementary Tables 1 and 2). We found a negative relationship between vanilla yield and butterfly species richness (Fig. 1G, estimate = −0.179, p-value < 0.001) and endemic butterfly species richness (Fig. 2G, estimate = −0.104, p-value = 0.043), a positive relationship with amphibians (Fig. 1E, estimate = 0.110, p-value = 0.045), and, depending on land-use history, an either positive (Fig. 2F, forest-derived, estimate = 0.289, p-value = 0.028) or negative (Fig. 2F, fallow-derived, estimate = −0.145, p-value = 0.016) relationship with endemic reptiles. Species richness of trees (Fig. 1B), reptiles (Fig. 1F) and mean normalized richness of all combined taxa (Fig. 1A) were higher in forest-derived than in fallow-derived vanilla agroforests (Supplementary Table 12). Similarly, when looking at endemics, land-use history mattered for species richness of trees, herbaceous plants, birds, and ants and mean normalized endemic richness, with higher values in forest-derived compared to fallow-derived vanilla agroforests (Fig. 2).

**Effects of land-use history on biodiversity of agroforests.** The direction and magnitude of biodiversity responses at the local scale, i.e., the plot level, differed by land-use history and taxa. When compared with old-growth forest, we observed significant losses in species richness in forest-derived vanilla for birds (−38%), trees (−51%), and amphibians (−51%) as well as mean normalized richness of all taxa combined (−23%); whereas butterflies significantly gained species (+82%) (Supplementary Table 3). Herbaceous plants, reptiles, butterflies, and ants showed no significant difference between land uses (Fig. 1; Supplementary Tables 3–6). We also found significant losses in endemic species richness of amphibians (−57%), trees (−58%), birds (−69%), and mean normalized endemic richness (−47%) in forest-derived vanilla compared to old-growth forest. Gamma diversity was highest for old-growth forest across all taxa, except for the overall richness of herbaceous plants, butterflies, and ants as well as the endemic richness of butterflies (Supplementary Table 7). Compared with forest fragments, only butterflies significantly gained species (+122%) in forest-derived vanilla agroforests; losses or gains in species richness of all other taxa and of mean normalized richness were not statistically significant, also when focusing on endemic species (Supplementary Table 8).

When compared with fallows, we observed significant gains in overall species richness in fallow-derived vanilla agroforest only for trees (+149%) (Fig. 1; Supplementary Tables 4–6, S9). When looking at endemic species richness, we found significant gains for reptiles (+38%), and ants (+164%) as well as mean normalized endemic richness (+38%) when comparing fallow-derived vanilla agroforest to fallow (Fig. 2; Supplementary Tables 4–6, S9). Gamma diversity was higher for fallow-derived vanilla agroforest compared to fallow for all taxa, except for the

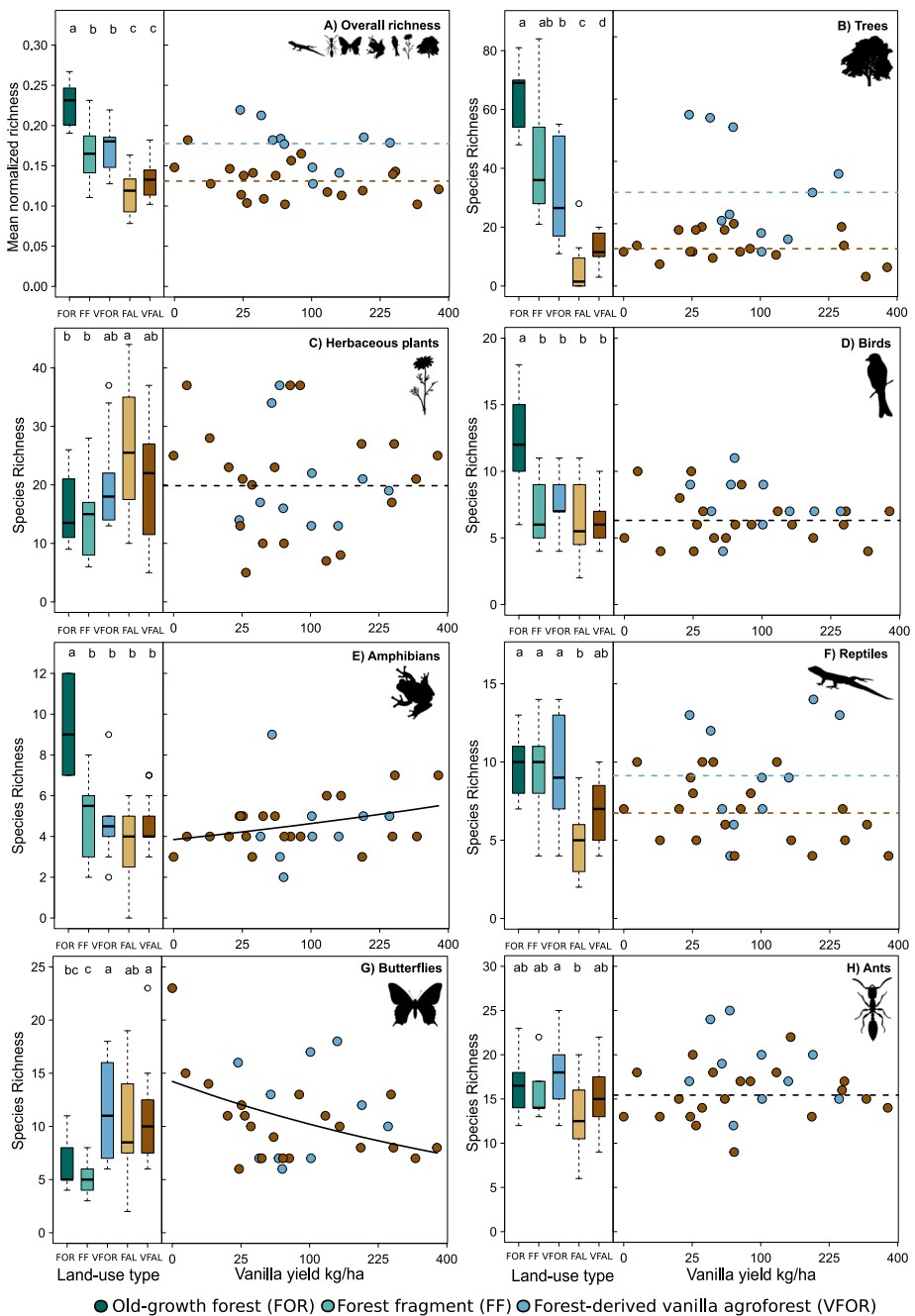

**Fig. 1 Overall species richness (mean normalized richness across all taxa) and individual species richness of seven taxonomic groups across land-use types and with increasing vanilla yield.** Shown are boxplots of plot-level mean normalized richness across taxa (**A**) and species richness of seven taxa individually (**B–H**) in old-growth forest (FOR = dark green), forest fragment (FF = light green), forest-derived vanilla agroforest (VFOR = blue), fallow (FAL = yellow) and fallow-derived vanilla agroforest (VFAL = brown). $n = 10$ for each FOR, FF and VFOR & $n = 20$ for each FAL and VFAL. The line inside the boxplot represents the median of each land-use type. The lower and upper boundaries of the boxplot show the 25th–75th percentiles of the observational data, respectively, and the whiskers show the 1.5 interquartile range. Outliers are shown as dots. Letters indicate significant differences between land-use types based on pairwise Tukey's honest or Wilcoxon significance tests (Statistical test results in Supplementary Tables 4–6). Scatterplots (VFOR = blue and VFAL = brown) show the relationship between plot-level mean normalized richness (**A**) and plot-level species richness of the seven taxa (**B–H**) with vanilla yield. Lines indicate model predictions (Statistical test results in Supplementary Tables 1 and 2). Horizontal dashed lines are intercept-only linear models (lines are based on the mean of the distribution). Solid lines indicate statistically significant relationships ($p < 0.05$). Two colored lines (VFOR = blue and VFAL = brown) are shown as dashed lines if land-use history was significant as an additive term but there was no significant relationship between species richness with vanilla yield. Solid colored lines indicate that the effect of vanilla yield was moderated by land-use history. Sample size: $n = 70$ plots for herbaceous plants, birds, amphibians, reptiles, butterflies, and ants; $n = 68$ plots for trees; $n = 30$ plots for vanilla yield. Note the sqrt-scale for vanilla yield/kg. Icons from phylopic.org (see Supplementary Table 19 for attributions).

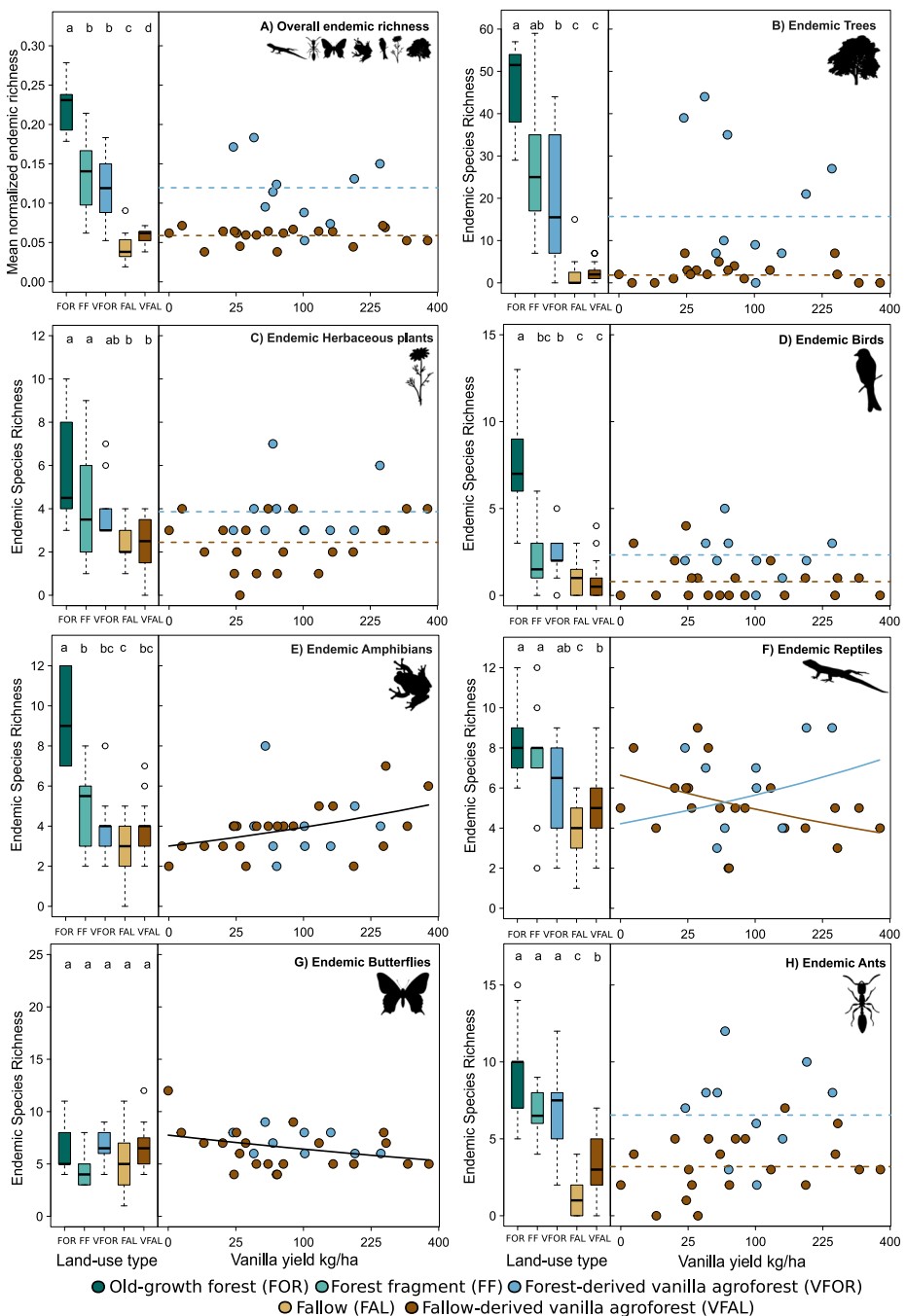

**Fig. 2 Overall endemic species richness (mean normalized endemic richness across all taxa) and individual endemic species richness of seven taxonomic groups across land-use types and with increasing vanilla yield.** Shown are boxplots of plot-level mean normalized endemic richness (**A**) and endemic species richness of seven taxa individually (**B**–**H**) in old-growth forest (FOR = dark green), forest fragment (FF = light green), forest-derived vanilla agroforest (VFOR = blue), fallow (FAL = yellow) and fallow-derived vanilla agroforest (VFAL = brown). $n = 10$ for FOR, FF, and VFOR & $n = 20$ for FAL and VFAL. The line inside the boxplot represents the median of each land-use type. The lower and upper boundaries of the boxplot show the 25th–75th percentiles of the observational data, respectively, and the whiskers show the 1.5 interquartile range. Outliers are shown as dots. Letters indicate significant differences between land-use types based on pairwise Tukey's honest or Wilcoxon significance tests (Statistical test results in Supplementary Tables 4–6). Scatterplots (VFOR = blue and VFAL = brown) show the relationship between plot-level mean normalized endemic richness (**A**) and plot level endemic richness of the seven taxa (**B**–**H**) with vanilla yield. Lines indicate model predictions (Statistical test results in Supplementary Tables 1 and 2). Horizontal dashed lines are intercept-only linear models (lines are based on the mean of the distribution). Solid lines indicate statistically significant relationships ($p < 0.05$). Two colored lines (VFOR = blue and VFAL = brown) are shown as dashed lines if land-use history was significant as an additive term but there was no significant relationship between endemic species richness with vanilla yield. Solid colored lines indicate that the effect of vanilla yield was moderated by land-use history. Sample size: $n = 70$ plots for endemic herbaceous plants, birds, amphibians, reptiles, butterflies, and ants; $n = 68$ plots for trees; $n = 30$ plots for vanilla yield. Note the sqrt-scale for vanilla yield/kg. Icons from phylopic.org (see Supplementary Table 19 for attributions).

overall richness of herbaceous plants, birds and butterflies as well as the endemic richness of trees (Supplementary Table 7). Notably, the proportion of the recorded species that are endemic to Madagascar varied strongly among the studied taxa, with 51% for trees, 20% for herbaceous plants, 61% for birds, 98% for amphibians, 74% for reptiles, 58% for butterflies, and 45% for ants, respectively. Taxa with high levels of endemicity are particularly prone to irrevocable biodiversity loss if land use negatively affects them. Sampling coverage across land-use types was on average 84% for all taxa, indicating a satisfactory sampling effort (Supplementary Table 10). In addition, the rarefaction curve for fallows based on species richness per sampling unit tended to reach an asymptote in fallows for amphibians, birds, reptiles, and trees (Supplementary Fig. 3). The rarefaction curves of ants, butterflies, and herbaceous plants did not reach an asymptote across all land-use types; thus, differences between land-use types may further increase if sampling effort is increased.

**Effects of land-use history on species composition inside agroforests.** Across all taxa, species composition changed significantly from old-growth forest to forest-derived vanilla agroforest (Supplementary Fig. 4, Supplementary Table 11). Forest fragments and forest-derived vanilla agroforests differed significantly in species composition for herbaceous plants, butterflies, reptiles, amphibians, and marginally for ants. Comparing fallow to fallow-derived vanilla agroforest, we found significant changes in the composition of trees, herbaceous plants, birds, reptiles, butterflies, and ants (Supplementary Fig. 4). Amphibians did not differ in species composition between fallow and fallow-derived vanilla agroforest.

**Effects of environmental and management variables on yield.** Vanilla yields varied widely and averaged at 105 ± SD 100 kg/ha (Supplementary Table 12). Vanilla yields increased with planting density (estimate = 2.901, SE = 0.415, $p$-value < 0.001) and vanilla vine length (estimate = 2.650, SE = 0.393, $p$-value < 0.001; Table 1; Supplementary Fig. 5 and Supplementary Tables 13 and 14). Moreover, vanilla yield tended to increase with labor input for hand pollination of vanilla flowers (estimate = 0.897, SE = 0.469, $p$-value = 0.069; Table 1). Importantly, vanilla yield was not related to canopy closure, slope, landscape forest cover, understory vegetation cover, soil characteristics, or elevation, suggesting a high intensification potential without the need for shade or vegetation removal (Supplementary Fig. 5). Furthermore, the age of vanilla plants did not influence vanilla yield. Lastly, vanilla yields also did not differ between fallow- and forest-derived vanilla agroforests (Tukey post-hoc test: estimate = 0.77, SE = 1.94, $p$ = 0.691; Supplementary Fig. 6).

**Effects of environmental and management variables on biodiversity.** To understand the effect of underlying yield-related management variables on biodiversity, we analyzed the relationship of species richness with four management and six environmental variables. We found trade-offs between yield-increasing variables (vanilla planting density and vanilla vine length) and overall and endemic species richness of trees and reptiles (Table 1, Supplementary Tables 15–18, Supplementary Fig. 7) but win-wins with endemic species richness of herbaceous plants.

Firstly, agroforests with higher vanilla planting density had lower tree richness (estimate = −0.155 SE = 0.066, $p$-value = 0.019) and lower endemic tree richness (estimate = −0.337 SE = 0.096, $p$-value < 0.001) but higher endemic herbaceous plant species richness (estimate = 0.176, SE = 0.063, $p$-value = 0.005. Secondly, vanilla vine length was related to overall fewer tree species (estimate =

−0.221, SE = 0.056, $p$-value = 0.001), endemic tree species (estimate = −0.553, SE = 0.104, $p$-value = 0.001) and reptile species (estimate = −0.0166, SE = 0.071, $p$-value = 0.019) as well as marginally fewer endemic reptile richness (estimate = −0.096, SE = 0.052, $p$-value = 0.068) at plot level. Notably, species richness of birds, amphibians, butterflies, and ants was driven by environmental and management variables unrelated to vanilla yields (Table 1, Supplementary Tables 15–18, Supplementary Fig. 7). Apart from differences in species richness due to elevation, slope and soil characteristics, we found that canopy closure, a structural parameter strongly affected by farmers' management decisions, was positively associated with species richness of trees (estimate = 0.284, SE = 0.082, $p$-value < 0.001), reptiles (estimate = 0.178, SE = 0.074, $p$-value = 0.016), endemic reptiles (estimate = 0.136, SE = 0.057, $p$-value = 0.017), endemic ants (estimate = 0.387, SE = 0.111, $p$-value < 0.010), and marginally with the species richness of endemic herbaceous plants (estimate = 0.145, SE = 0.078, $p$-value = 0.063). Additionally, a denser understory vegetation of shrubs and herbaceous plants was associated with a higher richness of endemic birds (estimate = 0.460, SE = 0.179, $p$-value = 0.010), but tended to reduce the richness of butterflies (estimate = −0.127, SE = 0.072, $p$-value = 0.077) and endemic butterflies (estimate = −0.109, SE = 0.049, $p$-value = 0.062). Landscape forest cover, mainly mediated by remaining forest fragments in the agricultural matrix, was positively associated with more species of trees (estimate = 0.270, SE = 0.074, $p$-value < 0.001), endemic trees (estimate = 0.827, SE = 0.107, $p$-value < 0.001), endemic herbaceous plants (estimate = 0.236, SE = 0.064, $p$-value < 0.001), endemic ants (estimate = 0.214, SE = 0.084, $p$-value = 0.011) and marginally reduced amphibian richness (estimate = −0.097, SE = 0.052, $p$-value = 0.062) in vanilla agroforests. Amphibian richness was lower on farms situated on steep slopes (estimate = −0.125, SE = 0.054, $p$-value = 0.021), whereas herbaceous plant richness was higher at higher elevations (estimate = 0.283, SE = 0.055, $p$-value < 0.001).

## Discussion

Here, we studied biodiversity–yield relationships in smallholder vanilla agroforests in Madagascar, a global biodiversity hotspot, with high pressure on the remaining old-growth forest and major sustainability challenges[19]. Vanilla is an important cash crop and a major export commodity of Madagascar that has the potential to lift tens of thousands of smallholder farmers out of poverty[34]. In contrast to common expectations of ecological-economic trade-offs[3,35], we show that increasing yields within the current range of vanilla agroforestry management practices (productivity benchmark [reference point for an average maximum of yield] in Madagascar: 350 kg/ha[36]) are not generally associated with biodiversity losses. Moreover, our study highlights the great potential of vanilla agroforestry to restore the biodiversity value of degraded and fallow lands, which are prevalent in the main vanilla production region of northeastern Madagascar[11,29]. This potential is underlined by the fact that 70% of vanilla agroforests in the study region are already fallow-derived[22]. With targeted incentives for establishing vanilla agroforests on fallow land, vanilla agroforestry could further contribute to land restoration. By providing an alternative income it may also prevent the degradation of the last remnants of old-growth rainforest through shifting cultivation[37]. This is particularly important as our study shows that old-growth forests harbor high levels of species richness and unique species composition, which underlines their conservation value.

While vanilla yield increased with greater vanilla planting density and longer vanilla vines, we found trade-offs of biodiversity with yield-related variables for trees, endemic herbaceous plants, and reptiles. However, the species richness of the

**Table 1 Overview of the direction of effects of environmental and management variables on yield and species richness across seven taxa (trees, herbaceous plants, birds, amphibians, reptiles, butterflies, and ants).**

| Predictor | Yield | Species richness |
| --- | --- | --- |
| Vanilla planting density (no/ha) | + | − Trees<br>− Endemic trees<br>+ Endemic herbaceous plants |
| Vanilla vine length (cm) | + | − Trees<br>− Endemic trees<br>− Reptiles<br>(− Endemic reptiles) |
| Vanilla plant age (yrs) | | |
| Pollination labor input (hrs/ha) | (+) | |
| Soil characteristics (PC1) | | − Endemic trees<br>+ Endemic reptiles<br>+ Butterflies<br>+ Endemic butterflies<br>− Endemic birds<br>(+ Herbaceous plants) |
| Canopy closure (%) | | + Trees<br>+ Reptiles<br>+ Endemic reptiles<br>+ Endemic ants<br>(+ Endemic herbaceous plants) |
| Slope (°) | | − Amphibians<br>(− Endemic amphibians) |
| Landscape forest cover (%) | | + Trees<br>+ Endemic trees<br>+ Endemic herbaceous plants<br>+ Endemic ants<br>(− Amphibians) |
| Understory vegetation cover (%) | | + Endemic birds<br>(− Endemic butterflies)<br>(− Butterflies) |
| Elevation (m) | | + Herbaceous plants |

Positive (+) or negative (−) effects are shown if statistically significant ($p < 0.050$). Symbols in parentheses indicate marginally significant relationships ($0.050 <= p < 0.100$). See Figs. S7, S8 for visualizations of relationships and Supplementary Tables 13 and 14 and 15–18 for statistical test results.

majority of taxa in vanilla agroforests was determined by non-yield-related management variables such as canopy closure and landscape forest cover, providing opportunities for smallholders to increase agricultural productivity. Importantly, these productivity increases do not come at the expense of biodiversity. Likewise, land-use history mattered for biodiversity but not for vanilla yields, indicating equal opportunities for profitable agroforestry on fallow land without further forest loss[38]. Higher landscape forest cover and higher canopy closure promoted endemic herbaceous plants and endemic species of trees, as well as endemic reptiles and ants in agroforests. Also, other studies from Madagascar have highlighted the importance of trees for biodiversity and identified the loss of canopy closure associated with species loss and community composition change across taxa[39,40]. Here, conservation of remaining forest fragments as well as farmer incentives for maintaining dense canopy structures is needed. Vanilla yield also tended to increase with labor input for hand pollination. While hand pollination is critical for achieving high vanilla yields[41], this finding needs to be interpreted carefully, since it may also reflect an overall increase of pollination input with a high number of vanilla flowers present. Surprisingly, vanilla age was unrelated to vanilla yields, as the two are commonly positively related[38].

Contrasting with the stable species richness of the majority of taxonomic groups, we found that agroforests with a high planting density of vanilla and long vanilla vines had reduced species richness of trees (overall and endemic) and reptiles as well astended to support fewer endemic reptile species. This indicates

trade-offs between conservation and production goals, particularly for trees, which represent a keystone structure on which other species depend[42]. Doubling planting density from 3000 to 6000 vanilla plants/ha or vine length from 300 to 600 cm, corresponded to a decrease in tree richness by 27% or 23%, respectively. By contrast, almost tripling planting densities to 8500 plants/ha or quadrupling to 1200 cm vine length lowered tree species richness by 55% and 52%, respectively. Thus, intermediate increases in planting density and vanilla vine length can represent a compromise for tree conservation and vanilla production. High variations in planting density, as well as additional effects affecting yields (e.g., labor input), may be the drivers of tree species decreasing with more and longer vanilla plants, but not with increasing yields. Furthermore, our study highlights that tree richness strongly depends on landscape forest cover as well as canopy closure which can be achieved by conservation of remaining forest fragments as well as farmer incentives for maintaining old trees and dense tree canopies. While the mechanisms underlying the negative relationship of vanilla planting density with tree richness seem obvious, more research is needed on the mechanisms behind the negative relationships between vine length and tree as well as reptile diversity.

Our study supports findings from other major agroforestry systems such as cacao, demonstrating no relationship between cacao yield and multiple plants and animal taxa in Indonesia[43] and Peru[44]. In contrast, a study on Cameroonian cacao agroforests found a negative relationship between ant richness and cocoa yields, suggesting trade-offs with biodiversity at high yield levels if shade trees are removed[45]. Vanilla smallholder agroforestry is not subject to similar trade-offs because the amount of shade vegetation is not related to vanilla yields[38]. However, vanilla has potential for intensification by increasing planting density and increasing vanilla vine length, which can be achieved by smallholder farmers without the need to reduce canopy closure. For example, doubling planting density from 3000 to 6000 vanilla plants/ha increases yields by 193%, that is, from 71 kg (2200 Euro/ha gross revenue) to 208 kg or 6400 Euro/ha at high vanilla prices of 2016[22]. Similarly, doubling vine length from 600 to 1200 cm increases yields by 191% (66 kg 2000 Euro/ha to 192 kg 5900 Euro/ha). Notably, planting density was generally low (mean = 3284 plants/ha; SD = 1444; Supplementary Table 12) in our study region, compared to intensively managed vanilla systems elsewhere. In Mexico, vanilla planting densities reach up to 5000 plants/ha in monocultures and up to 15000–20000 plants/ha in shade houses[46]. Generally, vanilla yields in Madagascar still have a high intensification potential[38].

Systematic reviews suggest that in about 80% of all studies, species richness in small-scale tropical agroforests is lower than in forests[47]. In line with our findings for trees, studies have highlighted that plants are more negatively affected by forest conversion than more mobile taxa like insects[48,49]. Nevertheless, agroforests can also support biodiversity levels similar to forests, if their transformation occurred recently and management remained extensive[50,51]. Vanilla agroforests, in contrast to coffee and cacao agroforestry systems, are generally extensively managed with highly variable yields, averaging at less than 500 kg green vanilla per hectare globally[52]. Malagasy vanilla is managed extensively and manually without inputs of fertilizers or pesticides[22]. The extensive management of vanilla as well as its setting in a diverse mosaic landscape with forest remnants contribute to their high biodiversity and species richness that for some taxa (e.g. butterflies, ants) can equal that of old-growth forest at the plot level[53,54].

Our findings confirm recent calls that land-use history needs more consideration in agroforestry research and management[15]. The increase of biodiversity in fallow-derived vanilla agroforestry

compared to fallows, particularly of endemics, presents considerable conservation opportunities in line with the goals of the UN Decade on Ecosystem Restoration (2021–2030) and the recent IPBES report[5,55,56]. In addition, we found compositional differences between fallow and fallow-derived vanilla agroforests for trees, herbaceous plants, birds, reptiles, butterflies, and ants, which may be explained by the tree regrowth and species colonizing fallow-derived vanilla agroforests. Once they are established, vanilla agroforests are unlikely to be transformed into other land-use types due to their high profitability[57] and thus present a leverage point for breaking out of the shifting cultivation cycle that degrades much of the agricultural land in Madagascar[57]. Vanilla agroforests offer longlasting opportunities for biodiversity and tree stand structures to recover[12,38], allowing associated biodiversity and ecosystem services to increase[57] and species composition to be partially restored compared to fallow land[54]. Tree regrowth in fallow-derived agroforests is particularly valuable to supplement and connect the few remaining forest fragments across the agricultural matrix[58]. In contrast, fallow land under shifting cultivation experiences repetitive burning[59], which limits the establishment and growth of trees[11]. However, ecological restoration also has to consider the value of fallow land to reconcile livelihood and conservation needs[60]. Fallow land can take different forms providing a wide array of benefits to people[61]. For example, right after subsistence rice production, fallows are often used for livestock grazing[11]. Indeed, the transformation of fallow land into agroforests can result in losses of provisioning ecosystem services (e.g., firewood, wild foods, timber)[61]. However, there is not yet a shortage of fallow land in northeastern Madagascar, and the loss of provisioning services through conversion to agroforests is readily offset by the associated benefits (e.g., cash crops, carbon storage)[61,62]. A heterogeneous landscape with multiple land uses is important to satisfy the needs of rural communities.

Madagascar has high levels of poverty, with around 65% of people depending on agriculture[63,64]. Inefficient land management and weak law enforcement are major challenges to biodiversity conservation and solutions to Madagascar's biodiversity crisis are urgently needed[20,65]. To guide restoration measures and sustainable intensification, efforts need to be supported through well-designed policies and economic incentives. Farmer training should emphasize that agroforests on fallow land are as productive as forest-derived ones and that high canopy closure does not conflict with high yields. Contract farming with sustainability standards (e.g., Fair Trade, Rainforest Alliance) or compensations schemes (e.g., Payments for Ecosystem Services) may promote agroforestry on fallow land and the use of endemic trees. Such contractual arrangements may not only favor conservation but also farmers through greater income stability as well as guaranteed minimum and premium prices[36,66].

Our study on Malagasy vanilla agroforestry is a prime example of how win-win solutions that combine high yields with high biodiversity can be achieved in tropical agriculture. While the last remnants of Madagascar's old-growth rainforest need strict protection to conserve their unique biodiversity and species incompatible with agriculture, fallow-derived vanilla agroforestry can support the restoration of biodiversity and important ecosystem services within agricultural lands and provide a profitable alternative to further expansion into old-growth forest. Management strategies in vanilla agroforests allow both yields and biodiversity to be increased. Thereby, vanilla agroforestry opens up great opportunities for economically and ecologically sustainable land management in Madagascar, aligning with the UN ecosystem restoration goals in this outstanding tropical biodiversity hotspot.

## Methods

**Ethical statement.** Ethics approval was obtained for this study from the ethics committee of the University of Goettingen (Chair: Prof. Dr. Peter-Tobias Stoll) under the reference number 17./04.22Wurz.

**Study area.** All plots were situated in northeastern Madagascar in the SAVA region (Supplementary Fig. 1). The natural vegetation is tropical lowland rainforest, but deforestation rates are high[30,67].

The region is globally and nationally one of the most biodiverse places with high levels of endemism[17,68]. Forest loss is mainly driven by slash-and-burn shifting hill rice cultivation[58]. The region is characterized by a warm and humid climate with an annual rainfall of 2255 mm and a mean annual temperature of 23,9 °C (mean value of 60 plots extracted from CHELSA climatology[69]). Vanilla is the main cash crop in the SAVA region, making Madagascar the main vanilla producer globally[21,22]. Vanilla prices have shown strong fluctuations over the past years, with a price boom between 2014 and 2019 triggering an expansion of vanilla agroforestry in the region[22,23].

**Study design.** We selected 10 villages based on the 60 villages selected within the Diversity Turn in Land Use Science project[22] (Supplementary Fig. 1). We selected the villages based on the list of villages for our study region from official election lists which listed all villages within a fokontany individually[22]. Village boundaries, demographics, infrastructure were defined based on a rapid survey with the village chief. Among the 60 villages, we considered all villages without coconut plantations, with less than 40% water (river, sea, and lakes) to avoid a strong influence of water elements and with forest fragments and shifting cultivation present within a 2 km radius around the village. Two of these 17 villages overlapped within a 2 km radius of the villages, thus we randomly selected one of them, resulting in 14 villages. We visited these 14 villages in a randomized order and stopped after we found 10 villages which fulfilled the necessary criteria (all land-use types present, willing to participate). In each of the 10 villages, we selected three vanilla agroforests, one forest fragment, and two fallows. Overall, we studied 60 plots across 10 villages and 10 plots in one protected old-growth forest (Marojejy National Park). All plots had a minimum distance of 260 m and a mean minimum distance of 794 m (SD = 468 m) to each other. Plot elevation ranged between 10 and 819 m.a.s.l. (mean = 205 m, SD = 213 m; Supplementary Table 20).

**Plot selection.** In each of the 10 villages, we selected three vanilla agroforests with low, medium, and high canopy closure, respectively, covering a within village canopy cover gradient. To refine our vanilla agroforest classification, we used interviews with the plot owners to categorize all vanilla agroforests based on land-use history into fallow- and forest-derived agroforests[15]. Forest-derived vanilla agroforests are established within forest fragments, which have been manually thinned of dense understory vegetation. Fallow-derived vanilla agroforests are established on formerly slashed and burned plots, where vegetation has been cleared for hill rice production (shifting cultivation system locally called *tavy*). Out of our 30 vanilla agroforests, 20 vanilla agroforests were fallow-derived and 10 vanilla agroforests were forest-derived, roughly matching the proportion of fallow- and forest-derived vanilla agroforests across the study region (70% are fallow-derived vanilla agroforests, 27% are forest-derived vanilla agroforests and 3% of unknown origin[22].

In addition to vanilla agroforests, we selected one forest fragment in each village. Forest fragments were located inside the agricultural landscape and were remnants of the once continuous forest; these fragments are frequently used for natural product extraction. Forest fragments have not been burned or clear cut in living memory, yet the ongoing resource extraction results in a much simplified stand structure and fewer large trees compared to old-growth forest[12]. Furthermore, we chose one herbaceous and one woody fallow in each of the 10 study villages. Both fallow types form part of the shifting hill rice production cycle and represent the fallow period at different stages after the crop production. Herbaceous fallows have been slashed and burned multiple times with the last cultivation cycle at the end of 2016, one year prior to the first species data collection in 2017, and thereafter left fallow[11]. The continuous succession of herbaceous fallows turns them into woody fallows with the domination of woody plants including shrubs, trees, and sometimes bamboo. Our 10 woody fallows have last burned 4–16 years before data collection. In this study, we combine both herbaceous and woody fallows into the category "fallow". Generally, fallows occur in different forms in the study region. The characteristics of fallows depend on the frequency of past fires and the length of fallow periods in between crop cultivation[11]. Frequent burning results in a loss of native and woody species and a dominance of exotic species and grasses[11]. In later fallow cycles, fern species increasingly appear[11].

Due to the commonly repeated slashing and burning, secondary forests are very rare in the study region. Shifting cultivation prevails in Madagascar[70], because it is an important option for people to grow food because means for agricultural intensification are scarce. According to our baseline survey (performed in 60 villages in our study region), 90% of the interviewed farmers grow rice for subsistence in addition to growing vanilla[22]. Out of this sample, 64% of farmers grow rice in irrigated paddies and 26% of farmers use shifting cultivation.

We also studied 10 plots at two sites in Marojejy National Park, the only remaining, continuous old-growth forest at a low altitude in our study area[71]. We chose accessible old-growth forest plots with a minimum distance of 250 m from the forest edge. Five of the 10 old-growth forest plots were located in Manantenina Valley, the other five old-growth forest plots were situated in the eastern part of Marojejy National Park, called Bangoabe area. Illegal selective logging has occurred in some parts of the park. During our plot selection, we avoided sites with traces of selective logging.

**Land-use history classification**. To collect information on the land-use history or farm history, interviews with farmers are common[72,73]. We did interviews with the plot owner. Questions on land-use history were binary (forest-derived or fallow-derived) and did not include information on the detailed land-use history (e.g. frequency of burning, past crop systems). Thus, we consider this selfreported data very reliable. The land-use categorization derived by farmers was confirmed by our visual plot inspections (forest-derived vanilla agroforests do have a quite distinctive vegetation structure compared to fallow-derived vanilla agroforests). Additionally, data on tree species composition and soil characteristics show evident differences between the categories and back up the binary land-use history categorization. Analysis of tree species composition showed that fallow- and forest-derived vanilla agroforests differ significantly in tree species composition[12]. Soil analysis (see Fig. S9) showed that our fallow-derived vanilla agroforests are associated with fertility-related variables such as an increase in calcium, pH, nitrogen, and phosphorus, which is common after slas-and-burn agriculture[74,75].

**Plot design**. We collected species data on plots with a radius of 25 m (1964 m², 0.1964 ha). We established our circular plots in a homogeneous area of the land-use type or forest. Adjacent land uses were usually different because farmers generally own small-scale land with a mean size of 0.66 ha (mean size of agroforests). We assessed vanilla plant data (yield, vine length, vine age, planting density) on 36 vanilla *pieds* on each of 30 circular vanilla plots (Supplementary Fig. 8). We defined one vanilla *pied* (foot in French) as the combination of a vanilla vine and a minimum of one support tree. The 36 vanilla *pieds* were evenly selected in each of the circular plots based on a sampling protocol to ensure comprehensive and unbiased sampling. We chose vanilla *pieds* independent of age, length or health condition. We marked the 36 selected vanilla *pieds* per plot with a unique barcode to assess vanilla yield (April 2018) and other plant health variables on the same plant (not used in this study). However, for 37 vanilla *pieds* (out of a total of 1080 marked vanilla *pieds*), the barcodes were lost or unreadable and we selected a new plant closest to the original position (independent of age, length, or condition) and marked it with a new unique barcode. We measured the size of the vanilla agroforest by walking with the agroforest owner and a hand-held GPS device at the perimeter of the plot.

**Vanilla planting density**. We counted each vanilla *pied* on each 25 m circular plot by dividing the plot in four-quarter segments. We calculated the area of each 25 m radius plot including slope correction and calculated vanilla planting density (vanilla *pieds* per hectare) by dividing the number of vanilla *pieds* by the slope-corrected plot area.

**Vanilla yield**. We measured yield on 30 vanilla plantations (10 forest-derived vanilla plantations and 20 fallow-derived vanilla plantations); three in each of our 10 study villages. We measured vanilla yield on a total of 36 vanilla *pieds* between March and April 2018. We assessed the vanilla yield before harvest to ensure an accurate yield assessment due to two reasons. Firstly, vanilla pods are commonly harvested successively due to their differing pollination date and maturity requiring multiple visits over several weeks. Secondly, theft of vanilla pods is commonplace around harvest time. We, therefore, estimated the weight of the on-plant-hanging vanilla pods by measuring pod volume and relating this to a prior established volume–weight correlation. This is possible because vanilla pods only grow in length and width in the first 8 weeks of their development[76]. Our yield assessment consisted of one interview part with the plot owner and one measurement part. The interview part included questions about the occurrence of theft and early harvest on the plantation. During the measurement part, we assessed the number, diameter, and length of all vanilla pods. We measured vanilla pod length with a ruler starting at the junction of stem and pod until the tip of the pod without considering the bending of the pod. We measured the diameter at the widest part of the pod using a caliper. We firstly calculated pod volume based on the standard volume cylinder formula using the measured diameter (cm) and length (cm): $V = \pi r^2 h$.

Secondly, we calculated the weight (g) of each pod by using the linear regression equation ($y = bx + a$) of a weight–volume correlation of 114 vanilla pods from 114 different agroforests (weight, length, and diameter of these 114 green vanilla was assessed post-harvest in 2017). We calculated the weight of all measured pods of the harvest in 2018 based on the formula:

$$volume = \pi(diameter(mm)/20)\char`^2 * length(cm)$$

Here, we divided the pod diameter (mm) by 20 to obtain the radius and to transform millimeters to centimeters. Weight was defined as volume*0.5662 + 0.9699. No vanilla pods were stolen or already harvested on our

36 vanilla *pieds* and hence we did not need to account for it in our vanilla yield calculation.

**Vanilla vine length**. We assessed vanilla vine length for all 36 vanilla *pieds* (same vanilla *pieds* as used for the yield assessment) on each plot by measuring the total length of the vine from the lowest to the highest part with a measuring stick. If the vanilla vine was looped on the support tree (= vanilla vine is hanging in multiple loops on the support tree), we measured from the top height of the looping of the vanilla vine until the lowest height of the vine. At the medium height of the vanilla vine, we counted the number of times the vanilla vine passed through. We calculated the total length of the liana by multiplying the maximum height of the vanilla vine by the number of times the vine passed through the middle. In some cases, the vanilla vine looped at two different heights, we thus considered the middle between the two looping heights as the top height. If vanilla vines grew on two different support trees, we considered them as one vanilla *pied* if support trees were <30 cm apart. If the distance between both support trees exceeded 30 cm, we considered only the support tree with the most vines for the measurement.

**Pollination labor input**. We performed a longitudinal survey with the plot owners of our 30 vanilla agroforests from October 2017 to October 2018. The questionnaire was pre-tested with 30 farmers in September 2017. All participants were trained by the four research assistants on how to use pictogram-supported questionnaires. Subsequently, a feedback workshop was held to adapt the pictograms and optimize the entire questionnaire. The pictograms had to be filled every day as a diary. Besides pollination labor input, we assessed the time spent on plantation establishment, planting, weeding, pruning, plantation safeguarding, harvesting, preparing (fermenting, drying, sorting), and selling of vanilla (not considered in this analysis). Every fortnight, trained assistants visited farmers to collect the diary questionnaires. Data entries that appeared unusual were verified with the farmers by the assistants. The diary questionnaires included questions on family labour input for pollination as well as other agricultural activities, such as weeding, harvesting, curing of vanilla, and others.

We decided to use pollination labour input as the only variable of labour input in our analysis for the following two reasons. Firstly, pollination is the most important and labour intensive part of the production[77]. Secondly, pollination has a defined time frame because vanilla only flowers between October and December[22]. Thus, the hours of pollination labour input are easy to disentangle and define for the farmers. In contrast, other tasks such as weeding, pruning, and planting often happen continuously and in parallel and are thus harder for the farmer to depict in working hours. Pollination is a labor-intensive activity as every vanilla flower is pollinated manually and requires agricultural know-how. Thus, the working hours can be related to the number of flowers or/and the worker's know-how. We calculated pollination labour input per hectare by summing all working hours (Oct 2017–Oct 2018) and dividing it by the slope-corrected agroforest size. For three out of 30 vanilla agroforests pollination labour input was missing; we thus used the mean value of all 30 agroforests for the three missing values.

The household head who filled the pictograms received 10,000 Ariary (roughly 2.50€) per month. Pictograms are drawings made by a local artist which visually describe each of the working steps of vanilla cultivation (e.g. planting vanilla vine, weeding plantation, pollination). The amount of compensation was recommended as a reasonable compensation by locally experienced Malagasy project members. The sum was handed out by the local research assistants at the end of each month. All participants of the surveys were informed that participation is voluntary, that they can leave the survey anytime, and that all data is anonymized, i.e., no personal data will be published or shared with third parties. Guidelines of "Good Scientific Practice" by the University of Göttingen were adopted (adapted based on the recommendations of the codex for good scientific practice from the DFG, German Research Foundation; https://www.unigoettingen.de/en/good+scientific+practice/567647.html).

The interview methodology (i.e. informed consent by the test persons as well as the questionnaires) was evaluated by the ethics committee of the University of Goettingen (https://www.unigoettingen.de/en/534983.html) and complied with their principles of the Higher Education Act of Lower Saxony (NHG) and the constitutionally protected right of academic freedom (Reference number: 17./04.22-Wurz). Co-author H.H. designed the survey and trained the local assistants with B.F. until the assistants were able to enter data in a consistent and standardized manner. F.A., F. S. B., and the trained assistants conducted the interviews. The research assistants collected the data bi-weekly but visited the households weekly. F.A. and F. S. B. checked entry data bi-weekly and clarified false entries. Our research assistants were recruited mainly from the student population of the regional CURSA university center in Antalaha, which is located in the research area. The students speak the local Malagasy dialects. Questionnaire in the original language, used pictograms, and the reporting sheet for farmers are available on Open Science Framework: https://osf.io/z5uxs/?view_only=1bd699c5cda64023963e058254a33eec.

**Vanilla plant age**. We assessed vanilla plant age by asking the farmer for each of the 36 vanilla *pieds* per plot. The Malagasy field researchers Evrard Benasoavina, Thorien Rabemanantsoa, and Gatien Rasolofonirina walked with the farmer to each vanilla

plant (see the question "Ask farmer: How many years ago was the liana at this pied planted?" in uploaded original interview (yield assessment) on Open Science Framework: https://osf.io/wa2xn/?view_only=1bd699c5cda64023963e058254a33eec). Here, the age referred to the vanilla vine but not the support tree. In preparation for the farmer interviews, we prepared handouts in Malagasy language to inform the farmers about the scientific goals and the content of our data collection (see handout on Open Science Framework: https://osf.io/ndrxg/?view_only=1bd699c5cda64023963e058254a33eec).

**Canopy closure**. We measured mean canopy closure at five subplots of our circular plots by taking hemispherical images with a Nikon D5100 camera, equipped with a Sigma Circular Fisheye (180°) 4.5 mm 1:2.8 lens. The camera was fixed on a tripod at 2.4 m height above vanilla support trees and understory vegetation. We selected the images with the best contrast of sky and vegetation using the histogram-exposure protocol and calculated canopy closure using a minimum thresholding algorithm[78,79].

**Slope and elevation**. We used the 30 m-resolution digital surface model "ALOS World 3D" by Japan Aerospace Exploration Agency (JAXA) to assess the mean slope and the mean elevation of each plot.
For all values, we applied slope correction[80].

**Landscape forest cover**. We calculated forest cover in a 250 m radius around each plot center based on binary forest cover data from 2017 with a 30 m resolution[30] and the R-package raster[81]. To reliably identify forest, tree cover maps and satellite imagery with a tree cover threshold of 75% were combined[30]. Here, agricultural lands such as tree plantations are excluded by combining historical forest maps with up-to-date forest cover change maps[82]. We chose a 250 m radius as a compromise between mobile and immobile taxa.

**Understory vegetation cover**. We estimated the vegetation cover (percentage woody and herbaceous cover) visually for the 0–2 m layers in % of five subplots on each plot (located in the plot center and at 16.6 m from the center in each cardinal direction) and calculated the mean understory vegetation cover per plot. We did not consider vanilla *pieds* in the estimation of the understory vegetation cover.

**Soil characteristics (PC1)**. We took soil samples with a MacFadyen soil corer (5 cm diameter, 295 ml, 0–15 cm depth). We divided the plots into eight subplots, four subplots at 8.3 m distance to the plot center (inner area) and four subplots at 16.6 m distance to the plot center (outer area). In total, we collected four cores in the inner and outer area each, resulting in two mixed soil samples per plot. We stored each soil sample in a zip-lock bag until laboratory analysis. In the laboratory, we measured pH ($H_2O$) with the fresh soil samples using 1:10 humus/water suspension after 24 h of equilibration. We measured pH (KCl) by adding 1.86 g KCl. Mean pH values by plot were calculated in logarithmic and back-transformed by using exponential. The remaining soil was dried at 70 °C and ground. We measured organic carbon ($C_{org}$) (mmol/g dry soil), total carbon (C) (mmol/g dry soil) and nitrogen (N) (mg/g dry soil) concentrations, and organic C-to-N ratio (mol/mol) by using the C/N elemental analyzer (Vario EL III, elementar, Hanau, Germany). Additionally, we determined effective cation exchange capacity in µmol/g dry soil of potassium (K), magnesium (Mg), calcium (Ca), iron (Fe), manganese (Mn), hydrogen (H), and aluminum (Al) by digesting oven-drier soil material in 65% $HNO_3$ at 195 °C for 8 h. Samples were analyzed with inductively coupled plasma optical emission spectrometry (ICP-OES) (Optima 3000 XL, Perkin Elmer, USA). We calculated the total effective cation exchange capacity (µmol/gTB) by the sum of H, P, Mg, Ca, Fe, Mn, and Al and total base saturation (%) as the sum of K, Mg, and Ca. Extractable phosphorus (P) Resin (µmol/gTB) was measured using resin bags, which were placed in a soil–water suspension. Then, P was re-exchanged with NaCl and NaOH solutions and quantified colorimetrically after blue-dyeing.
Due to the high number of soil characteristics measured and possible multicollinearity, we calculated Spearman correlations using the cor and corrplot function from the corrplot R-package[83]. Based on collinearity, we excluded total effective cation exchange capacity, total base saturation, total C, organic C, effective cation exchange capacity of Mg, effective cation exchange capacity of pH percolate, and the effective cation exchange capacity of H percolate (Correlation matrix, Supplementary Fig. 9). We used the remaining variables, i.e. effective cation exchange capacity of Ca, K, Al and Mn, pH(KCl), total N, resin P, and organic C-to-N ratio to perform a principal component analysis (PCA) using the R-packages ggbiplot[84] and factoextra[85]. The soil PC1 (explained 45%) was mostly related to effective cation exchange capacity (µmol/gTB) of Ca, and K as well as pH(KCl), nitrogen (mmol/gTB), P(resin) (µmol/gTB) and the organic carbon–nitrogen ratio (mol/mol) while the soil PC axis 2 (explained 21%) was related to exchange capacity of Mg and Al and the Corg-to-N ratio (Supplementary Fig. 10). The coordinates of PC axis 1 were used as a proxy of soil characteristics for further analysis.

**Trees**. We sampled trees on all land-use types except herbaceous fallows between September 2018 and January 2019[12]. Access was denied to two fallow-derived

vanilla plantations, resulting in 58 plots assessed overall (including 28 vanilla agroforests). We did a full inventory of all trees with freestanding stems of ≥8 cm diameter at breast height in each plot. This included trees, arborescent palms, herbs, and tree ferns but excluded lianas. We identified tree species with the help of a local tree expert (Chrysostome Bevao) and a taxonomic expert (Patrice Antilahimena) from Missouri Botanical Garden (Antananarivo, Madagascar). We derived information on origin and endemism for each species from the Tropicos Madagascar Catalog[86]. Voucher specimens are kept at the National Herbarium Tsimbazaza, Antananarivo (TAN) and the herbarium of the University of Mahajanga. Out of the 454 assessed species in this inventory, 276 (51%) were endemic to Madagascar.

**Herbaceous plants**. We sampled herbaceous plants in eight subplots of 4 $m^2$ each (32 $m^2$ overall) between September 2018 and December 2019. In each subplot, we assessed vascular plant species without apparent wood at maturity[40]. We accounted for the possible seasonality variation of the plant phenology by sampling one village after another. In each village, we collected data on each land-use type except for the old-growth forests. Hence, the observations for each land-use type cover the study period along with the possible phenology variations. We determined each species' endemism status from the Tropicos Madagascar Catalogue[86]. We stored all herbarium specimens at the Plant Biology and Ecology Department at the University of Antananarivo in Madagascar. From the 299 species assessed in this study, 59 species (20%) were endemic to Madagascar.

**Birds**. We sampled birds during two 40 min point counts per plot with two observers per point count[87] following a commonly used standardized method[88]. In all villages, we conducted one point count between September and December 2017 with co-author D.M. as the main observer and co-author R.A. as a second observer. The second point count was done between August and December 2018 with Eric Rakotomalala as the main observer and the co-authors D.M. or S.D. as the second observer. We exchanged the order of plot visits in the second year to minimize seasonal bias. For old-growth forest, we did all point counts in 2018; one in August 2018 with E.R. as the main observer and D.M. as the second observer, and a second count in December 2018 with E.R. as the main observer and S.D. as a second observer. E.R. and D.M. are both experienced birders and familiar with the encountered bird species due to previous fieldwork in Madagascar. S.D. and R.A. were trained prior to fieldwork to recognize calls and morphology of Malagasy bird species. The second observer was responsible for entering of data and supporting the identification. Main observers were responsible for spotting, hearing, and identifying the birds. On 63 plots we started one-point count around sunrise and one at least one hour after sunrise; on 7 plots this alteration was not possible due to logistical constraints. After arriving at the plot center, we waited for a minimum of three minutes to allow the birds to settle. We noted the conditions including rain (no rain, drizzle, light rain, heavy rain) and wind (Beaufort 1–12[89]) before each point count. We only started point counts under good weather conditions, meaning no rain and wind equal to or less than Beaufort 4. If weather conditions deteriorated for more than 10 min from the start of the point count, we aborted the point count and started again later or the next day under better conditions. For calculating bird species richness per plot, we disregarded observations only in flight and outside the 25 m radius of plots. We defined species as endemic if only occurring in Madagascar according to BirdLife species fact sheets[90]. Overall, we assessed 51 bird species of which 31 species (61%) were endemic to Madagascar.

**Amphibians and reptiles**. We sampled amphibians and reptiles using repeated time-standardized search walks for 45 min by two observers[91]. We visited each plot both during the day and at night both during the driest (one nocturnal and one diurnal search between October and December 2017; one nocturnal and one diurnal search between August and December 2018) and the wettest period (one nocturnal and one diurnal search between January and April 2018 or in February 2019). We did so during the driest period and the wettest period. We systematically walked the circular plot in a zigzag pattern always starting from the West part toward North, East, South, and end in West to avoid counting twice the same individual during observation in one of the plots. We actively checked microhabitats to detect individuals hiding therein (e.g., individuals hiding under rocks, in leaf axils, tree barks, tree holes, leaf litter, or deadwood). When encountering an individual, we stopped the standardized search time and identified the individual[92]. We identified individuals based on morphological characteristics with the help of field guides[93–95]. We took DNA samples to determine the species for those individuals that proved difficult to identify using morphological characteristics only. To retrieve a DNA sample, we collected muscle or toe clips as tissue samples, conserved in 90% of alcohol. We stored DNA samples at the Evolutionary Biology laboratory at TU Braunschweig. We also took photos of specimens that we did not identify to species level (ventral, back, and flank view). Until release, we kept them in a ventilated bag to retain moisture. We released all specimens after completing the full-time-standardized search. We categorized endemic reptiles and endemic amphibians as species/morphospecies only occurring in the country of Madagascar[96,97]. We found in total 58 amphibian species of which 57 species (98%) were endemic. In terms of reptiles, we found 61 species of which 74%

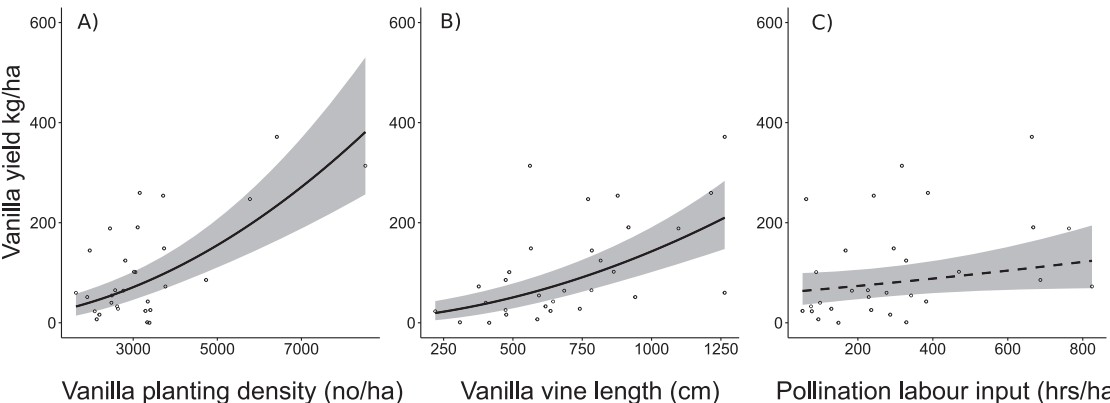

**Fig. 3 Management and environmental variables influencing vanilla yield in 30 vanilla agroforests in north-eastern Madagascar based on a linear mixed-effect model with yield sqrt-transformed.** Dots are raw data and solid and dashed lines indicate statistically significant ($p < 0.050$) and marginally significant effects ($0.050 < p < 0.100$), respectively. Trend lines show average values of the backtransformed model predictions of the final model after using likelihood ratio tests using maximum-likelihood estimation, shaded areas indicate 95% confidence intervals. The full model included: vanilla planting density, pollination labour input, vanilla vine length, vanilla plant age, soil characteristics, canopy closure, slope, landscape forest cover, understory vegetation cover, and elevation. The final model included: vanilla planting density, pollination labour input, vanilla vine length. Statistical test results in Supplementary Tables 6 and 7.

(45 species) were endemic to Madagascar (and 7 species without defined endemism level).

**Butterflies**. We sampled butterflies with fruit traps and time-standardized netting between August and December 2018[53]. We baited fruit traps with fermented bananas and deployed the cylindrical nets for 24 h. Before deployment, we fermented bananas for 48 h in an air-tight container.

On each plot, we installed a total of 8 fruit traps. We deployed four fruit traps at 16.6 m distance from the plot center in the four main cardinal directions and the other four fruit traps at 20 m distance from the center in the four intercardinal directions. We used fish lines covered with vaseline creme to hang the fruit traps. The Vaseline prevented ants to intrude on the fruit traps. In addition, we avoided any contact with branches on the fruit traps. We caught butterflies with a fruit trap with a 20 cm Cone Opening (90 cm long hanging 1.5 m above the ground. On plots without trees, we installed fruit traps on a support stick (in rice paddy and herbaceous fallow). During the 30 min time-standardized netting, we caught butterflies within an imaginary 2 m wide box to each side of the net while walking at a slow and steady speed in a zig-zag to equally cover the plot area. The net had a circular frame with a nylon mesh on a 1.5 m telescopic handle. We performed the timestandardized netting in dry and low-wind conditions only, either in the morning (8 a.m. to 12 p.m.) or in the afternoon (1 p.m. to 5 p.m.). We then collected and dried all captured butterflies and identified them to species level in the laboratory (moths excluded). Identification was done by Annemarie Wurz, David Lees and Sáfián Szabolcs. We categorized endemic butterflies as species/morphospecies only occurring in the country of Madagascar and updated with expert consultation by David Lees[98]. All identified specimens remain at the insect collection in the Department of Crop Sciences, section Agroecology, University of Göttingen, Germany. Overall, we found 84 butterfly species of which 49 species (58%) were endemic to Madagascar.

**Ants**. We sampled ground-foraging ants using bait and pitfall traps[54]. We conducted the sampling in all villages between October and December 2017, and in the old-growth forest in August and December 2018. We established five sampling stations per plot: one at the plot center, and four at 16 m distance from the plot center; one in each cardinal direction. We then set bait and pitfall traps 10 m apart at each sampling station. We baited the bait traps using sardine and sugar on two white flat plastic plates with a diameter of 13 cm and placed the two plates about 5 cm apart. We left the baited traps for 30 min before collecting ants for 30 s. We buried the pitfall traps (plastic cups of 9 cm top diameter, 11 cm depth, and 6 cm bottom diameter) in the soil and filled them one-third with 70% alcohol and a few drops of soapy water. We emptied the pitfall traps after 48 h and identified ants to species/morphospecies level in the laboratory. Identification was done by Anjaharinony Rakotomalala, using available identification keys[99–101]. Cross-checking of the identification of the species was done with expert consultation by Jean Claude Rakotonirina (species of *Leptogenys*), Nicole Rasoamanana (species of *Camponotus*), and Manoa Ramamonjisoa (species of *Tetramorium*). We defined endemic ant species as those species only present in the country of Madagascar[102]. We stored voucher specimens at Madagascar Biodiversity Center, Antananarivo, Madagascar (MBC). We recorded in total 123 ant species of which 55 species (45%) were endemic to Madagascar.

**Statistical analysis**. We performed all statistical analyses in R version 3.6.3[103]. To assess the overall species richness across all taxa, we calculated their mean normalized (endemic) richness. To ensure the equal weight of all taxa despite differences in the range of species richness per taxa, we normalized species richness measures with *Min−Max Scaling*, which is a common method in multi-taxa studies to compare species richness across different taxa and to calculate their overall richness[104–106]. Using this method, the data are linearly transformed to $y = (x − \min(x))/(\max(x) − \min(x))$, where $x$ is the set of observed values of species richness. As a result, the normalized species richness ($y$) then ranges between 0 and 1, with 0 referring to the lowest observed value and 1 to the highest observed value, respectively. We then calculated the average of the normalized values to obtain the mean normalized species richness, i.e., the overall richness of all taxa at the plot level. Our results were robust (no relationship between biodiversity and yield) independently of the metric used, i.e., normalized richness or multidiversity.

We investigated the relationship of species richness or normalized (endemic) richness with vanilla yield in 30 vanilla agroforests using glmmTMB models[107] or a linear mixed-effects model[108], respectively. We used glmmTMB due to its bigger flexibility (especially in the case of zero inflation) and its higher speed when using multiple fixed effects as well as random effects[107]. We treated vanilla yield (sqrttransformed) in interaction with land-use history (fallow vs. forest-derived) as an explanatory variable and site (village or old-growth forest site) as a random effect. We scaled and sqrt-transformed vanilla yield due to a few high-yielding plots inflating the data distribution.

We assessed the environmental and management-related covariates of species richness with a glmmTMB model[107] and vanilla yield using a linear mixed effect model[108] with yield sqrt-transformed. We tested for correlation among the covariates using the Spearman coefficient (Supplementary Fig. 11). For both models (species richness and vanilla yield) we used canopy closure, soil characteristics, slope, landscape forest cover, understory vegetation cover, elevation, planting density, pollination labour input, vanilla vine length, vanilla plant age as explanatory variables. We added site as a random effect and scaled all explanatory variables with the *scale* function. The function divides the centered columns (value per land-use type−mean value across land-use types) by their standard deviation across all land-use types. For the variable pollination labour input, data from three plots was missing. To avoid omitting these plots from the analysis, we performed a linear regression of available household labour with pollination labour/ha and found a good correlation between the two ($p$-value: 0.005, estimate = 52.82, SE = 17.38, Supplementary Fig. 12). Based on the number of household members (for which we have available data for the three missing plots), we then predicted expected pollination labour input for the three missing plots, using the estimated relationship of the two from the mentioned regression.

For all models assessing the environmental and management-related covariates of species richness, we used likelihood ratio tests using maximum-likelihood estimation[109] to assess the statistical significance of individual variables. Specifically, we compared the full model versus a reduced model without the individual variable (single term deletion). The final model included all explanatory variables with residual deviance $\chi^2 < 0.05$ ("chi-squared value") in the model comparison with the full model.

We assessed differences in species richness or normalized (endemic) richness between forest, forest fragment, forest-derived vanilla agroforests, fallow-derived

vanilla agroforest, and fallows by fitting a glmmTMB model[107] or a linear mixed-effects model[108], respectively. We used a glmmTMB model for all analyses with count/discrete data as a response. For mean normalized richness, we used a linear mixed-effects model due to the continuous data. Here, we treated land-use type as an explanatory variable and site as a random effect. We tested if assumptions for normality and homogeneity of variances were met. If the residuals were homoscedastic, we used the *glht* function from the multcomp package[110] applying Tukey's all-pair comparisons with Bonferroni correction to assess differences in species richness between land-use types. If the residuals were heteroscedastic (withingroup deviation from uniformity significant with DHARMa quantile test[111]), we used the nonparametric Kruskal–Wallis test, followed by pairwise Wilcoxon test. For all our models we used simulation plots implemented in the DHARMa package to validate our model fit[111]. If our glmmTMB models were under- or over-dispersed, we changed the model family to compois (Conway–Maxwell Poisson distribution) and negative binominal (nbinom2), respectively. We assumed normal distribution for the models with mean normalized (endemic) richness as a response. To assess marginal and conditional $R^2$, we used the delta method using the *rsquaredGLMM* function of the package MuMIn[112].

We plotted all model results using RBase (Figs. 1, 2)[103]. The line inside the boxplot represents the *median* of each land-use type. We plotted scatterplots by using estimated regression lines from our fitted models and removed the vanilla agroforest type (forest- or fallow-derived vanilla agroforests) from the model if solely vanilla yield had a significant effect on species richness (Fig. 1 panels 5, 7). We extracted the model fits with the *allEffects* function from the effect package[113,114]. In all other cases (if no predictors were significant (e.g., Fig. 1 panel 3) or solely land-use history was significant as an additive or interactive effect (e.g., Fig. 1 panel 2) we retained the original model for plotting. In our graphs (Figs. 1 and 2), dashed horizontal lines are intercept-only linear models (lines are based on the *mean* of the distribution) and solid lines show statistically significant generalized linear regressions ($P < 0.05$). We show two colored lines as *dashed lines* if land-use history was significant as an additive term but no relationship of species richness with vanilla yield existed, or show them as *solid lines* if the effect of vanilla yield was moderated significantly by land-use history (Fig. 3).

To investigate differences in species composition among land-use types, we computed a permutational multivariate analysis of variance (PERMANOVA) using the *adonis* function of the *vegan* package[115]. Also, we used the *pairwise.adonis* function of the *pairwiseAdonis* package with false discovery rate correction to test for differences between land-use types[116]. Prior to PERMANOVA, we tested if homogenous dispersion existed among land-use types by using the *betadisp* function and *permutest* function of the *vegan* package (PERMDISP test)[115]. Our results show that heterogeneous dispersion significantly affected the differences in species composition for trees, birds, amphibians, butterflies, and ants (Table S14), thus differences are not only explained by the location of centroids but also by differences in dispersion between land-use types. We computed species composition by using non-metric multidimensional scaling (NMDS) with Jaccard dissimilarity distance. To compute the NMDS for amphibians, we excluded all plots where no amphibians occurred.

We generated sample-size-based rarefaction and extrapolation curves of species diversity of each taxon across land-use types to assess whether the curves reached an asymptote, indicating sampling completeness. To do so, we used *ggiNEXT* function of the iNext package[117] with the following settings: type = 1, datatype = incidence_raw, q = 0 (species richness), endpoint = 20. Furthermore, we computed the sampling coverage in the percentage of each taxon across land-use types by using the function *iNext* with the same settings. We calculated gamma species diversity for each land-use type by summing up all species found across 10 plots and therefore excluded 10 out of 20 fallow-derived vanilla agroforests randomly (Table S18).

**Reporting summary**. Further information on research design is available in the Nature Research Reporting Summary linked to this article.

## Data availability
The data generated in this study have been deposited in the Open Science Framework database under accession code https://osf.io/j54fx/?view_only=1bd699c5cda6402396 3e058254a33eec.

## Code availability
The code used for this study has been deposited in the Open Science Framework database under accession code https://osf.io/j54fx/?view_only=1bd699c5cda640239 63e058254a33eec.

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

## Acknowledgements

We thank all village leaders, village facilitators, and regional authorities who supported, communicated, and permitted our research. Special thanks go to all plot owners and their family members who allowed us to work on their fields and committed their time to facilitate our research. We thank Madagascar National Parks and the Ministry of Environment and Sustainable Development for granting us access to sites and allowing us export of samples. We express our appreciation to Rainer Marggraf and Andrea D. Bührmann as project leaders. Thanks go to Theudy Alexis, Patrice Antilahimena, Evrard Benasoavina, Claudine Bemamy, Jean Chrysostome Bevao, Ronik Botra, Joel Arnaud Harisaina, Dietrich Hertel, David Lees, Adriane März, Johannes Osewold, Thorien Rabemanantsoa, Dorah Ramaharobandro, Manoa Ramamonjisoa, Julien Randriampenomanana, Cédric Randrianantenaina, Eric Rakotomalala, Nicole Rasoamanana, Nantenaina Herizo Rakotomalala, Gatien Rasolofonirina, Joel Razafinantenaina, Dominik Schwab, Jean Claude Rakotonirina, Szabolcs Sáfián, Guillaume Velotody, Miguel Vences, and Maria S. Vorontsova who were indispensable to data collection, sample processing and sample identification. We collected data under research permits Nos. 100/17/MEEF/SG/DGF/DSAP/SCB.Re, 163/17/MEEF/SG/DGF/DSAP/SCB.Re, 18/18/MEEF/SG/DGF/DSAP/SCB.Re, and 254/18/MEEF/ SG/DGF/DSAP/SCB.Re granted by the Ministry for Water, Ecology and Forest (MEEF), Antananarivo. This study was financially supported by the Niedersächsisches Vorab of Volkswagen Foundation as part of the research project 'Diversity Turn in Land Use Science' (Grant number 11-76251-99-35/13 (ZN3119)) and *German Academic Exchange Service* (DAAD) within the 'Partnerships for Supporting Biodiversity in Developing Countries' initiative (Project No. 57449386). NG-R thanks the Dorothea Schlözer Postdoctoral Program of the Georg-August-Universität Goettingen for their support. Portions of this paper were developed from a dissertation chapter of Annemarie Wurz and Dominic A. Martin.

## Author contributions

A.W., T.T., I.G., D.A.M., K.O., E.R., F.A., T.R.F., A.A.N.A.R., M.R.S., A.A., J.B., H.H., D.H., H.K., B.R., H.L.T.R., F.M.R., L.H.R. conceived the idea; A.W., D.A.M. and K.O. developed the yield assessment protocol and collected the yield data; D.A.M. prepared yield data; A.W., D.A.M., K.O., E.R. T.R.F., M.R.S., A.A.N.A.R., R.R., R.A., and S.D. collected and processed species data; F.S.B. coordinated logistics; F.A., F.S.B., H.H. collected labour input data; A.W. analyzed and visualized the data; N.G.-R., A.A.N.A.R. and E.R. assisted analysis; A.W. led the writing of the manuscript. All authors contributed to the writing and gave final approval for publication. Authors are ordered (1) in alphabetic order for contributing Ph.Ds (FA-MRS) and (2) in alphabetic order for contributing project members, PostDocs and Professors (R.A. and L.H.R.).

## Funding

## Competing interests

The authors declare no competing interests.
