## [Peer Review File · Nature Communications]

Reviewers' Comments:

Reviewer #1:

Remarks to the Author:

When this invitation to review came in I was particularly overloaded (and with existing reviews on my desk). However when I read the abstract I had to accept the opportunity to review. There is rightly enormous interest in the question of whether increasing agricultural yields are always associated with biodiversity loss. That this paper claimed to demonstrate a win-win between biodiversity and vanilla productivity in such a biodiversity hotspot (and locally economically important agricultural commodity as vanilla) was fascinating.

The paper did not disappoint. The paper brings together a huge and very impressive data set which is the result of a major collaborative project involving multiple PhDs (looking at different taxonomic groups, at vanilla farming etc etc). I have seen many of the smaller underpinning papers from this study coming out over the last year or so but this brings everything together in a novel and synthetic analysis.

The paper is really well written and framed in the wider literature. The results are very clearly presented and interesting. Their central result is extremely important and has wide reaching consequences and I expect will stimulate further research.

I have a slight doubt about what their data really means and how applicable the findings are beyond the specific set of plots they looked at. The suggestions I make will improve the ability of the reader to judge its wider applicability but despite questions I have about whether the results would hold over a different set of vanilla agro-forestry, I do not doubt it should be published (as long as the full data is archived alongside). Such data is really hard to collect and it is not surprising they could not look at a wider range of plots. As long as the data is publicly archived so this can be combined with future studies and really move the field forward, and my concerns about the weak reporting of the social research methods are addressed, this paper is worth publishing in Nature Communications and I am confident it will be widely cited and used.

Below I suggest a number of mostly pretty small issues which could be easily addressed to improve the clarity of the paper and/or make the methods more transparent.

My most important question/clarification I would like is on the data availability. I did have a quick look at what was available in the OSF link and this seems to mostly be metadata and summary data tables. I could not see the raw data eg on what the species (of tree/amphibian etc) which were found at each plot. I feel this is essential and without this data being archived the paper is substantially less valuable.

My other most significant concern is the way in which social data was collected and how it is reported. The whole results hinge on two bits of data which were collected with social research methods (land use history classification of the agroforestry plots, and the labour inputs to the farming system). Both types of data are difficult to collect (I have done both in very similar settings in rural Madagascar. In total, they devote a single paragraph to social research methods in the main text and nothing further in the SI (that I saw). This is concerning and needs addressing. Best practice suggests all survey instruments should be archived alongside the paper ideally in original language and English eg in the SI and we certainly need an ethics statement.

Fig 1 and 2 are excellent and extremely informative and contain a very impressive richness of data. I do wonder if the order of the colours/habitat types presented in the legend would be better aligned with the order they appear in the plot? I also felt it was slightly jarring using A, B and C etc and a, b, c in the same plot. I also felt the blobs needed to be slightly larger to allow the reader to see the colours (blobs are colour coded by land use category right?)

In the heading for Table 1 you say that mean values from all 30 agroforests was used to input data for missing values of pollination labour input. This doesn't seem ideal. At least I'd like to see a sensitivity test of excluding those three sites from this important analysis.

While the methods (especially when read in association with the SI) were mostly clear, I did have a few queries which I feel could have been addressed in the methods.

1) You refer to villages a lot in your study design. These 'villages' obviously have spatial boundaries associated as a criteria for selection was having <40% water. I wonder if you were using fokontany boundaries. Of course I can see why you would want to minimize use of that word not familiar to many readers but fokontany doesn't really translate directly as village and if you were using fokontany boundaries it would be useful to say this (you could just say 'lowest administrative unit in Madagascar, hereafter referred to as village' or something like that.

2) In the section on study design you talk about selecting 3 vanilla agroforests and 2 fallows from

each village. Presumably you don't talk about representing the two categories of vanilla agroforestry at this point as this was defined later. This is OK but I would like to know somewhere how the two categories were represented across the 10 sites (maybe this comes in plot selection). I think the map in Fig S1 is also critical to this. I was keen to know if, for example, all your Vfor were close to blocks of natural forest and all your VFal were far from forest blocks. I do think this is important information.

3) Were the vanilla yield measurements made on 36 vanilla pieds per plot? I think so from reading plot design but it isn't explicit in 'vanilla yield' section if that is 36 pieds total or per plot.

4) I am not expert on bird counts but I was surprised only 2 repeat counts were done for each of the 30 plots. There is also no mention of the time of day or weather conditions when bird point counts were done (I'd expect them to need to be restricted to certain conditions especially with such few repeats).

5) While the Si is helpful, I did feel it could contain a bit more information and/or better links to. For example the main text mentions taxonomic voucher specimens etc but I am not sure from what is presented that these could be found. On that note, is the raw data available alongside the paper? This would be extremely valuable (and is certainly best practice). Please make the data and code available if at all possible.

6) The data relies on farmers to provide quite a bit of information eg on the land use history, age of plants and of course also the study of pollination labour input. There is no information provided about ethical review of this social research or basic information on who did the interviews, compensation etc etc. I also really feel, given the importance of the pollination labour in the model and the overall results, I'd like a bit more exploration of this data to help convince the reader that this is a valid measure of true labour input. Self reported data on agricultural inputs can be difficult (though they had a nice short recall period which is reassuring). If you have data it might be nice to look to see if pollination labor input is predicted by available labour in the HH (only because if it is, it would give a reason why it varies so much, beyond simply being nose derived from the method). The land use history classification is critical and again needs more exploration.

Very minor points:

I wonder if the abstract could be slightly reordered to find a way to mention earlier on which (very impressive) range of taxa the paper covers. The 1st few sentences talk only about 'biodiversity'. Line 125 is slightly awkward-are you saying there has been decades of research about land use history, or agro-forestry generally (NB this is a TINY point). Line 628 could do with a reference.

Reviewer #2:

Remarks to the Author:

(Please note, I attached a PDF of all these comments for ease of reading.)

Thank you for a very interesting manuscript which presents some very robustly-collected evidence for yield-biodiversity trade-offs in vanilla agroforestry systems in Madagascar, which is an important topic in the context of global change, sustainable production, sustainable development in one of the poorest parts of the world, and wildlife conservation. It is also relevant to current global initiatives around forest and landscape restoration, although the framing of the paper around this topic needs some improvement. I enjoyed reading it, and the results are nicely presented in Figure 1 and 2.

However, I have some concerns about how the results are described in text (seeming to gloss over results that don't fit the narrative), a lack of clarity in presenting the relationships between management variables and yields/biodiversity, and the use of a 'multidiversity' measure that is hard to understand and which I don't think is informative. I think attention should also be paid to species composition and to total species richness accumulated over all plots – not just plot-level species diversity, as well as some more detail around the percentage endemism of the various taxa (differs greatly, I think, which is of importance for interpreting the results). I also think the language around restoration and degradation needs some careful consideration.

- What are the noteworthy results?

Vanilla production can produce high yields while supporting high levels of biodiversity in agroforestry systems, and offer a sustainable production system and a restoration option (for carbon and biodiversity) relative to shifting cultivation of rice in the studied landscape. They cannot replace old-growth contiguous forests in terms of biodiversity value, but represent a better alternative than land clearance. Meanwhile, there seems to be considerable room to improve vanilla yields through management without impacting biodiversity value, although currently this isn't shown as clearly as it could be.

- Will the work be of significance to the field and related fields? How does it compare to the established literature? If the work is not original, please provide relevant references.

Yes, the work is original and contributes an extension and expansion of existing work on yield-biodiversity trade-offs in tropical countries by assessing a different crop, accounting for land use history, and by using rigorous methods to measure yields and biodiversity across multiple taxa. It contributes to current new debates on forest and landscape restoration options, as well as pre-existing on-going debates around sustainable development and sustainable agriculture.

- Does the work support the conclusions and claims, or is additional evidence needed?

Some additional analysis and clearer presentation of evidence is needed, as outlined in detailed comments to authors.

- Are there any flaws in the data analysis, interpretation and conclusions? - Do these prohibit publication or require revision?

I strongly recommend revisions to the analysis and interpretation as outlined in line-by-line comments to authors. I don't anticipate the key findings will necessarily change, but they should be more clearly presented, and I have suggested they conduct additional analysis on species composition which will then produce new and additional conclusions.

- Is the methodology sound? Does the work meet the expected standards in your field?

Yes the data collection methodology is very sound, and the statistical analysis applied is broadly fine, but I have made some suggestions around additional analyses that would add depth, and also asked for better presentation of the relationships, as I don't think one of the sets of models (yield-biodiversity direct relationship) is logical.

- Is there enough detail provided in the methods for the work to be reproduced?

Yes, but a few additional details requested below to make this completely reproducible – namely more information about the land use types studied.

Abstract

102: fewer, not less i.e. "fewer species" not "less species"

102-103: please reconsider the use of brackets for the endemic species vs all species reporting, as this is hard to understand. Did you compare fallow-origin agroforests with old-growth forest?

Worth stating that old-growth forest is best for biodiversity in the abstract.

105: do longer vines mean older vines? So long-established vanilla vines yield more? Or is length independent of age and related to something else?

106: is "landscape forest cover" here old-growth forest, degraded forest, old fallows, or forest-derived-agroforests? Could you tell from the forest cover dataset? Are the forest fragments zero-yield, or are they productive somehow? I think that you only surveyed biodiversity in the old-growth forest park, but not in forest remaining in the landscape so would be helpful to make a distinction as this isn't clear from the abstract.

108: biodiversity is unrelated to vanilla yields within agroforests, but not overall (old growth forest is best) so this should be rephrased to make that clear

110: please be careful with the use of the terms 'degraded' and 'restoring' here – I'm sure you're aware of the diversity of definitions for these terms, so this should be better placed in its specific context i.e. development of vanilla agroforestry on non-productive and low-biodiversity-value scrub habitats formed by shifting agriculture (fallows) would be better for biodiversity (and livelihoods?) than... which alternatives? Re-clearance of fallows for rice or other crops? Allowing fallows to regenerate back to forest (does this happen, or are these areas degraded in the sense that they cannot recover back to the original ecosystem if left alone?)

Introduction

117: I think agroforestry is also widely seen as unprofitable and 'backwards' relative to intensified agriculture by e.g. government agencies, agricultural development agencies. So either rephrase to omit the idea that this is a commonly-held belief, or perhaps better, explore the idea that agroforestry is a 'niche' idea in many ways, and needs further evidence to convince those involved in agricultural policies that it is a good idea?

119: Please be a bit clearer about the definition of 'degraded land' here, and of 'restoration'. Are you terming agricultural land that is productive as 'degraded' (reasonable from a biodiversity point of view, but unreasonable from a productivity point of view), or are you terming the fallows within shifting cultivation as 'degraded' because they are non-productive whilst they are fallow, but also worse for biodiversity than undisturbed ecosystems (although better than agriculture?) – in which case there is a difficulty when considering fallowing as an essential part of the sustainability of agriculture on tropical soils. Or, are you using degraded to define only areas that have been through the fallow/clearance process so many times that the soils are so degraded that they can neither regenerate to forest, nor support agriculture? Which I think would be unequivocal. Or, are you even able to explore perceptions of what degraded land means for people living in those landscapes? For 'restoration' – the typical SER definition involves moving a system back to a reference ecosystem, but that's not what we are talking about here, so please provide a brief definition.

124: The effect of agroforestry on biodiversity at a landscape level may also be linked to their yields – if an agroforestry system yields less than a monocultural system, this may increase demand for land and drive deforestation (for example if farmers encouraged to adopt agroforestry to support biodiversity find their income declining as a result) - or indirectly cause deforestation by displacing forest clearance elsewhere (if agroforestry is done by different people)? You could briefly include some references here around these points for a broader context e.g. Kremen, Claire. "Reframing the Land-Sparing/Land-Sharing Debate for Biodiversity Conservation." *Annals of the New York Academy of Sciences*, 2015, 1–25. <https://doi.org/10.1111/nyas.12845>. Please also check again the use of the term 'degraded' here – you seem to be using it as a synonym for cultivated land here. Perhaps once you have defined it earlier in this paragraph this can be clearer.

135: are forest-derived agroforests established within the protected forest area (the old growth forest used as a reference in the study) or within forest fragments in the landscape? If the latter, it seems you did not conduct any surveys within the forest fragments – do they differ much from the old growth continuous forest, and would this affect your conclusions around the biodiversity value of forest-derived agroforests in relation to forest?

136: what does fallow land look like? Is it shrub cover, recovering forest, or grassland? Would forest recover if fallows were left long enough, or is the soil too degraded? Do fallows have a social purpose?

141: the word "urging" does not quite make grammatical sense here – suggest "generating a need" or similar instead.

143: need comma after "30 vanilla agroforests"

Results

153: why have you put brackets around "endemic"? I think it would be helpful in the introduction to explain that you assess all species, and endemic species separately, and why.

I think it would also be helpful to explain what 'multidiversity' means here given the structure of the paper with the Methods at the end. Is this just species richness of all taxa combined? There is a threshold mentioned in the caption of Figure 1 but this is not explained. The reader should be able to interpret the figure in its entirety with the caption without referring to the methods, so please explain it here.

Note – reference nos. 72 and 73 (in Methods) do not use the term 'multidiversity' at all, but refer to metrics of ecosystem function based on e.g. soil microbial diversity, where a threshold is applied to ecosystem functioning, not to the level of biodiversity used to explain ecosystem functioning? This does not seem relevant to the message of this paper, which does not state that it aims to explore ecosystem functioning. Instead, 'multidiversity' appears as a synthetic metric for diversity of the other taxa, but it is not very clear why this is informative relative to the taxon

specific species richness measures. Perhaps this just needs better justification?

155: You state "did not find" but then "few exceptions" – so in fact, you did find some relationships, so this is a little disingenuous. Please rephrase. I think it would be helpful here to mention the effect sizes – the numbers of species per plot is rather small, and so is the effect size where there are statistically significant relationships. Please also report the key results of the statistical tests in-line, or on Figure 1 – the reader should not have to look in the supplementary information to find out the the results of the statistical test and effect sizes.

160: Again, please increase the clarity of writing around endemic vs all-species patterns e.g. change "multidiversity and species richness did not differ" could be rephrased as "For all measured species, multidiversity and species richness only differed for...." Because again, you state "did not differ" and then go on to say there were differences after all.

Can you also provide a little detail on the % of all species that are endemic? I believe endemism of some taxa in Madagascar is extremely high, making those groups of potentially increased conservation concern.

169: the phrasing reads "biodiversity changes...was influenced by... taxa" which doesn't quite make sense – I think you mean, responses differed by taxon?

169 – 171: you start with "on the one hand" but then use "whereas" instead of completing with "on the other hand" – you could revise.

171: at this point, and when looking at Figure 1, I really want to know about species composition, not just richness, especially when comparing forest to vanilla agroforests. Species richness per plot can mask huge changes in species composition, with implications for conservation concern and ecosystem functioning. I don't expect the same 10 reptile species per forest plot are found in herbaceous fallows. An NMDS analysis clustering species by system (forest, vfor, vfall, and fall) could be informative and be placed in the SI with a brief description of results in main text if you are struggling for space. Similarly I think cumulative species richness across each system (i.e. rarefaction curves) would be helpful to catch species accumulation in each system. I would like to see how the rarefaction curve flattens (or not) for each sampled system to assess sampling completeness, and also to get an idea of how each system contributes to biodiversity more widely i.e. are the same species in fallows found again and again as they are more homogenous than the agroforests?

170 onwards: what are these % changes reporting? Differences in median values? Great to have effect sizes, but also please incorporate them in the section on yields vs biodiversity (as mentioned above)

179: please finish the sentence by reminding the reader which comparison this refers to e.g. vanilla vs fallows.

Figure 1: Looking at the per-plot species richness patterns, I am interested to know more about two things – cumulative species richness (i.e. rarefaction of species richness with accumulated sampling) between the four measured systems), and species composition. To me, these would be more informative and interesting than the multidiversity measure, which I find difficult to interpret and understand in relation to application or policy based on these findings. I have comments about the multidiversity measure elsewhere in this review, but if you are keeping it in, I think you need to explain what it means in the figure legend, so the figure can be interpreted standalone. Please explain the boxes and whiskers as well as the median line for boxplots. Please provide stats results on the figure, if not in-line in the main text referring to this figure.

Figure 2: if you need to save words and the journal allows, you could avoid duplicating the explanation of the colours/lines/boxplots and just say "as for Figure 1"

180: please provide figures of yield relationships in the main text, not just tables (nobody likes tables really, they slow down interpretation). Does vanilla vine length correspond to age, or is it something else i.e. faster growing vines due to being better fertilised, or different planting stock? Please explain this further.

181: "were" should be replaced "was" in relation to "yields"

183: Can Table 1 include effect sizes if you are keeping it in (i.e. if not replacing it with a figure with the statistical results reported on it). Re: hand pollination, why don't people pollinate more? Is it labour constraints?

191: Again, a problem with the phrasing around direction of results here – "in general...unrelated" but then when you look at Table 2, you find that there do in fact appear to be trade-offs for multiple variables (vine length, vine density, pollination). Indeed, there seem to be some key trade-offs that need further interrogation, namely why increased vanilla plant density was related to reduced tree species, and why longer vines were associated with trees and reptiles (perhaps for

the discussion, perhaps you have good reason to believe these relationships are non-causal, but needs not to be glossed over) I think here you need to link back to Figures 1 and 2 to explain that these variables may be behind the relationships between yield and biodiversity.

Indeed, I am actually unsure whether yield and biodiversity should be modelled directly in relation to each other in a GLMM as you have done, at all. Biodiversity does not respond directly to yields – as you explore here, it responds to management, as does yields, so implying that biodiversity responds to yields or vice-versa doesn't quite make sense. Although I am reluctant to posit alternative analytical approaches unnecessarily, in this case it seems it would be more logical to link together the yield-management-biodiversity relationships together more clearly in your main figures, not just the yield-biodiversity relationships. Whether you would be best doing this using a combined model of some kind (e.g. using a path analysis type approach) or whether you are able to display the results of the yield and biodiversity responses to each of the management variables on the same figure, but modelled independently, I am not sure.

192: this is the wrong supp table – should be Tables S7 and S8 I think. Please provide key stats results in the main text or a figure, and don't force the reader to the supp material.

Table 1: why were the datapoints with missing data for a key variables (pollination labour input) imputed and not omitted? Do results differ if you omit these plots? Table S11 shows the pollination hours have a very wide range, spanning one order of magnitude, as do vanilla yields. Out of interest, what happened in the plot with 0 vanilla yields? Was there more than one with total failure of yields?

I don't understand why the p-values in Table 1 are different to Table S6 where you report the full results of the statistical test. Indeed, the full model results, estimate and SE should all be reported in the main text. Are these results from the full model (inc all variables as in Table S5) or the best reduced model (Table S6)? The p-values don't match up to either. I actually don't understand Table S5 at all – what do the χ^2 values refer to?

Table S2: Should reference Supp Table S7 and S8 not S5 and S6?

Discussion

266: your plots seem to yield about 1/3rd of the 'productivity benchmark' mentioned here – why?

269: please explain further the role that fallow land plays in these landscapes – do they have a social or other agricultural function? Are there both costs and benefits of turning fallows into vanilla agroforests, are they done by different people? Do fallows have cultural value?

275: I don't think that this assertion that management variables affecting biodiversity are different to those affecting yield are evidenced strongly in the results, as currently presented. This may be because a different type of analysis is needed, or the same analysis but presented differently so that in one figure a reader can readily see that yields and biodiversity have opposing responses to the same variable. As the results are currently presented, it seems the text emphasises the lack of relationships between yield and biodiversity, while the figure and tables reveal there are some apparent relationships which then need to be explained away.

309: But what about species composition? And accumulated species richness?

341: "complementary" (not complimentary, which is to pay a compliment i.e. say something nice to someone")

Methods

553: can you give percentage endemism here, or else, report in the results (already mentioned above)

554: please provide more information here about the hill rice system – what are the fallows like, why is this system used, and are the people growing rice the same as those growing vanilla?

564: why did you exclude villages with coconut?

570: why did you not conduct surveys in forest fragments? My concern is whether forest fragments contain the same species as continuous forest, which are then lost on conversion to agroforest, potentially losing tiny scattered and vital populations important for connectivity etc. I would like to see a little discussion around this in the main text, as well as explaining why they weren't surveyed directly here in the methods.

577: this answers my earlier question about how and where forest-derived vanilla systems are established.

579-580: please can you provide more information about what the fallows are like? Some photos

in the supplementary information would be very informative indeed.

588: so there is succession on the fallows if they are left – so does vanilla establishment promote or facilitate recovery to a forest-like state, or are the agroforests just older than the non-agroforest fallows because they get cleared more frequently, and therefore there is no direct facilitation effect? But an indirect facilitation effect (i.e. by providing a long-term land use not in conflict with forest cover, trees are allowed to grow naturally)? This is important for the discussion of restoration – I suggest adding something on this to the discussion.

596: so forest likely disturbed? How similar to the remaining forest fragments in the wider landscape in terms of structure?

602: please give sample plot area in ha to match the following sentence

622: “across the” not “in each of” – you didn’t do $30 \times 10 = 300$ plots!

661: were any questions asked about other labour activities? Why did pollination vary?

666: as mentioned above, why not omit these plots? Please justify, perhaps with model results with and without the imputed values.

673: so vanilla support trees are quite short – are there trees above them in either agroforestry system? Photos may again help here.

682: does this forest cover map distinguish forest from agroforest? Do you think that would matter?

686: rephrase to “percentage woody cover”

691: why did you measure leaf damage? No hypothesis around leaf damage is outlined in this paper.

740: so there are no trees in herbaceous fallows? Photos please!

741: so $n = 48$ plots in total were assessed including the forest sites, with $n = 30$ agroforests? or $n = 60$ but only $n = 48$ with tree assessments? Were two fallow-derived plots excluded from all analyses or just the trees?

744: a local who? A local farmer, assistant, person, guide, botanist?

758: is there potential interannual seasonal effects not detected in the forest? Species accumulation/rarefaction curves could be illuminating here

760: what time of day were the point counts conducted – in the morning? What % of birds are endemic?

768: i.e. moving logs, leaf litter?

769: how were DNA samples taken? How did you identify species – are there field guides (if so please cite) or other online resources?

776: more info on trap design please – and interesting to know what % of butterfly captures were from nets vs manual netting please. Any problems with rain, or is there no rain in the sampled season? Any traps lost to wildlife or people?

787: please give full location name for butterfly samples

798: who did the ID, and how? Please give field guide references/online info.

800: address/city?

803: please see my earlier notes on the multidiversity idea, which could be addressed here. I find the explanation of the approach quite unclear. What are rigorosity and mildness and why do they matter? From looking briefly at the cited papers, they discuss a measure of ecosystem functioning, which is not what you are trying to capture here. I do not find that the multidiversity measure provides any additional insight or information useable to the reader, and would prefer more analysis that looks at species composition, as mentioned earlier in my review, as this links more closely to thinking about conservation. Species richness hides a multitude of change. I don’t see how the multidiversity measure concept can transfer to management, policy or conservation measures, nor how it informs e.g. ecosystem functioning or services as the cited papers do.

814: why not use total richness, why penalise and reduce the impact of species losses on your metric?

818: I don’t understand what this figure helps you understand about the biodiversity response in the context of conservation, or management.

820: please explain what glmmTMB models do differently compared to linear mixed effect models, and why you used them here.

836: please explain this in the caption of Table S5. So you have sequentially reduced the model by one variable each time, or taken out each variable in turn to test change in variance? Are these chi-square tests comparing the model with only this variable, or with the full model minus only this variable?

839: when did you use which model type and why?

844: please give full name for "compois"

849: i.e. prediction? If so, you can provide effect sizes (change in y per unit change in x) for significant results, on the relevant figure in the main text.

851: you have not actually explained the fitting of this mode – only the fitting if yield or species richness response to predictors. As mentioned in my comments on the results section, directly assessing yields vs richness doesn't quite make sense to me.

I am surprised no attempt made to assess species composition using NMDS or RDA.

Supp info

Table S2: no need to highlight the significance of the intercept

Reset page numbers to run through whole SI if using (i.e. across the section breaks) and perhaps provide a table of contents at the start?

Fig S3: VFAL not VFST?

Fig S4: there are no circles – check caption

Little correlation between age and vine length here, so what affects vine length?

Fig S5: "only marginally associated" here really means "a significant effect with small effect size" – move this to main text, or something similar, as per comments on results, above.

Fig S7: not a major concern as soil doesn't really get discussed, but why isn't soil C included in the PCA?

Table S13: VFST should be VFOR?

Table S14: VFAL not VFLW?

Reviewer #3:

Remarks to the Author:

I have been invited to review the manuscript "", specifically to determine if the biodiversity surveys, especially birds, have been conducted appropriately, and invited to take a broader look at the paper. Given this paper is far from my expertise, I will only comment on the biodiversity surveys.

Overall, I think it looks fine but I have several comments requesting a bit more information. First, I think the authors could add tables in SI with species sampled and if they are endemic or not (currently authors cite external source, which may disappear / evolve at some point) and it would be good to have some stand-alone information for reproducibility and transparency. Second, there is often some delay between first and last sampling (eg August to December) and it is unclear whether this can have a strong impact. I suggest the authors state a bit better how this way conducted (especially it's important to know if the order was random, treatment by treatment, or village by village). For instance, if all sampling in the National Park has been done in December and if there is a strong seasonal effect, this can have important effect on results. A possible way to address that is to analyse the effect of seasonality on biodiversity metrics used in the study (and possibly to control for this in the analyses).

Trees: The sampling seems appropriate to me. I would just ask the authors to add in Supplementary Information a list of species, specifying which species are considered as endemic so that readers can find all information (and will be able to access it in the future as well)

Herbaceous plants: same comment. Can you also detail how plant phenology can affect the sampling in the 4 months study period and if how was conducted the sampling (village after village / random / first all forests and then all fallows...)

Birds: I think the sampling duration and number of observers are appropriate but that some information is lacking. Specifically, while this is less of an issue with plants, the expertise of observers is key for birds. So I suggest the authors add some information on the people involved in sampling (are they all birders, how well do they know birds from the region and especially their calls and songs, how many different people were involved, can there be a bias if some people went only in some villages and other only in other villages...). I am also concerned with the time window

(from August to December) which can affect birds detected. I suggest the authors add in SI some analyses looking at the variation in the bird metrics used (eg richness) over time regardless of treatment. If there is a strong pattern, it should be accounted for in the analyses. Similarly to my questions on plants, how was this timing distributed across sampling units (eg villages one by one / random / treatment 1 then treatment 2...)?

Reptiles and amphibians: Same comments, the sampling method seems appropriate but I would appreciate some information on the randomisation of sampling order and the expertise of fieldworkers.

Butterflies: Perfect

Ants: I don't know anything about ants but it sounds fine

Editor's comment Dr. Pilar Morera Margarit,

Thank you again for submitting your manuscript "High crop yields without biodiversity loss in tropical agroforestry" to Nature Communications. We have now received reports from 3 reviewers and, on the basis of their comments, we have decided to invite a revision of your work for further consideration in our journal. Your revision should address all the points raised by our reviewers (see their reports below).

I would like to emphasise the importance of making data available as well as improving the transparency of the methods by reporting more clearly on social data collection (including ethics statement) and the stats and model results. I remind you that the Methods section is not taken into account for the word count so you can extend this section as much as you want to improve the clarity of the methodology used.

Reviewers overall agreed on the need of having more information on the biodiversity surveys so it is important to return a revised version of the manuscript with additional details on the surveys including species taken into consideration and if these are endemic.

It is also necessary to tone down potential overstatements on agroforestry as none of us would like these results to be misinterpreted and used as an excuse to transform old forests into agroforestry.

When resubmitting, you must provide a point-by-point verbatim response to the reviewers' comments. Please show all changes in the manuscript text file with track changes or colour highlighting. If you are unable to address specific reviewer requests or find any points invalid, please explain why in the point-by-point response.

I. RESPONSE: We thank the editor for these important comments. We now provide species-by-site matrix on OSF (Open Science Framework), included more details about the collection of the social data (including details on the rules of good scientific practice we followed) and reported model results also in the main text. We extended the method section for all taxa to include details on repetitions, observer identity and how seasonal biases were minimized. We also provide percentages for the endemism level of each taxon. We made sure to present each result transparently and discuss the limitations of our study in the discussion to avoid a potential overstatement on agroforestry. Please see our detailed replies to these and other concerns of the reviews below.

Data available in Open Science Framework:

https://osf.io/j54fx/?view_only=1bd699c5cda64023963e058254a33eec

Reviewers' comments

Reviewer #1 (Remarks to the Author):

When this invitation to review came in I was particularly overloaded (and with existing

reviews on my desk). However, when I read the abstract I had to accept the opportunity to review. There is rightly enormous interest in the question of whether increasing agricultural yields are always associated with biodiversity loss. That this paper claimed to demonstrate a win-win between biodiversity and vanilla productivity in such a biodiversity hotspot (and locally economically important agricultural commodity as vanilla) was fascinating.

The paper did not disappoint. The paper brings together a huge and very impressive data set which is the result of a major collaborative project involving multiple PhDs (looking at different taxonomic groups, at vanilla farming etc etc). I have seen many of the smaller underpinning papers from this study coming out over the last year or so but this brings everything together in a novel and synthetic analysis.

The paper is really well written and framed in the wider literature. The results are very clearly presented and interesting. Their central result is extremely important and has wide reaching consequences and I expect will simulate further research.

2. RESPONSE: Thank you very much for the positive feedback. We are very happy to hear that our collaborative project is well known and that our research can spark new discussions and contribute to a deeper understanding.

I have a slight doubt about what their data really means and how applicable the findings are beyond the specific set of plots they looked at. The suggestions I make will improve the ability of the reader to judge its wider applicability but despite questions I have about whether the results would hold over a different set of vanilla agro-forestry, I do not doubt it should be published (as long as the full data is archived alongside).

3. RESPONSE: We agree with the reviewer that yield-biodiversity relationships may change in more intensively managed vanilla-farming systems (e.g., on La Reunion). In our paper, we emphasize that our results refer to a specific range of management practices, e.g., in line 343: “In contrast to common expectations of ecological-economic trade-offs^{3,35}, we show that increasing yields within the current range of vanilla agroforestry management practices (productivity benchmark [reference point for an average maximum of yield] in Madagascar: 350 kg/ha³⁶) are not generally associated with biodiversity losses.”. Therefore, while we consider our study is applicable to systems with a similar range of management intensities, we are aware that this may not hold to all kinds of vanilla systems across the world (e.g., traditional systems (Acahual) of vanilla in Mexico; Díaz-Bautista et al. (2019)). This has been clarified in our discussion (line 416). Nevertheless, Madagascar is the most important vanilla-growing region globally, producing roughly half of the world’s vanilla (FAO, 2020). Thus, the systems investigated in our study do not represent niche systems, but a common and environmentally and socially important type of vanilla cultivation.

For reproducibility, we now provide all data (species by site matrix and yield, environmental, and management data) online on OSF (Open Science Framework).

https://osf.io/j54fx/?view_only=1bd699c5cda64023963e058254a33eec

Uploaded files include:

- Species by site matrix (Presence/Absence) for 7 taxa
- Compiled data set for analysis for 5 land-use types
- Compiled data set for analysis for vanilla agroforests
- Pictograms and questionnaires for assessment of pollination labour input

- Data of work input invested in each agroforest with household members
- Questionnaire and farmer hand out for yield assessment
- R script

Such data is really hard to collect and it is not surprising they could not look at a wider range of plots. As long as the data is publicly archived so this can be combined with future studies and really move the field forward, and my concerns about the weak reporting of the social research methods are addressed, this paper is worth publishing in Nature Communications and I am confident it will be widely cited and used.

4. RESPONSE: Thanks for your remarks! Below (Response 6) we reply in detail to your concern about the social research methods. In addition, we uploaded species-by-site data on OSF (Open Science Framework).

Below I suggest a number of mostly pretty small issues which could be easily addressed to improve the clarity of the paper and/or make the methods more transparent.

My most important question/clarification I would like is on the data availability. I did have a quick look at what was available in the OSF link and this seems to mostly be metadata and summary data tables. I could not see the raw data eg on what the species (of tree/amphibian etc) which were found at each plot. I feel this is essential and without this data being archived the paper is substantially less valuable.

5. RESPONSE: We agree and updated the raw data on Open Science Framework. It now also includes a species by site matrix (presence/absence) for each taxon.

My other most significant concern is the way in which social data was collected and how it is reported. The whole results hinge on two bits of data which were collected with social research methods (land use history classification of the agroforestry plots, and the labour inputs to the farming system). Both types of data are difficult to collect (I have done both in very similar settings in rural Madagascar. In total, they devote a single paragraph to social research methods in the main text and nothing further in the SI (that I saw). This is concerning and needs addressing. Best practice suggests all survey instruments should be archived alongside the paper ideally in original language and English eg in the SI and we certainly need an ethics statement.

6. RESPONSE: We now uploaded the questionnaire in the original language on OSF. It includes the questionnaires for the farmers, the data input sheets for the team members collecting the data, and the pictograms used.

We also added more details in the method section on interviewers, data collection, and compensation (see method section on “pollination labour input” in line 883). Furthermore, we added information about which guidelines of “Good Scientific Practice” were followed during the interviews. Guidelines of “Good Scientific Practice” by the University of Göttingen were adopted for all interviews (adapted based on the recommendations of the codex for good scientific practice from the DFG, German Research Foundation; <https://www.uni-goettingen.de/en/good+scientific+practice/567647.html>). In addition, we give information on the advisory board of the project which advised the project throughout the research process, including on potential ethical considerations (see method section on “pollination labour input” in line 883).

Below (see Response 14) we provide argumentation supporting our land-use history classification (soil analysis and tree species composition analysis) and the quality of the pollination labour input data (correlation of pollination labour input with available household labour and details on data collection).

Fig 1 and 2 are excellent and extremely informative and contain a very impressive richness of data. I do wonder if the order of the colours/habitat types presented in the legend would be better aligned with the order they appear in the plot? I also felt it was slightly jarring using A, B and C etc and a, b, c in the same plot. I also felt the blobs needed to be slightly larger to allow the reader to see the colours (blobs are colour coded by land use category right?)

7. RESPONSE: Thanks for these valuable suggestions. We kept A, B, C... because we saw this is common in Nature Communication publications. We increased the dot sizes in the graph. Furthermore, we adapted the order of land-use types presented in the legend. Yes, the colours of the dots in the yield-biodiversity graph are colour-coded by land-use history. We added now colour explanations for each land-use type below the graph.

In the heading for Table 1 you say that mean values from all 30 agroforests was used to input data for missing values of pollination labour input. This doesn't seem ideal. At least I'd like to see a sensitivity test of excluding those three sites from this important analysis.

8. RESPONSE: Thanks for this idea! We used averaged values of pollination labour input for the three missing plots as omitting these plots would have meant losing richness and yield data for these three plots as well when including these in joint models. Nevertheless, we acknowledge that the averaging is not ideal and thus adapted our approach. We performed a linear regression of available household labour with pollination labour /ha and found a good correlation between the two (p-value:0.005, estimate=52.82, $R^2= 0.270$). Based on the number of household members (for which we have available data for the three missing plots), we then predicted expected pollination labour input for the three missing plots, using the estimated relationship of the two from the mentioned regression. We performed a sensitivity test including and excluding the three plots. Comparing the model results of the yield predictors using three different approaches (including averaged values, including fitted values, excluding missing plots) shows that pollination labour input has a marginally significant or no significant effect. Thus, the choice of method does not affect our final research conclusions.

Model results of linear model predicting pollination labour input by number of household members:

Predictor	Estimate	SE	p-value
Intercept	12.15	102.87	0.90690
Number of household members	52.82	17.38	0.00549

Figure showing the fitted line of the relationship of pollination labour input with number of household members.

Model results of yield determinants **including predicted values** for pollination labour input based on the model fit of the linear regression of pollination labour input with number of household members.

Parameter	Estimate	SE	p-value
Intercept	9.005	0.804	<0.001
Vanilla planting density (no/ha)	2.901	0.415	<0.001
Pollination labour input (hrs/ha)	0.897	0.469	0.069
Vanilla vine length (cm)	2.650	0.393	<0.001

Model results of yield determinants **including averaged values** for the three plots with missing pollination labour input:

Predictor	Estimate	SE	p-value
Intercept	9.005	0.803	<0.001
Vanilla planting density (no/ha)	2.878	0.415	<0.001
Pollination labour input (hrs/ha)	0.943	0.473	0.058
Vanilla vine length (cm)	2.630	0.391	<0.001

Model results of yield determinants **excluding values** of the three plots with missing pollination labour input:

Predictor	Estimate	SE	p-value
Intercept	9.362	0.748	<0.001
Vanilla planting density (no/ha)	2.604	0.465	<0.001
Pollination labour input (hrs/ha)	0.613	0.535	0.263
Vanilla vine length (cm)	2.719	0.538	<0.001

While the methods (especially when read in association with the SI) were mostly clear, I did have a few queries which I feel could have been addressed in the methods.

1) You refer to villages a lot in your study design. These ‘villages’ obviously have spatial boundaries associated as a criteria for selection was having <40% water. I wonder if you were using fokontany boundaries. Of course, I can see why you would want to minimize use of that word not familiar to many readers but fokontany doesn’t really translate directly as village and if you were using fokontany boundaries it would be useful to say this (you could just say ‘lowest administrative unit in Madagascar, hereafter referred to as village’ or something like that.

9. RESPONSE: Thanks for your comment. To our knowledge, ‘fokontany’ can also refer to more than one village and is composed of different sectors (“secteur”). Thus, we refrain from calling it ‘fokontany’ or lowest administrative unit as this does not in all cases conform with all studied villages. We selected the villages based on the list of villages for our study region from official election lists which listed all villages within a fokontany individually (Hänke et al., 2018). Village boundaries, demographics, infrastructure were defined based on a rapid survey with the chef du village. We selected only villages with <40% water because we wanted to avoid a strong influence of water elements (sea, rivers, lakes) on the landscape as our study was focused on agriculture. We added these explanations to the method section, part “study design”.

2) In the section on study design you talk about selecting 3 vanilla agroforests and 2 fallows from each village. Presumably you don’t talk about representing the two categories of vanilla agroforestry at this point as this was defined later. This is OK but I would like to know somewhere how the two categories were represented across the 10 sites (maybe this comes in plot selection). I think the map in Fig S1 is also critical to this. I was keen to know if, for example, all your Vfor were close to blocks of natural forest and all your VFal were far from forest blocks. I do think this is important information.

10. RESPONSE: Thanks for raising this important point. We now added a map to show the distribution of plot types across sites (see adapted Figure S1). Here you can see depicted in colors in Figure part d) that forest-derived vanilla agroforests are evenly distributed across villages but missing in two villages.

Fig. S1: Study design overview. a) The island of Madagascar in the Indian Ocean off the coast of East Africa with the SAVA region indicated. b) SAVA region. c) Study area with forest cover in 2017 (Vieilledent et al., 2018) roads, rivers, and the three main cities Sambava, Antalaha, and Andapa as well as the 10 study villages and Marojeje National Park with its two sampling sites therein. d) Schematic overview of the distribution of land-use types across 10 villages and two sampling sides in Marojeje National Park. e) Overview of transformation pathways of unburned and burned land-use types in the study region.

3) Were the vanilla yield measurements made on 36 vanilla pieds per plot? I think so from reading plot design but it isn't explicit in 'vanilla yield' section if that is 36 pieds total or per plot.

11. RESPONSE: Yes, the yield measurements were done on 36 vanilla plants (*pieds*) per plot. We added a graph with the plot design to support clarity (see Fig. S2). Furthermore, we added in the text "*pieds per plot*" whenever applicable.

Fig. S2: Design on the left shows the distribution of 36 vanilla *pieds* on a 25-meter radius plot in a vanilla agroforest (*pied* = combination of a vanilla vine and minimum one support tree). Photo on the right shows unique barcode label on vanilla *pied*.

4) I am not expert on bird counts but I was surprised only 2 repeat counts were done for each of the 30 plots. There is also no mention of the time of day or weather conditions when bird point counts were done (I'd expect them to need to be restricted to certain conditions especially with such few repeats).

12. RESPONSE: Thanks for this remark! Two repetitions were a compromise between logistical constraints and the maximum number of repetitions possible in the early morning hours when birds were most active. To compensate for the relatively low number of repetitions, we conducted long point counts of 40 min each (longer than it is typical in other ornithological studies, which often limit counts to e.g. 10 min). We now added more details on bird counts in the method section (see Line 1019). This made sense since the time needed to reach the plots from the village/campsite was rather long, making many short repetitions impossible. We took care to only do point counts in favorable conditions, to avoid potential biases. Weather conditions were thus indeed considered. We noted the conditions including rain (no rain, drizzle, light rain, heavy rain) and wind (Beaufort 1-12; Beer, 2013) before each point count. We only started point counts under good weather conditions, meaning no rain and wind equal or less than Beaufort 4. If weather conditions deteriorated for more than 10 min from the start of the point count, we aborted the point count and started again later or the next day under better conditions. More details on the bird counts can also be found in Martin et al. (2021).

5) While the SI is helpful, I did feel it could contain a bit more information and/or better links to. For example, the main text mentions taxonomic voucher specimens etc but I am not sure from what is presented that these could be found. On that note, is the raw data available alongside the paper? This would be extremely valuable (and is certainly best practice). Please make the data and code available if at all possible.

13. RESPONSE: We agree and updated the data available on OSF. It now also includes all species-by-site matrices for all taxa:

https://osf.io/j54fx/?view_only=1bd699c5cda64023963e058254a33eec

Uploaded files include:

- Species by site matrix (Presence/Absence) for 7 taxa
- Compiled data set for analysis for 5 land-use types
- Compiled data set for analysis for vanilla agroforests
- Pictograms and questionnaires for assessment of pollination labour input
- Data of work input invested in each agroforest with household members
- Questionnaire and farmer hand out for yield assessment
- R script

Thanks for highlighting the importance of voucher specimens. In the method section of butterflies, reptiles, amphibians, ants, and plants you can find the final location of storage, respectively: For amphibians and reptiles: “We stored DNA samples at the Evolutionary Biology laboratory at TU Braunschweig.” (LINE 1058). For butterflies: “All identified specimens are deposited at the insect collection in the Dept. of Crop Sciences, section Agroecology, University of Göttingen.” (LINE 1083). For ants: “We stored voucher specimens at Madagascar Biodiversity Center (MBC).” (LINE 1102). For trees: “Voucher specimens are kept at the National Herbarium Tsimbazaza (TAN), Antananarivo and the herbarium of the University of Mahajanga.” (LINE 1016). For herbaceous plants: “We stored all herbarium specimens at the Plant Biology and Ecology Department at the University of Antananarivo in Madagascar.” (LINE 1006).

6) The data relies on farmers to provide quite a bit of information eg on the land use history, age of plants and of course also the study of pollination labour input. There is no information provided about ethical review of this social research or basic information on who did the interviews, compensation etc etc. I also really feel, given the importance of the pollination labour in the model and the overall results, I'd like a bit more exploration of this data to help convince the reader that this is a valid measure of true labour input. Self-reported data on agricultural inputs can be difficult (though they had a nice short recall period which is reassuring). If you have data it might be nice to look to see if pollination labor input is predicted by available labour in the HH (only because if it is, it would give a reason why it varies so much, beyond simply being noise derived from the method). The land use history classification is critical and again needs more exploration.

14. RESPONSE: Thank you for these important remarks. We reply to them one by one:

1. Information on the assessment of age of plants

The age of plants was determined by interviews with farmers in the field. The Malagasy field researchers Evrard Benasoavina, Thorien Rabemanantsoa and Gatien Rasolofonirina walked with the farmer to each vanilla plant (see the question “Ask farmer: How many years ago was the liana at this pied planted?” in uploaded original interview (yield assessment) on OSF: https://osf.io/wa2xn/?view_only=1bd699c5cda64023963e058254a33eec). In preparation for the farmer interviews, we prepared handouts in Malagasy language to inform the farmers about the scientific goals and the content of our data collection (see handout on OSF: https://osf.io/ndrxg/?view_only=1bd699c5cda64023963e058254a33eec).

2. Information on the land-use history classification

Now, we provide a detailed description supporting our land-use history categorization in the supplementary material; including biotic and abiotic parameters such as tree composition and soil conditions.

To receive information on land-use history or farm history, interviews with farmers are common (Egeskog et al., 2016; Inoue et al., 2007). We did interviews with the plot owner. Questions on land-use history were binary (forest-derived or fallow-derived) and did not include information on the detailed land-use history (e.g., frequency of burning, past crop systems). Thus, we consider this self-reported data very reliable. The land-use categorization derived by farmers was confirmed by our visual plot inspections (forest-derived vanilla agroforests do have a quite distinctive vegetation structure compared to fallow-derived vanilla agroforests). Additionally, data on tree species composition and soil characteristics show evident differences between the categories and back up the binary land-use history categorization: (followed by the two examples).

Our interview-based categorization of land-use history was confirmed by the following analysis:

(1) Analysis of **tree species composition** shows that fallow- and forest-derived vanilla agroforests differ significantly in tree species composition. See the figure below (from Osen et al., 2021) showing the tree species composition of the five land-use types as revealed by non-metric multidimensional scaling (NMDS) based on Bray–Curtis dissimilarity.

Figure from (Osen et al., 2021) showing the species composition of the five land-use types as revealed by non-metric multidimensional scaling (NMDS) based on Bray–Curtis dissimilarity ($k = 2$; stress level 0.19855). We tested for dispersion effects among groups (followed by a pairwise permutation

test with 999 permutations) and identified dispersion differences between old-growth forest and all other land-use types. Pairwise permutational multivariate analysis of variance (PERMANOVA, 999 permutations; with Bonferroni correction) between land-use types revealed significant differences between old-growth forest and all other land-use types. Forest-derived and fallow-derived vanilla agroforests also showed significant differences in species composition; all other pairs were not significantly different from each other.

(2) **Soil analysis** (see Fig. S7) shows that our fallow-derived vanilla agroforests are associated to fertility-related variables such as an increase of calcium, pH, nitrogen and phosphorus which is common after slashing and burning the land (Béliveau et al., 2014; Hölscher et al., 1997).

Fig. S7: PCA based on eight soil characteristics including calcium, aluminum, pH (KCl), potassium, magnesium, nitrogen, phosphorus and carbon-to-nitrogen ratio. PC1 explained 45.2% and PC2 20.9% of the variation. The red ellipse groups variation of soil characteristics in fallow-derived vanilla agroforests and the blue ellipse groups the variation of soil characteristics in forest-derived vanilla agroforests.

3. Convincing arguments why self-reported data is reliable:

The self-reporting questionnaires were designed as diaries involving pictograms, e.g. to identify agricultural activities. This method was chosen as literacy levels are low in the region (Hänke et al., 2018). Hence, we did not ask participants for detailed documentation but simple daily counts of the number of working hours and the amount of money spent for hiring labour. The pictograms had to be filled every day as a diary. There is evidence that self-reported data can be more reliable than recall-survey (interviews done long after activity) in comparable rural settings, especially when using pictograms (Deininger et al., 2012; Wiseman et al., 2005). In our study region, pictograms were also successfully used to investigate the food security and food quality of vanilla farmers (Andriamparany et al., 2021).

4. Information on the assessment of pollination labour input

(1) Compensation:

“The household head who filled the pictograms received 10,000 Ariary (roughly 2,50 €) per month. Pictograms are drawings made by a local artist, which visually describe each of the working steps of vanilla cultivation (e.g. planting vanilla vine, weeding plantation, pollination). The amount of compensation was recommended as a reasonable compensation by locally

experienced Malagasy project members. The sum was handed out by the local research assistants at the end of each month.” (LINE 910)

(2) Ethical statement:

“All participants of the surveys were informed that participation was voluntary, that they can leave the interviews anytime and that all data is anonymized, i.e., no personal data will be published or shared with third parties. Guidelines of “Good Scientific Practice” by the University of Göttingen were adopted (adapted based on the recommendations of the codex for good scientific practice from the DFG, German Research Foundation; <https://www.uni-goettingen.de/en/good+scientific+practice/567647.html>).

The overall research project had an interdisciplinary advisory board that advised us on potential ethical concerns. The advisory board consisted of economists, ecologists, sociologists and jurists (find full advisory board information here: <https://www.uni-goettingen.de/en/project+advisory+board/531902.html>). Two of them work in the field of ethics (Prof. Dr. Franz-Theo Gottwald, Dr. Uta Eser). During yearly meetings of the project, methods were presented and discussed. “ (LINE 921)

(3) Pre-tests, interval of data reporting, intermediate data checks

“For the longitudinal survey, the questionnaire was pre-tested with 30 farmers in September 2017. All farmers were trained on how to fill out pictogram-supported questionnaires by the 4 research assistants. Subsequently, a feedback workshop was held in order to adapt and optimize the pictograms.

The pictograms had to be filled every day as a diary. Every fortnight, trained assistants visited farmers to collect the diary questionnaires. Data entries that appeared unusual were verified with the farmers by the assistants.” (LINE 891)

(4) Details on interviewer identity

“Co-authors HH designed the survey and trained the local assistants together with co-author BF. FA and the trained assistants conducted the interviews. The research assistants collected the data bi-weekly but visited the households weekly. Our research assistants were recruited mainly from the student population of the regional CURSA university center in Antalaha, which is located in the research area. The students speak the local Malagasy dialects. “ (LINE 927)

5. Why including only pollination labour input in the analysis?

We decided to use pollination labour input as only labour variable for the following two reasons. Firstly, pollination is by far the most labour-demanding part of vanilla production (Davis, 1983). Weeding, a commonly labour intensive task in crop production (Parish, 2012), is performed minimally for vanilla to avoid damaging of the vanilla roots (Nair, 2021; Odoux & Grisoni, 2010). The time spent on hand pollination of vanilla flowers is particularly critical, as no natural pollinators exist in Madagascar, and yields without pollination are extremely low (Westerkamp & Gottsberger, 2001; Wurz et al., 2021). Overall, hand pollination can make up to 40% of the production costs (Gregory et al., 1967) and trained people require one day to pollinate 1000-2000 flowers (Purseglove et al., 1981). Hence, we expected that the time invested into pollination is most decisive for vanilla yield. Secondly, pollination has a defined time frame because vanilla only flowers between October and December (Hänke et al., 2018). Thus, the hours of pollination labour input are easy to disentangle and define for the farmers, as family members are often directly involved or workers specifically hired. In contrast, other tasks such as weeding, pruning and planting often happen continuously and in parallel with other activities of farm households, and are thus harder for the farmer to depict in terms of working hours.

6. Is pollination labor input predicted by available labour in the HH?

We conducted a linear regression of available household labour with pollination labour /ha and found a strong correlation of the two. This correlation (p-value:0.005, estimate=52.82, $R^2=0.27$) supports the validity of our work input data.

Model results of linear model predicting pollination labour input by number of household members:

Predictor	Estimate	SE	p-value
Intercept	12.15	102.87	0.90690
Number of household members	52.82	17.38	0.00549

Very minor points:

I wonder if the abstract could be slightly reordered to find a way to mention earlier on which (very impressive) range of taxa the paper covers. The 1st few sentences talk only about ‘biodiversity’.

15. RESPONSE: Thanks for highlighting this. We included the range of taxa at the beginning of the abstract (LINE 112).

“Here, we focused on the relation between seven different taxa (trees, herbaceous plants, birds, amphibians, reptiles, butterflies, and ants) and yields in vanilla agroforests in Madagascar.”

Line 125 is slightly awkward-are you saying there has been decades of research about land use history, or agro-forestry generally (NB this is a TINY point).

16. RESPONSE: Thanks for spotting this. We added now agroforestry (“Surprisingly, despite decades of research in agroforestry, land-use history is not considered in most studies on tropical agroforestry” (LINE 140).

Line 628 could do with a reference.

17. RESPONSE: We added the following reference:

Van Dyk, S., Holford, P., Subedi, P., Walsh, K., Williams, M., & McGlasson, W. B. (2014). Determining the harvest maturity of vanilla beans. *Scientia Horticulturae*, 168, 249-257.

Reviewer #2 (Remarks to the Author):

(Please note, I attached a PDF of all these comments for ease of reading.)

Thank you for a very interesting manuscript that presents some very robustly-collected evidence for yield-biodiversity trade-offs in vanilla agroforestry systems in Madagascar, which is an important topic in the context of global change, sustainable production, sustainable development in one of the poorest parts of the world, and wildlife conservation. It is also relevant to current global initiatives around forest and landscape restoration, although the framing of the paper around this topic needs some improvement. I enjoyed reading it, and the results are nicely presented in Figure 1 and 2. However, I have some concerns about how the results are described in text (seeming to gloss over results that don't fit the narrative), a lack of clarity in presenting the relationships between management variables and yields/biodiversity, and the use of a 'multidiversity' measure that is hard to understand and which I don't think is informative. I think attention should also be paid to species composition and to total species richness accumulated over all plots – not just plot-level species diversity, as well as some more detail around the percentage endemism of the various taxa (differs greatly, I think, which is of importance for interpreting the results). I also think the language around restoration and degradation needs some careful consideration.

18. RESPONSE: Thanks for these important remarks, we have assessed them one by one below:

1) Result description and relationships between management variables and yield/biodiversity

To increase the transparency and clarity of our results, we state now directly where neutral, positive or negative relationships were found. Also, we added estimates and p-values to the text:

„ Comparing species richness and vanilla yield (kg/ha) in forest- and fallow-derived vanilla agroforests, we found that higher vanilla yields did not decrease the overall species richness, nor the richness of endemic species, for trees, herbaceous plants, birds, amphibians, and ants. Moreover, the overall and endemic diversity of all taxa combined (i.e., their mean normalized richness) was also not related to vanilla yields at plot level (Figure 1; Supplementary Table S1-2). We found a negative relationship between vanilla yield and butterfly species richness (Figure 1G, estimate= -0.179, p-value<0.001) and endemic butterfly species richness (Figure 2G, estimate= -0.104, p-value=0.043), a positive relationship with amphibians (Figure 1E, estimate=0.110, p-value=0.045), and, depending on land-use history, an either positive (forest-derived, estimate=0.289, p-value=0.028) or negative (fallow-derived) relationship with endemic reptiles (Figure 2F). Species richness of trees (Figure 1B), reptiles (Figure 1F) and mean normalized richness of all combined taxa (Figure 1A) were higher in forest-derived than in fallow-derived vanilla agroforests (Supplementary Table S1-2). Similarly, when looking at endemics, land-use history mattered for species richness of trees, herbaceous plants, birds, and ants and mean normalized endemic richness, with higher values in forest-derived compared to fallow-derived vanilla agroforests (Figure 2). “. (LINE 172-186)

2) Multidiversity

Based on your comment, we now use a new, more informative and accessible metric of the species richness across multiple taxa, i.e., the mean of the mean normalized species richness. This metric is a common method to aggregate information on species richness across multiple taxa (Collen et al., 2014; Tittensor et al., 2010; Van Elsas et al., 2012). We are confident that the adoption of this index increases the clarity, understanding, and interpretation of our results, as compared to the more complicated multidiversity index that we used beforehand.

To calculate the mean of the normalized species richness across taxa, we first scaled the richness per taxon based on its minimum and maximum observed values. Using this method, the data are linearly transformed to $y = (x - \min(x)) / (\max(x) - \min(x))$, where x is the set of observed values of species richness. As a result, the normalized species richness (y) then ranges between 0 and 1, with 0 referring to the lowest observed value and 1 to the highest observed value, respectively. We then calculated the average of the normalized values per taxa and plot to obtain the mean normalized species richness.

Our results were robust independently of the metric used, i.e., normalized richness or multidiversity. Therefore, and because we found it more intuitive, we now decided to keep the approach using mean normalized species richness in our manuscript.

Original Table S1: Linear mixed effect model (LMM) analyzing the effect of vanilla yield kg/ha (scaled & sqrt transformed) in interaction with land-use history on 1) multidiversity at a threshold of 50% and 2) endemic multidiversity at a threshold of 50%.

1) Multidiversity (50%)	Estimate	Std. Error	df	t-value	p-value
Intercept	0.448	0.038	11.191	11.784	<0.001
Vanilla yield kg/ha	-0.056	0.031	25.894	-1.806	0.083
Land-use history (VFOR)	0.079	0.060	25.937	1.313	0.201
Vanilla yield kg/ha : Land-use history	0.035	0.079	24.740	0.447	0.659
2) Endemic multidiversity (50%)	Estimate	Std. Error	df	t-value	p-value
Intercept	0.275	0.040	11.448	6.796	<0.001
Vanilla yield kg/ha	-0.006	0.026	23.455	-0.239	0.813
Land-use history (VFOR)	0.207	0.051	24.241	4.044	<0.001
Vanilla yield kg/ha : Land-use history	0.014	0.065	21.837	0.213	0.833

Updated Table S1: Linear mixed effect model (LMM) analyzing the effect of vanilla yield kg/ha (scaled & sqrt transformed) in interaction with land-use history on 1) mean normalized richness and 2) mean normalized endemic richness.

1) Mean normalized richness	Estimate	Std. Error	df	t-value	p-value
Intercept	0.131	0.005	11.020	24.603	<0.001
Vanilla yield kg/ha	-0.007	0.005	25.783	-1.425	0.166
Land-use history (VFOR)	0.047	0.009	25.945	5.077	<0.001
Vanilla yield kg/ha : Land-use history	-0.013	0.012	25.784	-1.043	0.307
2) Mean normalized endemic richness	Estimate	Std. Error	df	t-value	p-value
Intercept	0.059	0.006	12.214	9.098	<0.001

Vanilla yield kg/ha	-0.001	0.005	25.609	-0.201	0.842
Land-use history (VFOR)	0.061	0.010	25.783	6.183	<0.001
Vanilla yield kg/ha : Land-use history	-0.012	0.013	24.318	-0.931	0.361

3) Gamma diversity and species composition

To provide a more complete overview of biodiversity at different scales, we now provide the total species richness accumulated over all plots (i.e., gamma diversity; see figure below) as well as alpha and gamma diversity of each land-use type in Table S7.

Please note that we however limit the information on gamma diversity to supplementary information on key aspects only, as these were not the main focus of our study, which is about local (= alpha) diversity relations to yield increases.

Fig. S11: Species accumulation curves for all 5 land-use types, shown by the extrapolation curves based on species richness per sampling unit (Hill number $q = 0$). The solid line represents the

interpolation, whereas the dashed line represents the extrapolation. The shaded region represents the 95% confidence intervals. Non-overlapping of the confidence intervals represents a significant difference between two or more land-use types.

Table S7: Table of alpha diversity (mean species richness across all plot of each land-use type) and gamma diversity (sum of species richness of all plots of each land-use type). For gamma diversity: 10 plots of each land-use type were randomly selected (Note: 5 plots of herbaceous fallow and 5 plots of woody fallow = 10 fallow plots).

	FOR	FFOR	VFOR	FLW	VFAL
Mean normalized alpha richness	0.23	0.17	0.18	0.12	0.13
Mean normalized alpha endemic richness	0.22	0.14	0.12	0.05	0.06
Alpha Tree richness	64.40	41.10	31.30	10.20	12.72
Gamma Tree richness	257.00	230.00	198.00	51.00	69.00
Alpha Endemic Tree richness	47.30	27.70	19.90	3.60	2.50
Gamma Endemic Tree richness	180.00	152.00	125.00	21.00	21.00
Alpha Herb richness	15.50	14.10	20.60	25.80	21.05
Gamma Herb richness	77.00	75.00	92.00	108.00	92.00
Alpha Endemic Herb richness	5.50	4.40	3.90	2.25	2.45
Gamma Endemic Herb richness	24.00	27.00	19.00	10.00	12.00
Alpha Bird richness	12.30	6.90	7.60	6.30	6.35
Gamma Bird richness	34.00	23.00	25.00	18.00	18.00
Alpha Endemic Bird richness	7.40	2.00	2.30	1.00	0.85
Gamma Endemic Bird richness	24.00	10.00	13.00	6.00	7.00
Alpha Amphibian richness	9.30	5.10	4.60	3.50	4.55
Gamma Amphibian richness	32.00	26.00	14.00	6.00	12.00
Alpha Endemic Amphibian richness	9.30	5.10	4.00	2.80	3.80
Gamma Endemic Amphibian richness	32.00	26.00	13.00	5.00	11.00
Alpha Reptile richness	19.10	14.70	14.00	8.45	11.40
Gamma Reptile richness	66.00	56.00	44.00	23.00	34.00
Alpha Endemic Reptile richness	9.30	5.10	4.00	2.80	3.80
Gamma Endemic Reptile richness	32.00	26.00	13.00	5.00	11.00
Alpha Butterfly richness	6.20	5.10	11.30	9.95	10.50
Gamma Butterfly richness	20.00	19.00	44.00	39.00	35.00
Alpha Endemic Butterfly richness	6.10	4.40	6.80	5.10	6.45
Gamma Endemic Butterfly richness	19.00	14.00	24.00	19.00	21.00
Alpha Ant richness	16.40	15.30	18.40	13.25	15.45
Gamma Ant richness	55.00	58.00	71.00	47.00	54.00
Alpha Endemic Ant richness	9.60	6.60	6.90	2.27	3.67
Gamma Endemic Ant richness	32.00	28.00	32.00	10.00	20.00

In addition, we also added graphs showing changes in species composition of each taxon (Figure S4, see figure below) across land-use systems, and integrated this information in the main text (LINE 220-227) and as Supporting Information.

Figure S4: Species composition of the five land-use types as revealed by non-metric multidimensional scaling (NMDS) based on Jaccard dissimilarity. See icon attribution in table S18.

4) Endemism

We highlight the different levels of endemism for each taxon (in the main text and in the individual method sections, see LINE 209-212) and acknowledge that this can influence the interpretation of our results:

“Notably, the proportion of the recorded species that are endemic to Madagascar varied strongly among the studied taxa, with 51% for trees, 20% for herbaceous plants, 61% for birds, 98% for amphibians, 74% for reptiles, 58% for butterflies, and 45% for ants, respectively. Taxa with high levels of endemism are particularly prone to irrevocable biodiversity loss, if land-use negatively affects them.”

5) Restoration and degradation

We now define the terms restoration, rehabilitation, and degradation more explicitly to avoid misunderstandings:

“In order to prevent, halt and reverse the degradation of ecosystems, the United Nations has declared the years 2021-2030 as the Decade on Ecosystem Restoration (United Nations, 2020). Restoration is an approach, that can at least partially restore levels of biodiversity and ecosystem services (Chazdon, 2008). In this context, degradation represents the decline in biodiversity and ecosystem services, and restoration aims to prevent, halt and reverse the degradation of ecosystems (Chazdon, 2008; Gann et al., 2019).“ (LINE 127-132)

- What are the noteworthy results?

Vanilla production can produce high yields while supporting high levels of biodiversity in agroforestry systems, and offer a sustainable production system and a restoration option (for carbon and biodiversity) relative to shifting cultivation of rice in the studied landscape. They cannot replace old-growth contiguous forests in terms of biodiversity value, but represent a better alternative than land clearance. Meanwhile, there seems to be considerable room to improve vanilla yields through management without impacting biodiversity value, although currently this isn't shown as clearly as it could be.

19. RESPONSE: Thanks for raising this. We now added two sentences to highlight the low planting densities in Madagascar which give room to increase yields:

“Notably, planting density was generally low (mean=3284 plants/ha; SD=1444) in our study region, compared to intensively managed vanilla systems elsewhere. In Mexico, vanilla planting densities reach up to 5000 plants/ha in monocultures and up to 15000-20000 plants/ha in shade houses (Hernández Hernández, Juan Lubinsky, 2010). Generally, vanilla yields in Madagascar still have a high intensification potential (Martin, Wurz, et al., 2021).“ (LINE 404-408)

- Will the work be of significance to the field and related fields? How does it compare to the established literature? If the work is not original, please provide relevant references.

Yes, the work is original and contributes an extension and expansion of existing work on yield-biodiversity trade-offs in tropical countries by assessing a different crop, accounting for land use history, and by using rigorous methods to measure yields and biodiversity across multiple taxa. It contributes to current new debates on forest and landscape restoration

options, as well as pre-existing on-going debates around sustainable development and sustainable agriculture.

20. RESPONSE: Thanks for highlighting the originality of our study.

- Does the work support the conclusions and claims, or is additional evidence needed? Some additional analysis and clearer presentation of evidence is needed, as outlined in detailed comments to authors.

21. RESPONSE: We now provide additional analysis on gamma diversity, species composition, a new metrics to combine richness values across taxa. We also improved the clarity of our results by including additional statistical test results in the main text and naming relationships directly as neutral, positive or negative.

- Are there any flaws in the data analysis, interpretation and conclusions? - Do these prohibit publication or require revision?

I strongly recommend revisions to the analysis and interpretation as outlined in line-by-line comments to authors. I don't anticipate the key findings will necessarily change, but they should be more clearly presented, and I have suggested they conduct additional analysis on species composition which will then produce new and additional conclusions.

22. RESPONSE: Thanks, we now present (LINE 220) and discuss (LINE 354) results on species composition in the manuscript (line and provide a Non-Metric Dimensional Scaling (NMDS) analysis with a graphical representation of changes in species composition in the supplementary material (Figure S4). Furthermore, we have assessed the specific comments regarding analysis and interpretation mentioned by the reviewer.

- Is the methodology sound? Does the work meet the expected standards in your field? Yes the data collection methodology is very sound, and the statistical analysis applied is broadly fine, but I have made some suggestions around additional analyses that would add depth, and also asked for better presentation of the relationships, as I don't think one of the sets of models (yield-biodiversity direct relationship) is logical.

23. RESPONSE: Thank you for this important remark. We agree that the effect of yield on biodiversity might be driven to a large extent by underlying yield-related management variables, which we present in-depth in the second analysis of biodiversity depending on four management and six environmental variables (Paragraph "Effects of environmental and management variables on biodiversity"; LINE 240). We decided against a structural equation model due to little hypothesis evidence and the suitability of glmm to incorporate random effects and a large number of fixed effects. We provide a more extensive response to your comment on the yield-biodiversity relationships in response 18.

- Is there enough detail provided in the methods for the work to be reproduced? Yes, but a few additional details requested below to make this completely reproducible – namely more information about the land use types studied.

24. RESPONSE: We added more details about the fallow systems in the method section (LINE 795), provide photos of all land-use types (Fig. S3) and a map of the distribution of land-use types in the study region (Fig. S1).

Abstract

102: fewer, not less i.e. “fewer species” not “less species”

102-103: please reconsider the use of brackets for the endemic species vs all species reporting, as this is hard to understand.

25. *RESPONSE*: We replaced “less” with “fewer” species and omitted using brackets for the endemic species numbers.

Did you compare fallow-origin agroforests with old-growth forest? Worth stating that old-growth forest is best for biodiversity in the abstract.

26. *RESPONSE*: Thanks! We included this result now in the abstract. “Agroforests established in forests supported overall 23% fewer species and 47% fewer endemic species than old-growth forests, and 14% fewer endemic species than forest fragments. In contrast, agroforests established on fallows had overall 12% more species and 38% more endemic species than fallows.” (LINE 113-115)

105: do longer vines mean older vines? So long-established vanilla vines yield more? Or is length independent of age and related to something else?

27. *RESPONSE*: Vine length increases with age but according to our results, age does not affect vanilla yield (in contrast to the positive effect of vine length). Thus, we can assume, that vine length is not only driven by age but other undetected variables (such as possibly soil characteristics, climatic conditions or vine management).

Figure showing the relationship between vanilla vine length (cm) and vanilla vine age (years). Dots are raw data and solid (statistically significant, $P < 0.05$) or dashed (marginally significant, $0.05 < p < 0.1$) lines are back-transformed model predictions including 95% confidence interval.

Table showing the results of the linear model predicting vanilla vine length based on vanilla age.

	Estimate	Std. Error	t-value	p-value	
(Intercept)	554.3	83.02	6.677	3.02E-07	***
Vanilla vine age	31.89	15.4	2.071	0.0477	*

106: is “landscape forest cover” here old-growth forest, degraded forest, old fallows, or forest-derived-agroforests? Could you tell from the forest cover dataset? Are the forest fragments zero-yield, or are they productive somehow? I think that you only surveyed biodiversity in the old-growth forest park, but not in forest remaining in the landscape so would be helpful to make a distinction as this isn’t clear from the abstract.

28. *RESPONSE:* Thanks for highlighting this. Our “landscape forest cover” variable builds on the binary forest/non-forest variable from 2017 provided by (Vieilledent et al., 2018). To create the forest-cover variable, Vieilledent et al. (2018) used tree cover maps and satellite imagery with a tree cover threshold of 75%, which they considered appropriate to properly identify the moist forest in Madagascar. Agricultural lands such as tree plantations are excluded by combining historical forest maps by Harper et al. (2007) with up-to-date forest cover change maps by Hansen et al. (2013). Thus, forest cover was measured only inside historically assessed forest areas (at 30m resolution). To our knowledge, this variable is the most accurate and up-to-date information on forest cover that is currently available for Madagascar. However, we are aware that this variable represents a simplification and does not provide further information beyond this binary categorization. We cannot rule out the possibility that the pixels categorized as forest include land use types that are already degraded even though they have a tree cover above the threshold of 75%. However, the binary variable provides a good approximation of land with high tree cover > 75%.

Generally, forest fragments are not used for the cultivation of any crop but are used by the communities for the extraction of wood and non-timber products (Zaehringer et al., 2017).

108: biodiversity is unrelated to vanilla yields within agroforests, but not overall (old growth forest is best) so this should be rephrased to make that clear

29. *RESPONSE:* We now reformulated it to: “Vanilla yields were unrelated to biodiversity losses for six out of seven taxa, opening up possibilities for sustainable conservation outside of protected areas and restoring degraded land to benefit farmers and biodiversity alike.” (LINE 112)

110: please be careful with the use of the terms ‘degraded’ and ‘restoring’ here – I’m sure you’re aware of the diversity of definitions for these terms, so this should be better placed in its specific context i.e. development of vanilla agroforestry on non-productive and low-biodiversity-value scrub habitats formed by shifting agriculture (fallows) would be better for biodiversity (and livelihoods?) than... which alternatives? Re-clearance of fallows for rice or other crops? Allowing fallows to regenerate back to forest (does this happen, or are these areas degraded in the sense that they cannot recover back to the original ecosystem if left alone?)

30. *RESPONSE:* Thanks for this important comment! We follow your advice and write now “Vanilla yields were unrelated to biodiversity losses for six out of seven taxa, opening up possibilities for sustainable conservation outside of protected areas and restoring degraded land to benefit farmers and biodiversity alike.” (LINE 112)

Furthermore, we define now restoration in the introduction (LINE 129). See Response 32 for more details on definitions around restoration.

“Restoration is an approach, that can at least partially restore levels of biodiversity and ecosystem services (Chazdon, 2008). In this context, degradation represents the decline in biodiversity and ecosystem services, and restoration aims to prevent, halt and reverse the degradation of ecosystems (Chazdon, 2008)” (LINE 129).

Due to the commonly repeated slashing and burning of land under shifting cultivation, secondary forests are very rare in the study region (Styger et al., 2007).

Introduction

117: I think agroforestry is also widely seen as unprofitable and ‘backwards’ relative to intensified agriculture by e.g. government agencies, agricultural development agencies. So either rephrase to omit the idea that this is a commonly-held belief, or perhaps better, explore the idea that agroforestry is a ‘niche’ idea in many ways, and needs further evidence to convince those involved in agricultural policies that it is a good idea?

31. RESPONSE: Thanks for this feedback! Agroforestry is indeed often controversially discussed. We now write “Agroforestry opens up promising opportunities for ecosystem restoration (Schroth et al., 2004; Tschora & Cherubini, 2020), but more system-specific knowledge is needed for an even wider implementation of win-win solutions.” (LINE 132)

119: Please be a bit clearer about the definition of ‘degraded land’ here, and of ‘restoration’. Are you terming agricultural land that is productive as ‘degraded’ (reasonable from a biodiversity point of view, but unreasonable from a productivity point of view), or are you terming the fallows within shifting cultivation as ‘degraded’ because they are non-productive whilst they are fallow, but also worse for biodiversity than undisturbed ecosystems (although better than agriculture?) – in which case there is a difficulty when considering fallowing as an essential part of the sustainability of agriculture on tropical soils. Or, are you used degraded to define only areas that have been through the fallow/clearance process so many times that the soils are so degraded that they can neither regenerate to forest, nor support agriculture? Which I think would be unequivocal. Or, are you even able to explore perceptions of what degraded land means for people living in those landscapes? For ‘restoration’ – the typical SER definition involves moving a system back to a reference ecosystem, but that’s not what we are talking about here, so please provide a brief definition.

32. RESPONSE: Thanks for this important remark, we understand the criticism. Our definition of degradation has a strong ecological focus and refers to the observed loss of species (Fulgence et al., 2021; Martin, et al., 2021), ecosystem functions (Schwab et al., 2020) and agricultural benefits (i.e. lower yields) (Bruelle et al., 2015) on fallows under shifting cultivation. We did not include the perceptions of people (although these are of course very interesting and potentially important), neither did we assess the number of fallow cycles. In our case, restoration does not refer to returning the system to a reference forest ecosystem but to an improvement over previous land use (shifting cultivation) through agroforestry. We follow here the definitions by Chazdon and colleagues (Chazdon, 2008; Chazdon et al., 2016) who define restoration as an approach, that can at least partially restore levels of biodiversity and ecosystem services. Here, agroforestry is explicitly mentioned as an intervention to achieve that.

We add now, in the main text, definitions for restoration and degradation:

“In order to prevent, halt and reverse the degradation of ecosystems, the United Nations has declared the years 2021-2030 as the Decade on Ecosystem Restoration (United Nations, 2020). Restoration is an approach, that can at least partially restore levels of biodiversity and ecosystem services (Chazdon, 2008). In this context, degradation represents the decline in biodiversity and ecosystem services, and restoration aims to prevent, halt and reverse the degradation of ecosystems (Chazdon, 2008; Gann et al., 2019). Agroforestry opens up promising opportunities for ecosystem restoration (Schroth et al., 2004; Tschora & Cherubini, 2020), but more system-specific knowledge is needed for an even wider implementation of win-win solutions. This is of particular importance for degraded land that has much reduced biodiversity and services provisioning (Santos et al., 2019), and which makes up large shares of tropical landscapes characterized by subsistence agriculture and shifting cultivation (Styger et al., 2007)“ (LINE 127-136)

124: The effect of agroforestry on biodiversity at a landscape level may also be linked to their yields – if an agroforestry system yields less than a monocultural system, this may increase demand for land and drive deforestation (for example if farmers encouraged to adopt agroforestry to support biodiversity find their income declining as a result) - or indirectly cause deforestation by displacing forest clearance elsewhere (if agroforestry is done by different people)? You could briefly including some references here around these points for a broader context e.g. Kremen, Claire. “Reframing the Land-Sparing/Land-Sharing Debate for Biodiversity Conservation.” *Annals of the New York Academy of Sciences*, 2015, 1–25. <https://doi.org/10.1111/nyas.12845>. Please also check again the use of the term ‘degraded’ here – you seem to be using it as a synonym for cultivated land here. Perhaps once you have defined it earlier in this paragraph this can be clearer.

33. *RESPONSE*: Thanks for this important comment. We changed it from degraded to the wider term (historically forested) “open land” as also referred to in our project publication on land-use history (Martin et al., 2020):

“Whether tropical agroforests contribute to halting deforestation or accelerate biodiversity declines thus depends on their land-use history, meaning whether they are established on open land (i.e. cropland, pasture, fallow or degraded land) or by thinning forest...” (LINE 140)

Also, we added a sentence on the importance of productivity and conservation value at the landscape level:

“Furthermore, the productivity of agroforestry systems is decisive for their overall biodiversity value, because low-yielding agroforestry systems need more land to meet the same demands as provided by high-yielding monocultures, possibly leading to more forest conversion and biodiversity loss on a landscape level (Kremen, 2015)”. (LINE 144)

135: are forest-derived agroforests established within the protected forest area (the old growth forest used as a reference in the study) or within forest fragments in the landscape? If the latter, it seems you did not conduct any surveys within the forest fragments – do they differ much from the old growth continuous forest, and would this affect your conclusions around the biodiversity value of forest-derived agroforests in relation to forest?

34. *RESPONSE*: Thanks for pointing this out. Forest fragments represent the original system of forest-derived vanilla agroforest. We now include an additional boxplot with species richness of forest fragments to provide a comparison to the previous land use. We kept old-growth forest, as it provides the ultimate reference for intact ecosystems in the region. And yes, comparing forest-derived vanilla agroforest to forest fragment shows that species richness does not change significantly between these two systems:

“Compared with forest fragments, only butterflies significantly gained species (+82%) in forest-derived vanilla agroforests; losses or gains in species richness of all other taxa and of mean normalized richness were not statistically significant, also when focusing on endemic species (Supplementary Table S15).” (LINE 199)

Interestingly, analysis of changes in species composition indicates differences between forest fragment and forest-derived vanilla agroforests:

“Forest fragments and forest-derived vanilla agroforests differed significantly in species composition for herbaceous plants, butterflies, reptiles, amphibians, and marginally for ants.”. (LINE 222)

Thus, although losses in species richness are more evident in the conversion from old-growth forest to forest fragments than in the conversion from forest fragments to forest-derived vanilla agroforests, compositional changes continue to prevail across both trajectories.

136: what does fallow land look like? Is it shrub cover, recovering forest, or grassland? Would forest recover if fallows were left long enough, or is the soil too degraded? Do fallows have a social purpose?

35. *RESPONSE*: We added the following information to the methodology: “Generally, fallows occur in different forms in the study region. The characteristics of fallows depend on the frequency of past fires and the length of fallow periods in between crop cultivation (Styger et al., 2007). Frequent burning results in a loss of native and woody species and a dominance of exotic species and grasses (Styger et al., 2007). In later fallow cycles, fern species increasingly appear (Styger et al., 2007).

Due to the commonly repeated slashing and burning, secondary forests are very rare in the study region. Shifting cultivation prevails in Madagascar (Curtis et al., 2018), because it is an important option for people to grow food because means for agricultural intensification are scarce. According to our baseline survey (performed in 60 villages in our study region), 90% of the interviewed farmers grow rice for subsistence on top of growing vanilla (Hänke et al., 2018). Out of this sample, 64% of farmers grow rice in irrigated paddies and 26% of farmers use shifting cultivation.” (LINE 795-805)

In parallel to the data collection of this study, we performed also a socio-ecological survey in the same study villages. This study by Estelle Raveloaritiana et al. is currently under revisions in *People & Nature*. It shows that fallow lands are important sources of provisioning ecosystem services for more than 50% of households. People collect products from fallows for food security, cultural needs and livelihoods. In particularly woody fallows are an important source of many uses providing medicine, firewood and fodder to households.

141: the word “urging” does not quite make grammatical sense here – suggest “generating a need” or similar instead.

36. *RESPONSE:* We adapted the wording.

143: need comma after “30 vanilla agroforests”

37. *RESPONSE:* We added the comma.

Results

153: why have you put brackets around “endemic”? I think it would be helpful in the introduction to explain that you assess all species, and endemic species separately, and why.

38. *RESPONSE:* Thanks for this suggestion. We put endemic into brackets to avoid repetitions of words, e.g. “species richness and endemic species richness..”. We follow your suggestion of writing it out and added now in the introduction “We differentiate between overall species richness and endemic species richness to account for Madagascar’s high share of endemic species and their vulnerability to land use (Broennimann et al., 2006; Chaudhary et al., 2015)” (LINE 165)

I think it would also be helpful to explain what ‘multidiversity’ means here given the structure of the paper with the Methods at the end. Is this just species richness of all taxa combined? There is a threshold mentioned in the caption of Figure 1 but this is not explained. The reader should be able to interpret the figure in its entirety with the caption without referring to the methods, so please explain it here.

Note – reference nos. 72 and 73 (in Methods) do not use the term ‘multidiversity’ at all, but refer to metrics of ecosystem function based on e.g. soil microbial diversity, where a threshold is applied to ecosystem functioning, not to the level of biodiversity used to explain ecosystem functioning? This does not seem relevant to the message of this paper, which does not state that it aims to explore ecosystem functioning. Instead, ‘multidiversity’ appears as a synthetic metric for diversity of the other taxa, but it is not very clear why this is informative relative to the taxon specific species richness measures. Perhaps this just needs better justification?

39. *RESPONSE:* Thank you for this comment. Based on the feedback of the other reviewers concerning multidiversity not being easy to interpret, we now use mean normalized richness, a common method used in multi-taxa studies (Collen et al., 2014; Tittensor et al., 2010). Please see our more detailed response to this concern in response number 18. The results are qualitatively the same with Mean normalized richness and multidiversity, highlighting the robustness of our findings. We are confident that this new measure of biodiversity across taxa is easier to interpret and more informative.

155: You state “did not find” but then “few exceptions” – so in fact, you did find some relationships, so this is a little disingenuous. Please rephrase. I think it would be helpful here to mention the effect sizes – the numbers of species per plot is rather small, and so is the effect size where there are statistically significant relationships. Please also report the key results of the statistical tests in-line, or on Figure 1 – the reader should not have to look in the supplementary information to find out the the results of the statistical test and effect sizes.

40. *RESPONSE:* Thanks for this. We deleted the sentence “overall, we did not find significant relationships between species richness or multidiversity and vanilla yield” and the entrance

“There are only a few exceptions.” and state now directly where neutral, positive or negative relationships were found. Also, we added estimates and p-values inside the text:

„ Comparing species richness and vanilla yield (kg/ha) in forest- and fallow-derived vanilla agroforests, we found that higher vanilla yields did not decrease the overall species richness, nor the richness of endemic species, for trees, herbaceous plants, birds, amphibians, and ants. Moreover, the overall and endemic diversity of all taxa combined (i.e., their mean normalized richness) was also not related to vanilla yields at plot level (Figure 1; Supplementary Table S1-2). We found a negative relationship between vanilla yield and butterfly species richness (Figure 1G, estimate= -0.179, p-value<0.001) and endemic butterfly species richness (Figure 2G, estimate= -0.104, p-value=0.043), a positive relationship with amphibians (Figure 1E, estimate=0.110, p-value=0.045), and, depending on land-use history, an either positive (forest-derived, estimate=0.289, p-value=0.028) or negative (fallow-derived) relationship with endemic reptiles (Figure 2F). Species richness of trees (Figure 1B), reptiles (Figure 1F) and mean normalized richness of all combined taxa (Figure 1A) were higher in forest-derived than in fallow-derived vanilla agroforests (Supplementary Table S1-2). Similarly, when looking at endemics, land-use history mattered for species richness of trees, herbaceous plants, birds, and ants and mean normalized endemic richness, with higher values in forest-derived compared to fallow-derived vanilla agroforests (Figure 2). “ (LINE 172-186)

160: Again, please increase the clarity of writing around endemic vs all-species patterns e.g. change “multidiversity and species richness did not differ” could be rephrased as “For all measured species, multidiversity and species richness only differed for...” Because again, you state “did not differ” and then go on to say there were differences after all.

41. RESPONSE: We implemented your suggestion and omitted the phrase “did not differ” and directly highlight the differences. See result section “ Effects on land-use history on biodiversity of agroforests (LINE 188-219)

Can you also provide a little detail on the % of all species that are endemic? I believe endemism of some taxa in Madagascar is extremely high, making those groups of potentially increased conservation concern.

42. RESPONSE: This is an important point underlying the conservation need. We now list the level of endemism of each taxon in the respective method section and result section:

“Notably, the studied taxa held different levels of endemism (endemic to the country of Madagascar) overall with 51% for trees, 20% for herbaceous plants, 61% for birds, 98% for amphibians, 74% for reptiles, 58% for butterflies, and 45% for ants, respectively. Taxa with high levels of endemism are particularly prone to irrevocable biodiversity loss if land-use negatively affects them.”

169: the phrasing reads “biodiversity changes...was influenced by... taxa” which doesn’t quite make sense – I think you mean, responses differed by taxon?

43. RESPONSE: Thanks! We write now: “The direction and magnitude of biodiversity responses at local scales, i.e., plot level, differed by land-use history and taxa.” (LINE 189)

169 – 171: you start with “on the one hand” but then use “whereas” instead of completing with “on the other hand” – you could revise.

44. *RESPONSE:* We omit now “on the one hand” and “on the other hand”.

171: at this point, and when looking at Figure 1, I really want to know about species composition, not just richness, especially when comparing forest to vanilla agroforests. Species richness per plot can mask huge changes in species composition, with implications for conservation concern and ecosystem functioning. I don’t expect the same 10 reptile species per forest plot are found in herbaceous fallows. An NMDS analysis clustering species by system (forest, vfor, vfall, and fall) could be informative and be placed in the SI with a brief description of results in main text if you are struggling for space. Similarly I think cumulative species richness across each system (i.e. rarefaction curves) would be helpful to catch species accumulation in each system. I would like to see how the rarefaction curve flattens (or not) for each sampled system to assess sampling completeness, and also to get an idea of how each system contributes to biodiversity more widely i.e. are the same species in fallows found again and again as they are more homogenous than the agroforests?

45. *RESPONSE:* We now provide rarefied and extrapolated curves for all five land-use types based on species richness per sampling unit (Hill number $q = 0$, presence-absence data; see Figure S3). Here, we found that the rarefaction curve tended to reach an asymptote in fallow for amphibians, birds, reptiles, and trees. The rarefaction curves of ants, butterflies, and herbs didn’t reach an asymptote across all land-use types. However, we have in general good sample completeness when computing the sampling coverage of each taxon across land-use types (see Table S10). Only, a few cases were observed to present lower sample completeness as such for trees in fallows, forest-derived vanilla, and forest fragment, and amphibians in forest fragments (see Table S10). In summary, overall sampling coverage was satisfactory across taxa and land-use types. Thus no sampling bias for any of the land-use types existed. We expect that increasing our sampling effort would have resulted in greater differences between land-use types.

In addition, as suggested, we now include an NMDS analysis of changes in species composition with land use (Figure S4). NMDS analysis shows, that although losses in species richness are more evident in the conversion from old-growth forest to forest fragments than in the conversion from forest fragments to forest-derived vanilla agroforests, compositional changes continue to prevail across both trajectories.

Please see Figure S3 (species accumulation curves) and Figure S4 (species composition) in Response 18.

We included the following sentence in the main text:

“Sampling coverage across land-use types was on average 84% across all taxa, indicating satisfactory sampling effort.” (LINE 214)

Table S10: Sampling coverage (SC) depending on taxon for each land-use type (FOR=Old-growth forest, FFOR= Forest fragment, VFOR=Forest-derived vanilla agroforest, FAL=Fallow, VFAL=Fallow-derived vanilla agroforest. Sampling computed with *iNext* function, Hill-number $q=0$, datatype=incidence raw, endpoint=20. The last row displays average values of sampling coverage for each taxon.

	SC Birds	SC Butterflies	SC Trees	SC Ants	SC Herbs	SC Reptiles	SC Amphibians
FOR	0.91	0.86	0.80	0.86	0.73	0.89	0.85
FFOR	0.86	0.81	0.70	0.80	0.69	0.87	0.67

VFOR	0.85	0.86	0.62	0.80	0.79	0.89	0.88
FAL	0.95	0.90	0.58	0.93	0.88	0.92	0.97
VFAL	0.95	0.91	0.87	0.92	0.86	0.91	0.89
Mean	0.90	0.87	0.71	0.86	0.79	0.90	0.85

170 onwards: what are these % changes reporting? Differences in median values? Great to have effect sizes, but also please incorporate them in the section on yields vs biodiversity (as mentioned above)

46. *RESPONSE*: These % changes were based on mean values. We specified now in the caption of Table S3, S8-9 how the percentage changes were calculated, e.g.:

“ **Table S3**: Mean values of (endemic) mean normalized richness and (endemic) species richness of seven taxa in forest-derived vanilla agroforest (**VFOR**) and old-growth forest (**FOR**) with absolute percentage difference (mean richness FOR- mean richness VFOR) and relative percentage difference (=mean richness VFOR*100/mean richness FOR; indicated in yellow) and relative percentage gain or loss from old-growth forest to forest-derived vanilla agroforest (=absolute difference*100/mean richness FOR).”.

To note, the statistical output of the land-use types comparison is provided in the supplementary Table S6.

179: please finish the sentence by reminding the reader which comparison this refers to e.g. vanilla vs fallows.

47. *RESPONSE*: Thanks, we added the two land-use types to which the comparison refers to (LINE 206).

Figure 1: Looking at the per-plot species richness patterns, I am interested to know more about two things – cumulative species richness (i.e. rarefaction of species richness with accumulated sampling) between the four measured systems), and species composition. To me, these would be more informative and interesting that the multidiversity measure, which I find difficult to interpret and understand in relation to application or policy based on these findings. I have comments about the multidiversity measure elsewhere in this review, but if you are keeping it in, I think you need to explain what it means in the figure legend, so the figure can be interpreted standalone.

48. *RESPONSE*: We omitted multidiversity and replaced it with mean normalized richness (see reply to comments before). We now added extrapolated curves of species diversity per sampling unit (Figure S3) and graphs on the species composition in the appendix (Figure S4).

We added in the main text “. Sampling coverage across land-use types was on average 84% for all taxa, indicating a satisfactory sampling effort. In addition, rarefaction curve for all five land-use types based on species richness per sampling unit tended to reach an asymptote in fallow for amphibians, birds, reptiles, and trees. The rarefaction curves of ants, butterflies, and herbs did not reach an asymptote across all land-use types.” (LINE 217)

Also, we dedicated a whole paragraph to describing the compositional changes across land-use types:

“Across all taxa, species composition changed significantly from old-growth forest to forest-derived vanilla agroforest (Figure S4). Forest fragment and forest-derived vanilla agroforest differed significantly in species composition for herbaceous plants, butterflies, reptiles, amphibians, and marginally for ants (p-value=0.023). Comparing fallow to fallow-derived vanilla agroforest, we found significant changes in the composition of herbaceous plants, birds, reptiles, butterflies, and ants. Trees and amphibians did not differ in species composition between fallow and fallow-derived vanilla agroforest.” (LINE 226)

Please explain the boxes and whiskers as well as the median line for boxplots. Please provide stats results on the figure, if not in-line in the main text referring to this figure.

49. *RESPONSE*: We added on top of “The line inside the boxplot represents the median of each land-use type. Outliers are shown as dots” the following new sentence: “The lower and upper boundaries of the boxplot show the 25th-75th percentiles of the observational data, respectively and whiskers show the 1.5 interquartile range.” (LINE 278).

Figure 2: if you need to save words and the journal allows, you could avoid duplicating the explanation of the colours/lines/boxplots and just say “as for Figure 1”

50. *RESPONSE*: Thanks for this good suggestion! We will apply it in case the journal asks us to shorten the caption!

180: please provide figures of yield relationships in the main text, not just tables (nobody likes tables really, they slow down interpretation). Does vanilla vine length correspond to age, or is it something else i.e. faster growing vines due to being better fertilised, or different planting stock? Please explain this further.

51. *RESPONSE*: We added now a graph of the variables marginally or significantly affecting vanilla yield (see Figure 3). Furthermore, we added Figure S5 showing raw data of vanilla yield kg/ha depending on all variables (planting density, vanilla vine length, pollination labour input, elevation, vanilla plant age, canopy closure, soil characteristics, understory vegetation cover, slope and landscape forest cover) and Figure S7 showing the positive or negative relationships of all taxa with environmental or management variables.

Thanks for the question on the relationship between vanilla vine length and age. As mentioned above (Response 27), we find that vine length increases with vine age. However, according to our results, age did not have an effect on vanilla yield (in contrast to the positive effect of vine length on yield). Thus, we can assume that vine length is not only driven by age but other undetected variables (such as possibly soil characteristics, climatic conditions or vine management). Further research is needed to identify the additional drivers of increasing vine length.

181: “were” should be replaced “was” in relation to “yields”

52. *RESPONSE*: Corrected.

183: Can Table 1 include effect sizes if you are keeping it in (i.e. if not replacing it with a figure with the statistical results reported on it). Re: hand pollination, why don't people pollinate more? Is it labour constraints?

53. *RESPONSE*: We replaced Tables 1 with Figure 3 and report estimates in Table S14 as well as in the result section (LINE 228).

Our analysis of pollination labour input depending on the number of household members shows a positive association between the two variables. As the pollination of vanilla flowers in Madagascar is mainly based on family labour, this may indeed indicate a pollination labour constraint. Nevertheless, we argue for a careful interpretation of the marginal positive effect of pollination labour on vanilla yield, as we could not control for the number of flowers present on the plantation (i.e. low pollination labour input can also be related to a low amount of vanilla flowers present). However, the number of flowers is a challenging variable to measure, because flowering happens successively between October to January. In addition, flowers are also purposely removed by farmers to produce longer beans (Havkin-Frenkel & Belanger, 2010). We included now in the main text:

“While hand pollination is critical for achieving high vanilla yields (Wurz et al., 2021), this finding needs to be interpreted carefully, since it may also reflect an overall increase of pollination input with a high number of vanilla flowers present.” (LINE 372)

191: Again, a problem with the phrasing around direction of results here – “in general...unrelated” but then when you look at Table 2, you find that there do in fact appear to be trade-offs for multiple variables (vine length, vine density, pollination). Indeed, there seem to be some key trade-offs that need further interrogation, namely why increased vanilla plant density was related to reduced tree species, and why longer vines were associated with trees and reptiles (perhaps for the discussion, perhaps you have good reason to believe these relationships are non-causal, but needs not to be glossed over) I think here you need to link back to Figures 1 and 2 to explain that these variables may be behind the relationships between yield and biodiversity.

54. RESPONSE: Thank you for this valid comment. We now start with the key trade-offs followed by yield-unrelated drivers of species richness (LINE 241).

“We found trade-offs between yield-increasing variables (vanilla planting density and vanilla vine length) and overall and endemic species richness of trees and reptiles (Table 1, Supplementary Table S15-17, Figure S7) but win-wins with endemic species richness of herbaceous plants. Firstly, agroforests with higher vanilla planting density had lower tree richness (estimate= -0.155 SE=0.066, p-value=0.019), lower endemic tree richness (estimate= -0.337 SE=0.096, p-value<0.001) and marginally reduced endemic reptile richness (estimate =-0.096, SE=0.052, p-value=0.068) but higher endemic herbaceous plant species richness (estimate= 0.176, SE=0.063, p-value=0.005. Secondly, vanilla vine length was related to overall fewer tree species (estimate= -0.221, SE=0.056, p-value=0.001), endemic tree species (estimate= -0.553, SE=0.104, p-value=0.001) and reptile species (estimate= -0.018, SE=0.071, p-value=0.016) at plot level. Notably, species richness of birds, amphibians, butterflies, and ants was driven by environmental and management variables unrelated to vanilla yields (Table 1, Supplementary Table S15-17, Figure S7).“

Furthermore, we pick up the trade-offs (e.g. tree species richness declines with increasing vanilla density) in the discussion and discuss compromises/solutions (LINE 375):

“Contrasting with the stable species richness of the majority of taxonomic groups, we found that agroforests with a high planting density of vanilla and long vanilla vines had reduced species richness of trees (overall and endemic) and tended to support fewer endemic reptile species. This indicates trade-offs between conservation and production goals, particularly for trees, which represent a keystone structure on which other species depend (Tews et al., 2004):

Doubling planting density from 3000 to 6000 vanilla plants/ha or vine length from 300 to 600 cm, corresponded to a decrease in tree richness by 27% or 23%, respectively. By contrast, almost tripling planting densities to 8500 plants/ha or quadrupling to 1200 cm vine length lowered tree species richness by 55% and 52%, respectively. Thus, intermediate increases in planting density and vanilla vine length can represent a compromise for tree conservation and vanilla production. High variations in planting density, as well as additional effects affecting yields (e.g., labour input), may be the drivers of tree species decreasing with more and longer vanilla plants, but not with increasing yields. Furthermore, our study highlights that tree richness strongly depends on landscape forest cover as well as canopy closure which can be achieved by conservation of remaining forest fragments as well as farmer incentives for maintaining old trees and dense tree canopies. While the mechanisms underlying the negative relationship of vanilla planting density with tree richness seem obvious, more research is needed on the mechanisms behind the negative relationships between vine length and reptile diversity.”

Indeed, I am actually unsure whether yield and biodiversity should be modelled directly in relation to each other in a GLMM as you have done, at all. Biodiversity does not respond directly to yields – as you explore here, it responds to management, as does yields, so implying that biodiversity responds to yields or vice-versa doesn’t quite make sense. Although I am reluctant to posit alternative analytical approaches unnecessarily, in this case it seems it would be more logical to link together the yield-management-biodiversity relationships together more clearly in your main figures, not just the yield-biodiversity relationships. Whether you would be best doing this using a combined model of some kind (e.g. using a path analysis type approach) or whether you are able to display the results of the yield and biodiversity responses to each of the management variables on the same figure, but modelled independently, I am not sure.

55. RESPONSE:

Thank you for these comments on our modeling approach. We agree that the effect of yield on biodiversity might be driven to a large extent by underlying yield-related management variables, which we present in-depth in the second analysis of biodiversity depending on four management and six environmental variables.

A pathway analysis (e.g. structural equation model) as you suggested, is indeed a useful statistical approach to highlight indirect as well as direct relationships. However, SEMs require valid hypotheses for direct and indirect paths (based on literature). In terms of vanilla, literature evidence for relationships of management and environmental variables with yield and biodiversity is scarce and scattered across a few literature resources such as Havkin-Frenkel et al. (2008) and Davis (1983). Therefore, it will be challenging to build robust links in the SEM as strong evidence for vanilla agroforest is missing.

Furthermore, we decided to use a Generalized Linear Mixed-Effects Model (glmm) instead to identify the drivers of yield and biodiversity because it allowed us to consider the study design (account for village ID as random effect). Also, it computed robust results based on the number of replicates (30) available and the included number of environmental/management variables (10).

In addition, we provide figures with the individual relationships between yield/biodiversity with each of the environmental/management variables (Figure S5/S7).

Finally, we argue to present the relationship between yield and biodiversity individually, as this is a commonly-used relationship (Geiger et al., 2010; Tamburini et al., 2020). It is also a central question (timely and politically important) as well as contributing to the debate on land-sharing/sparing which needs knowledge on yield-biodiversity relationships to find win-win solutions (Kremen, 2015). The strength of the relationship can be compared to other systems and provide a straightforward measure of the biodiversity value of a cropping system as provided also by influential research papers such as Clough et al. (2011) and Bisseleua (2009). Here, the nature of the relationship can take different forms such as linear, neutral or hump-shaped, which gives important information for practitioners as highlighted by Perfecto (2005).

Sample section of Fig. S7: (Endemic) species richness of seven different taxa depending on planting density, vanilla vine length, pollination labour input, elevation, vanilla plant age, canopy closure, soil characteristics, understory vegetation cover, slope and landscape forest cover. Dots are raw data and solid (statistically significant, $P < 0.05$) or dashed (marginally significant, $0.05 < p < 0.1$) lines are back-transformed model predictions including 95% confidence interval.

192: this is the wrong supp table – should be Tables S7 and S8 I think. Please provide key stats results in the main text or a figure, and don't force the reader to the supp material.

56. *RESPONSE:* We adapted the wrong numbering! We provide numbers of biodiversity increases or decrease in percentage in the section “Effects of land-use history on biodiversity of agroforests”.

Table 1: why were the datapoints with missing data for a key variables (pollination labour input) imputed and not omitted? Do results differ if you omit these plots?

57. *RESPONSE:* Please see a detailed response to your comment in Response No.8.

We performed a linear regression of available household labour with pollination labour /ha and found a good correlation between the two (p-value:0.005, estimate=52.82, $R^2 = 0.270$). Based on the number of household members (for which we have available data for the three missing plots), we then predicted expected pollination labour input for the three missing plots, using the estimated relationship of the two from the mentioned regression. We performed a sensitivity test including and excluding the three plots. Comparing the model results of the yield predictors using three different approaches (including averaged values, including fitted values, excluding missing plots) shows that pollination labour input has a marginally

significant or no significant effect. Thus, the choice of method does not affect our final research conclusions.

Table S12 shows the pollination hours have a very wide range, spanning one order of magnitude, as do vanilla yields. Out of interest, what happened in the plot with 0 vanilla yields? Was there more than one with total failure of yields?

58. *RESPONSE:* Thanks for spotting this! The plantation with zero kg yield was three years old (vanilla plant mean age: 2.9 yrs) and did not yield vanilla pods as expected. Typically, vanilla starts yielding with 3 years (Havkin-Freckel, 2008). We nonetheless decided to keep it in the dataset, as we thus span a broad gradient of vanilla farming from young to old plantations.

I don't understand why the p-values in Table 1 are different to Table S13 where you report the full results of the statistical test. Indeed, the full model results, estimate and SE should all be reported in the main text.

59. *RESPONSE:* You are correct – there was a mistake in the former table S13 (now table S14). We include now test results also in the main text.

“Vanilla yields varied widely and averaged at $105 \pm \text{SD } 100$ kg/ha. Vanilla yields increased with planting density (estimate= 2.901, SE=0.415, p-value=<0.001) and vanilla vine length (estimate= 2.650, SE=0.393, p-value=<0.001; Table 1; Supplementary Figure S5 & Table S14). Moreover, vanilla yield (kg/ha) tended to increase with labour input for hand pollination of vanilla flowers (estimate= 0.897, SE=0.469, p-value=0.069; Table 1).” (see LINE 233)

Are these results from the full model (inc all variables as in Table S5) or the best reduced model (Table S6)? The p-values don't match up to either. I actually don't understand Table S5 at all – what do the x2 values refer to?

60. *RESPONSE:* In former Table S5 (now Table S13), we used likelihood ratio tests using maximum likelihood estimation (Zuur et al., 2009) to assess the statistical significance of individual variables. Specifically, we compared the full model versus a reduced model without the individual variable (single term deletion). Only significant variables (p-value < 0.05) were retained in the final model (former Table S6, now Table S14). The symbol χ^2 refers to the Chisq-value of the anova comparison (full model against reduced model, e.g. minus variable pollination labour input). We now explain in the caption of S6: “Chisq-values based on model comparison with *anova* function = χ^2 .” We hope this improves clarity.

Table S2: Should reference Supp Table S7 and S8 not S5 and S6?

61. *RESPONSE:* Thanks for spotting this. We updated the table numbering.

266: your plots seem to yield about 1/3rd of the ‘productivity benchmark’ mentioned here – why?

62. *RESPONSE:* Yes, the majority of plots (20 plots) yield below/equal 100kg/ha. But 10 plots yield above 100 kg/ha up to 372 kg/ha showing yields above the common production benchmark are possible. Our highest yielding plots (372kg/ha & 314kg/ha) also hold the highest pied density (6415 plants/ha & 8520 plants/ha) confirming the results of our yield models where pied density was the strongest yield driver. As vanilla farming is highly labor-intensive (planting of vines, pruning of tutors, looping and cutting of vines, weeding of

understory, pollination, guarding of plantation from thieves, etc.) investment in vanilla agroforests strongly depends on available family labour and monetary resources for paid labour – based on previous research we know that field size, gender as well as the existence of contracts are good indicators of labour availability. Here, we found that only 15% of farmers in our study region have formal contract farming arrangements (mainly for certification such as organic farming, Rainforest Alliance and Fairtrade), fields are generally small, and female-headed households have fewer labour resources available, in particular when it comes to securing the land from thieves.

Also, due to the vanilla boom, many farmers are new to vanilla farming and some started rather recently to plant vanilla. 350 kg/ha thus equals a professionally managed vanilla plot if the farm has a “full-employment farm size” (=a maximum vanilla area that can be managed primarily through family labour) (Hänke, 2019). Finally, note that we also find that farmers often diversify their income sources, and are not always solely relying on vanilla production for livelihood (Andriamparany et al., 2021; Kunz et al., 2020).

269: please explain further the role that fallow land plays in these landscapes – do they have a social or other agricultural function? Are there both costs and benefits of turning fallows into vanilla agroforests, are they done by different people? Do fallows have cultural value?

63. *RESPONSE:* We now give more detail on the value of fallows for people and the costs and benefits of their transformation.

“However, ecological restoration also has to consider the value of fallow land to reconcile livelihood and conservation needs (Fischer et al., 2021). Fallow land can take different forms providing a wide array of benefits to people (Zaehring et al., 2017). For example, right after subsistence rice production, fallows are often used for livestock grazing (Styger et al., 2007). Indeed, the transformation of fallow land into agroforests can result in losses of provisioning ecosystem services (e.g., firewood, wild foods, timber) (Zaehring et al., 2017). However, there is not yet a shortage of fallow land in northeastern Madagascar, and the loss of provisioning services through conversion to agroforests is readily offset by the associated benefits (e.g., cash crops, carbon storage) (Zaehring et al., 2017). A heterogeneous landscape with multiple land-uses is important to satisfy the needs of rural communities.” (see LINE 445)

275: I don’t think that this assertion that management variables affecting biodiversity are different to those affecting yield are evidenced strongly in the results, as currently presented. This may be because a different type of analysis is needed, or the same analysis but presented differently so that in one figure a reader can readily see that yields and biodiversity have opposing responses to the same variable. As the results are currently presented, it seems the text emphasises the lack of relationships between yield and biodiversity, while the figure and tables reveal there are some apparent relationships which then need to be explained away.

64. *RESPONSE:* Thanks for this comment. We write now:

“Nonetheless, trade-offs existed between yields and species richness of endemic and overall butterflies as well as endemic reptiles. In conclusion, we identify the preservation of remaining old-growth forest and forest fragments as the top conservation priority. Vanilla yields were unrelated to biodiversity losses for six out of seven taxa, opening up possibilities for sustainable conservation outside of protected areas and restoring degraded land to benefit farmers and biodiversity alike.” (see LINE 121)

309: But what about species composition? And accumulated species richness?

65. *RESPONSE*: Please see our response (response 18) above and the associated additions in the manuscript (see LINE 220).

341: “complementary” (not complimentary, which is to pay a compliment i.e. say something nice to someone”)

RESPONSE: Thanks, corrected.

Methods

553: can you give percentage endemism here, or else, report in the results (already mentioned above)

66. *RESPONSE*: For each taxon, we provide now the number of endemic species found in the present study. (see result section in LINE 209).

554: please provide more information here about the hill rice system – what are the fallows like, why is this system used, and are the people growing rice the same as those growing vanilla?

67. *RESPONSE*: We now write in more detail what the fallows looked like, the reasons why shifting cultivation prevails and some numbers on the percentages of farmers growing rice on top of cultivating vanilla. (see methods, section “plot selection” LINE 209-212)

“Generally, fallows occur in different forms in the study region. The characteristics of fallows depend on the frequency of past fires and the length of fallow periods in between crop cultivation (Styger et al., 2007). Frequent burning results in a loss of native and woody species and a dominance of exotic species and grasses (Styger et al., 2007). In later fallow cycles, fern species increasingly appear (Styger et al., 2007). Due to the commonly repeated slashing and burning, secondary forests are very rare in the study region. Shifting cultivation prevails in Madagascar (Curtis et al., 2018), because it is an important option for people to grow food because means for agricultural intensification are scarce. According to our baseline survey (performed in 60 villages in our study region), 90% of the interviewed farmers grow rice for subsistence on top of growing vanilla (Hänke et al., 2018). Out of this sample, 64% of farmers grow rice in irrigated paddies and 26% of farmers use shifting cultivation. “ (LINE XY)

564: why did you exclude villages with coconut?

68. *RESPONSE*: Thanks for raising this issue. Importantly, we excluded villages with large coconut plantations. Specifically, there are a few coconut monocultures that stretch over 50 sq km along the coast in the study region. We excluded villages with such extensive coconut plantations to compare villages with similar land-use compositions in which we were able to find the all land-use types we were interested in.

570: why did you not conduct surveys in forest fragments? My concern is whether forest fragments contain the same species as continuous forest, which are then lost on conversion to agroforest, potentially losing tiny scattered and vital populations important for connectivity

etc. I would like to see a little discussion around this in the main text, as well as explaining why they weren't surveyed directly here in the methods.

69. RESPONSE: Thanks for highlighting this. Actually, we did also do biodiversity surveys in forest fragments. We have now included these data in our manuscript and provide species richness of forest fragments and compare it to the other land-use types (see Table 16). Based on our newly added analysis of species composition (Figure S4), we show that species composition of forest fragments significantly differs from old-growth forest. Commonly, forest fragments are used for resource extraction such as for firewood and timber (Zaehringer et al., 2017). Thus, forest fragments are structurally as well as compositionally different to old-growth forest (Osen et al., 2021).

577: this answers my earlier question about how and where forest-derived vanilla systems are established.

579-580: please can you provide more information about what the fallows are like? Some photos in the supplementary information would be very informative indeed.

70. RESPONSE: Thanks! We now provide exemplary photos of each land-use type (see Figure S2). We also added more information on the characteristics of fallows in the methods (see Response 35).

Fig. S3: Exemplary photos of sampled land-use types: Old-growth forest, Forest fragment, Forest-derived vanilla agroforest, Herbaceous fallow, Woody fallow, Fallow-derived vanilla agroforest (from top left to bottom right).

588: so there is succession on the fallows if they are left – so does vanilla establishment promote or facilitate recovery to a forest-like state, or are the agroforests just older than the non-agroforest fallows because they get cleared more frequently, and therefore there is no direct facilitation effect? But an indirect facilitation effect (i.e. by providing a long-term land use not in conflict with forest cover, trees are allowed to grow naturally)? This is important for the discussion of restoration – I suggest adding something on this to the discussion.

71. RESPONSE: Thanks for this thought!

Yes, we found strong indication that the establishment of agroforests on fallow land increased the stand structure: within our research project, Osen et al. (2021) showed that basal area in fallow-derived agroforests (median basal area of 8 m²/ha) increased significantly (238% increase; p=0.011) compared to fallow land (median of 3m²/ha in woody fallows). However, the median basal area in fallow-derived agroforests was lower than in forest fragments (median basal area of 20 m²/ha) and far lower than in old-growth forest (median basal area of 41 m²/ha). These results indicate that fallow-derived agroforests promote or facilitate stand structure on fallow land, but the stand structure is lower compared to forests. Additionally, even though stand structure in fallow-derived agroforests might recover over time (Martin, Wurz, et al., 2021) the tree species richness and composition in fallow-derived agroforests remains significantly different compared to forest (Osen et al 2021).

In terms of management during the conversion of fallow land to agroforests, we observed that the tree regrowth in fallow land was partly maintained and used as support structures for the vanilla plants (Osewold et al., 2022). This offers opportunities to integrate naturally regrowing tree species into the newly established agroforests and represents an alternative to shifting cultivation.

To acknowledge the dynamics of land use around fallows and fallow-derived vanilla agroforest we added the following sentences in the discussion:

“Once they are established, vanilla agroforests are unlikely to be transformed into other land-use types due to their high profitability (Martin et al., 2022), and thus present a leverage point for breaking out of the shifting cultivation cycle that degrades much of the agricultural land in Madagascar (Martin et al., 2022). Vanilla agroforests offer long-lasting opportunities for biodiversity and tree stand structures to recover (Martin, Wurz, et al., 2021; Osen et al., 2021), allowing associated biodiversity and ecosystem services to increase (Martin et al., 2022) and species composition partially to be restored compared to fallow land (Rakotomalala et al., 2021). Tree regrowth in fallow-derived agroforests is particularly valuable to supplement and connect the few remaining forest fragments across the agricultural matrix (J. Zaehring et al., 2015). In contrast, fallow land under shifting cultivation experiences repetitive burning (Laney, 2002), which limits the establishment and growth of trees (Styger et al., 2007). “. (LINE 437)

596: so forest likely disturbed? How similar to the remaining forest fragments in the wider landscape in terms of structure?

71. *RESPONSE*: Thanks for raising this. We now added a more detailed description of forest fragments in the method section and describe how they differ from old-growth forest:

“Forest fragments are located inside the agricultural landscape and are remnants of once continuous forest which are frequently used for natural product extraction. Forest fragments have not been burned or clear cut in living memory, yet the ongoing resource extraction results in a much-simplified stand structure and fewer large trees compared to old-growth forest (Osen et al., 2021).“ (LINE 787)

602: please give sample plot area in ha to match the following sentence

72. *RESPONSE*: We now also list plot area in ha. (LINE 827)

622: “across the” not “in each of” – you didn’t do 30 x 10 = 300 plots!

RESPONSE: We corrected this. (LINE 847)

661: were any questions asked about other labour activities? Why did pollination vary?

73.RESPONSE: We added: See Response 14.

“Besides pollination labour input, we assessed time spent on plantation establishment, planting, weeding, pruning, plantation safeguarding, harvesting, preparing (fermenting, drying, sorting) and selling of vanilla (all not considered in this analysis).“ . (LINE 887-890)

We decided to use pollination labour input as only labour variable for the following two reasons. **Firstly**, pollination is by far the most labour-demanding part of vanilla production (Davis, 1983). Weeding, a commonly labour intensive task in crop production (Parish, 2012), is performed minimally for vanilla to avoid damaging of the vanilla roots (Nair, 2021; Odoux & Grisoni, 2010). The time spent on hand pollination of vanilla flowers is particularly critical, as no natural pollinators exist in Madagascar, and yields without pollination are extremely low (Westerkamp & Gottsberger, 2001; Wurz et al., 2021). Overall, hand pollination can make up to 40% of the production costs (Gregory et al., 1967) and trained people require one day to pollinate 1000-2000 flowers (Purseglove et al., 1981). Hence, we expected that the time invested into pollination is most decisive for vanilla yield. **Secondly**, pollination has a defined time frame because vanilla only flowers between October and December (Hänke et al., 2018). Thus, the hours of pollination labour input are easy to disentangle and define for the farmers, as family members are often directly involved or workers specifically hired. In contrast, other tasks such as weeding, pruning and planting often happen continuously and in parallel with other activities of farm households, and are thus harder for the farmer to depict in terms of working hours.

Variation in pollination labour input is driven by available household labour (see analysis in previous response 14).

666: as mentioned above, why not omit these plots? Please justify, perhaps with model results with and without the imputed values.

74.RESPONSE: We performed the analysis with and without imputed values – pollination labour input remains in both not significant (see results in previous response). We refrained from omitting these three plots as we would have lost valuable richness data leading to only 26 plots for trees and 27 plots for the remaining taxa. Please see our detailed response (response 8) to this point above, and how we now included better-predicted values of pollination labour based on its correlation with household size.

673: so vanilla support trees are quite short – are there trees above them in either agroforestry system? Photos may again help here.

75.RESPONSE: Thanks for this question. Indeed, there are often old and tall trees in the agroforestry systems, adding to the canopy cover. We added exemplary photos of all land-use types (Figure S2).

Yes, typically shade trees exist in vanilla agroforestry building an additional vegetation layer above the vanilla-support trees: The canopy height of these shade trees is higher in forest-

derived agroforests (mean = 14.5 m, SD = 7.3) compared to open-land-derived agroforests (mean = 6.3 m, SD = 4.6) (Martin, Wurz, et al., 2021)

682: does this forest cover map distinguish forest from agroforest? Do you think that would matter?

76.RESPONSE: Yes it does distinguish forests from agroforests according to Vieilledent et al. (2018) Agricultural land, such as tree plantations, is excluded by combining historical forest maps by Harper et al. (2007) with up-to-date forest cover change maps by Hansen et al. (2013). Thus, assessed forest cover was measured only inside historically assessed forest areas (at 30m resolution).

We believe that including agroforest in the forest cover map would mask the effect of “real” forest as tree-containing agroforests are widespread in the study region (e.g. vanilla, coffee, clover) (Hänke et al., 2018).

686: rephrase to “percentage woody cover”

77.RESPONSE: We corrected this.

691: why did you measure leaf damage? No hypothesis around leaf damage is outlined in this paper.

78.RESPONSE: We reflected upon our hypotheses around leaf damage and decided to omit it from the analysis, as literature on leaf loss/damage in vanilla is scarce and we decided to focus on variables known to affect yield and biodiversity. Also, leaf damage was quite limited on vanilla plants, corresponding to our impression of only limited insect pest damage of the crop in the region. Leaf damage is therefore no longer included in the updated manuscript.

740: so there are no trees in herbaceous fallows? Photos please!

79.RESPONSE: Yes, all herbaceous fallows did not have trees. We now provide a photo of all land-use types (see Figure S2)

741: so n = 48 plots in total were assessed including the forest sites, with n = 30 agroforests? or n = 60 but only n = 48 with tree assessments? Were two fallow-derived plots excluded from all analyses or just the trees?

80.RESPONSE: Now we have a total of 58 plots for trees because forest fragments were added to the analysis. In terms of vanilla agroforest, trees were assessed in 28 vanilla agroforests. The missing two fallow-derived vanilla agroforest plots for tree data were only excluded for the analysis of tree species richness with environmental/management variables and the tree-yield relationship analysis. For all other taxa, we included 30 vanilla agroforests in the analysis.

We adapted the sentences for clarity:

“**Trees.** We sampled trees on all land-use types except herbaceous fallows between September 2018 and January 2019 (Osen et al., 2021). Access was denied to two fallow-derived vanilla plantations, resulting in 58 plots assessed overall (including 28 vanilla agroforests).” (LINE 1000)

744: a local who? A local farmer, assistant, person, guide, botanist?

81.RESPONSE: We added now names of the tree experts:

“...with the help of a local tree expert (Chrysostome Bevaio) and a taxonomic expert (Patrice Antilahimena)..” (LINE 1003)

758: is there potential interannual seasonal effects not detected in the forest? Species accumulation/rarefaction curves could be illuminating here

82.RESPONSE: We exchanged the order of plot visits in the second year to minimize seasonal bias. For old-growth forest we assessed birds one time in August/September (2018) and December (2018).

Overall, the sampling season fell in the region’s dry season. We chose this season because it offered a rather constant climate throughout the sampling period.

760: what time of day were the point counts conducted – in the morning? What % of birds are endemic?

83.RESPONSE: We specify the time of data collection (“On 63 plots we started one point count around sunrise and one at least one hour after sunrise; on 7 plots this alteration was not possible due to logistical constraints.” (LINE 1032) and added the endemism level of birds found in this study.

768: i.e. moving logs, leaf litter?

84.RESPONSE: Individuals were searched in removable microhabitats such as under rocks, in leaf axils, tree barks, tree holes, leaf-litter, or deadwood. We added this information. (LINE 1051)

769: how were DNA samples taken? How did you identify species – are there field guides (if so please cite) or other online resources?

85.RESPONSE: We added the following info:

“We identified individuals based on morphological characteristics with the help of field guides (Glaw & Vences, 2007; Rakotoarison et al., 2017; Ratsavina et al., 2019)” (LINE 1054)

“To retrieve a DNA sample, we collected muscle or toe clips as tissue samples of individuals. Until release, we kept them in a ventilated bag to retain moisture, conserved in 90% of alcohol. We stored DNA samples at the Evolutionary Biology laboratory at TU Braunschweig. We also took photos of specimens that we did not identify to species level (ventral, back and flank view). Until release, we kept them in a ventilated bag to retain moisture. We released all specimens after completing the full-time-standardized search.” (LINE 1057-1061)

776: more info on trap design please – and interesting to know what % of butterfly captures were from nets vs manual netting please. Any problems with rain, or is there no rain in the sampled season? Any traps lost to wildlife or people?

86.*RESPONSE*: We used fish lines covered with vaseline creme to hang the bait traps. The Vaseline prevented ants to intrude on the bait traps. In addition, we avoided any contact of branches on the bait traps. We had one case of praying mantis and Gryllacrididae predating butterflies. The identification of the specimen was possible based on the wings which were left by the predator (wings were fully intact, only the thorax was eaten). Sampling was performed in the dry season of the year in which climatic conditions are rather stable.

We attached a foliated information sheet on each trap and held village meetings before trap installation – thus we did not have any trap losses due to people.

We caught 18 species with bait traps, 46 with time-standardized catch and 21 species were shared among the two techniques. In terms of individuals, 1578 individuals were caught by time-standardized catch vs. 1065 with bait traps.

We added the following information:

“We baited fruit traps with fermented banana and deployed the cylindrical nets for 24 hours. Prior to deployment, we fermented bananas for 48 hours in an air-tight container. On each plot, we installed a total of 8 fruit traps. We deployed four fruit traps at 16.6 m distance from the plot centre in the four main cardinal directions and the other four fruit traps at 20 m distance from the centre in the four intercardinal directions. We caught butterflies with a bait trap with a 20 cm Cone Opening (90 cm long) hanging 1.5 m above the ground. On plots without trees, we installed bait traps on a support stick (in rice paddy and herbaceous fallow).” (LINE 1068-1074)

“The net had a circular frame with a nylon mesh on a 1.5 m telescopic handle.” (LINE 1077)

787: please give full location name for butterfly samples

87.*RESPONSE*: Thanks for raising this. We now state:

“All identified specimens remain at the insect collection of the Dept. of Crop Sciences, section Agroecology , University of Göttingen, Germany.” (LINE 1083)

798: who did the ID, and how? Please give field guide references/online info.

88.*RESPONSE*:

“Identification was done by Anjaharinony Rakotomalala, using available identification keys (Bolton & Fisher, 2014; Fisher & Smith, 2008; Rakotonirina & Fisher, 2014). Cross-checking of the identification of the species was done with expert consultation by Jean Claude Rakotonirina (species of Leptogenys), Nicole Rasoamanana (species of Camponotus), and Manoa Ramamonjisoa (species of Tetramorium).” (LINE 1100)

800: address/city?

89.*RESPONSE*: We added „Antananarivo, Madagascar. “ (LINE 1102)

803: please see my earlier notes on the multidiversity idea, which could be addressed here. I find the explanation of the approach quite unclear. What are rigorosity and mildness and why do they matter? From looking briefly at the cited papers, they discuss a measure of ecosystem functioning, which is not what you are trying to capture here. I do not find that the multidiversity measure provides any additional insight or information useable to the reader, and would prefer more analysis that looks at species composition, as mentioned earlier in my review, as this links more closely to thinking about conservation. Species richness hides a multitude of change. I don't see how the multidiversity measure concept can transfer to

management, policy or conservation measures, nor how it informs e.g. ecosystem functioning or services as the cited papers do.

90.RESPONSE: We replaced the multidiversity measure with mean normalized richness (see Response 18). This metric is a common method to aggregate information on species richness across multiple taxa (Collen et al., 2014; Tittensor et al., 2010; Van Elsas et al., 2012). We are confident that the adoption of this index increases the clarity, understanding, and interpretation of our results, as compared to the more complicated multidiversity index that we used beforehand. In addition, we added figures of species composition (Figure S4).

814: why not use total richness, why penalise and reduce the impact of species losses on your metric?

91.RESPONSE: Please note that our new measure of mean normalized species richness across taxa includes all species.

818: I don't understand what this figure helps you understand about the biodiversity response in the context of conservation, or management.

92.RESPONSE: Non-applicable anymore, as we do not use multidiversity any longer.

820: please explain what glmmTMB models do differently compared to linear mixed effect models, and why you used them here.

93.RESPONSE: In general, we expect glmmTMB's advantages over lme4 to be (1) greater flexibility (zero-inflation etc.); (2) greater speed for GLMMs, especially those with large number of "top-level" parameters (fixed effects plus random effects variance-covariance parameters) (Brooks et al., 2017). Furthermore, we could tailor the model with different families (compois, nbinominal, poisson) to achieve optimal residual distribution.

836: please explain this in the caption of Table S5. So you have sequentially reduced the model by one variable each time, or taken out each variable in turn to test change in variance?

94. RESPONSE: We used likelihood ratio tests using maximum likelihood estimation (Zuur et al. 2009) to assess the statistical significance of individual variables. Specifically, we compared the full model versus a reduced model without the individual variable (single term deletion). We added this information to our method section (LINE 1139).

Are these chi-square tests comparing the model with only this variable, or with the full model minus only this variable?

95. RESPONSE: Yes, the latter reflects our methodology.

We added now "full model vs. full model minus only one specific variable" to make it clear.

839: when did you use which model type and why?

96. RESPONSE: We added now:

"We used glmmTMB model for all analysis with count/discrete data as response and we used linear mixed effect models for models with continuous data as the response (e.g. mean normalized richness)." (LINE 1148)

844: please give full name for “compois”

97. *RESPONSE*: We added now:

“compois (Conway-Maxwell-Poisson distribution)” (LINE 1157)

849: i.e. prediction? If so, you can provide effect sizes (change in y per unit change in x) for significant results, on the relevant figure in the main text.

98. *RESPONSE*: We provide now estimates in the result section.

851: you have not actually explained the fitting of this mode – only the fitting of yield or species richness response to predictors. As mentioned in my comments on the results section, directly assessing yields vs richness doesn’t quite make sense to me.

I am surprised no attempt made to assess species composition using NMDS or RDA.

99. *RESPONSE*: We agree that correlation does not need to mean causation. We see that biodiversity is likely to be driven to a large extent by underlying yield-related management variables, which we present in-depth in the second analysis of biodiversity depending on four management and six environmental variables. Please see response 55 for more extensive details.

We now present an analysis of species composition (Figure S4) to detect changes in species identity. Furthermore, we decided to use a glmm model to identify the drivers of yield and biodiversity because it allowed us to consider the study design (account for the village as a random effect). Also, it computed robust results for the number of replicates (30) available and the included number of environmental/management variables (10).

To give more detail, we provide now figures with the individual relationships between yield/biodiversity with each of the environmental/management variables (Figure S5, S7).

Supp info

Table S2: no need to highlight the significance of the intercept

100. *RESPONSE*: We changed it accordingly.

Reset page numbers to run through whole SI if using (i.e. across the section breaks) and perhaps provide a table of contents at the start?

101. *RESPONSE*: We added now a table of content for the supplementary information.

Fig S3: VFAL not VFST?

102. *RESPONSE*: Corrected

Fig S4: there are no circles – check caption

103. *RESPONSE*: Corrected

Little correlation between age and vine length here, so what affects vine length?

104. *RESPONSE*: Yes, vine length increases with age (see Response 27 for graph and analysis) but according to our results, age did not have an effect on vanilla yield but vine length had a positive. Thus, we can assume, that vine length is not only driven by age but other undetected variables (such as possibly soil characteristics, climatic conditions or vine management) – our study cannot determine all drivers of vine length.

Fig S5: “only marginally associated” here really means “a significant effect with small effect size” – move this to main text, or something similar, as per comments on results, above.

105. RESPONSE: We moved Figure S5 to the main text (now Figure 3).

Fig S7: not a major concern as soil doesn't really get discussed, but why isn't soil C included in the PCA?

106. RESPONSE: Thanks for this important remark! C total was highly correlated with the exchange capacity of Calcium. Calcium had the highest factor loading in PC1 and thus was kept for the final projection.

Table S13: VFST should be VFOR?

Table S14: VFAL not VFLW?

107. RESPONSE: Corrected. Thanks for spotting this detail!

Reviewer #3 (Remarks to the Author):

I have been invited to review the manuscript “”, specifically to determine if the biodiversity surveys, especially birds, have been conducted appropriately, and invited to take a broader look at the paper. Given this paper is far from my expertise, I will only comment on the biodiversity surveys.

108. RESPONSE: We are happy to receive your feedback and we are confident we improved our method section based on your constructive comments.

Overall, I think it looks fine but I have several comments requesting a bit more information. First, I think the authors could add tables in SI with species sampled and if they are endemic or not (currently authors cite external source, which may disappear / evolve at some point) and it would be good to have some stand-alone information for reproducibility and transparency.

109. RESPONSE: We uploaded a species-by-site matrix including information on endemism for all taxa on OSF (Open Science Framework).

https://osf.io/j54fx/?view_only=1bd699c5cda64023963e058254a33eec

Second, there is often some delay between first and last sampling (eg August to December) and it is unclear whether this can have a strong impact. I suggest the authors state a bit better how this way conducted (especially it's important to know if the order was random, treatment by treatment, or village by village). For instance, if all sampling in the National Park has been done in December and if there is a strong seasonal effect, this can have important effect on results. A possible way to address that is to analyse the effect of seasonality on biodiversity metrics used in the study (and possibly to control for this in the analyses).

110. RESPONSE: Thanks for highlighting this important issue. We accounted for seasonality in the data collection. Firstly, for all land uses within villages, by visiting villages in subsequent order, we sampled each land-use type each week, avoiding a bias between land-use type and season in which it was assessed. Due to logistical constraints, we sampled 5 old-growth forest plots within one visit to the National Park. However, by visiting old-growth forest plots at the beginning and the end of the data collection period, we also managed to minimize bias. For data collection spanning two years (e.g. amphibians, reptiles and birds), we reversed the order of villages visited in the second year to further avoid a seasonal bias.

Given these precautions, we are confident that controlling for the day of data collection in the analyses is unnecessary.

Trees: The sampling seems appropriate to me. I would just ask the authors to add in Supplementary Information a list of species, specifying which species are considered as endemic so that readers can find all information (and will be able to access it in the future as well)

111. RESPONSE: We uploaded a species-by-site matrix including information on endemism for all taxa on OSF.

Herbaceous plants: same comment. Can you also detail how plant phenology can affect the sampling in the 4 months study period and if how was conducted the sampling (village after village / random / first all forests and then all fallows...)

112. RESPONSE: We accounted for the possible seasonal variation of the plant phenology when we planned the data collection. We visited one village after another and in each village, we collected data in each land-use type except for the old-growth forests. Hence, the observations for each land-use type cover the study period along with the possible phenology variations. We added this information to the method section (see LINE 1012).

Birds: I think the sampling duration and number of observers are appropriate but that some information is lacking. Specifically, while this is less of an issue with plants, the expertise of observers is key for birds. So I suggest the authors add some information on the people involved in sampling (are they all birders, how well do they know birds from the region and especially their calls and songs, how many different people were involved, can there be a bias if some people went only in some villages and other only in other villages...). I am also concerned with the time window (from August to December) which can affect birds detected. I suggest the authors add in SI some analyses looking at the variation in the bird metrics used (eg richness) over time regardless of treatment. If there is a strong pattern, it should be accounted for in the analyses. Similarly, to my questions on plants, how was this timing distributed across sampling units (eg villages one by one / random / treatment 1 then treatment 2...)?

113. RESPONSE: Thanks for raising these questions. We report now details on the observers, order of villages, and order of plot visitations in the methods.

In short, the point counts were done by experienced ornithologists with good knowledge of the Malagasy avifauna. Additionally, the same observers conducted point counts across various land-use types, avoiding a bias in observer by land-use type. To avoid seasonal biases, we optimized the order of plot visitations. Firstly, for all land-uses within villages, by visiting villages in subsequent order, we sampled each land-use type each week, leading to no season-land-use type bias. Due to logistical constraints, we sampled 5 old-growth forest plots within one visit to the National Park. However, by visiting old-growth forest plots at the beginning and end of the data collection period, we also managed to minimized bias. Additionally, we reversed the order of villages visited in the second year to further avoid a seasonal bias. Given these various precautions, we are confident that the bird data is robust and not subject to seasonal bias. Additional information on the bird sampling can be found in (Martin, et al., 2021).

Reptiles and amphibians: Same comments, the sampling method seems appropriate but I would appreciate some information on the randomisation of sampling order and the expertise of fieldworkers.

114. *RESPONSE*: Thanks to highlight this concern. The observation of amphibians and reptiles was done under a supervision of a specialist and led by experienced observers. To identify species level, we referred to, mostly, field guide (Glaw & Vences, 2007) and if in doubt, we collected tissue from specimens for genetic molecular analysis (see Fulgence et al. (2021) for more details). Collected tissue stored at the lab of the University of Braunschweig. The same observers conducted the observation throughout all land-use types. We were doing the same randomization of sampling sites and plots as for birds (see response 113).

Butterflies: Perfect

115. *RESPONSE*: Thanks so much!!

Ants: I don't know anything about ants but it sounds fine

116. *RESPONSE*: Thanks for your honesty and the rest of the review!

References

- Andriamparany, J. N., Hänke, H., & Schlecht, E. (2021). Food security and food quality among vanilla farmers in Madagascar: the role of contract farming and livestock keeping. *Food Security, 13*(4), 981–1012. <https://doi.org/10.1007/s12571-021-01153-z>
- Beer, T. (2013). Beaufort wind scale. In *Encyclopedia of Natural Hazards* (pp. 42–45). Springer. https://scholar.google.com/scholar?hl=de&as_sdt=0%2C5&q=Beer+2013+beaufort&btnG=
- Béliveau, A., Davidson, R., Lucotte, M., Do, L. O., Lopes, C., Paquet, S., & Vasseur, C. (2014). *Early effects of slash-and-burn cultivation on soil physicochemical properties of small-scale farms in the Tapajós region, Brazilian Amazon*. <https://doi.org/10.1017/S0021859613000968>
- Bolton, B., & Fisher, B. L. (2014). The Madagascan endemic myrmicine ants related to *Eutetramorium* (Hymenoptera: Formicidae): taxonomy of the genera *Eutetramorium* Emery, *Malagidris* nom. n., *Myrmisaraka* gen. n., *Royidris* gen. n., and *Vitsika* gen. n. *Zootaxa, 3791*(1), 1–99–1–99. <https://doi.org/10.11646/ZOOTAXA.3791.1.1>
- Broennimann, O., Thuiller, W., Hughes, G., Midgley, G. F., Alkemade, J. M. R., & Guisan, A. (2006). Do geographic distribution, niche property and life form explain plants' vulnerability to global change? *Global Change Biology, 12*(6), 1079–1093. <https://doi.org/10.1111/J.1365-2486.2006.01157.X>
- Brooks, M. E., Kristensen, K., van Benthem, K. J., Magnusson, A., Berg, C. W., Nielsen, A., Skaug, H. J., Maechler, M., & Bolker, B. M. (2017). *{glmmTMB} Balances Speed and Flexibility Among Packages for Zero-inflated Generalized Linear Mixed Modeling*. <https://journal.r-project.org/archive/2017/RJ-2017-066/index.html>
- Bruelle, G., Naudin, K., Scopel, E., Domas, R., Rabearisoa, L., & Tottonell, P. (2015). Short-to mid-term impact of conservation agriculture on yield variability of upland rice: Evidence from farmer's fields in Madagascar. *Experimental Agriculture, 51*(1), 66–84. <https://doi.org/10.1017/S0014479714000155>
- Chaudhary, A., Verones, F., De Baan, L., & Hellweg, S. (2015). Quantifying Land Use Impacts on Biodiversity: Combining Species-Area Models and Vulnerability Indicators. *Environmental Science and Technology, 49*(16), 9987–9995. https://doi.org/10.1021/ACS.EST.5B02507/SUPPL_FILE/ES5B02507_SI_004.XLSX
- Chazdon, R. L. (2008). Beyond deforestation: Restoring forests and ecosystem services on degraded lands. *Science, 320*(5882), 1458–1460. <https://doi.org/10.1126/SCIENCE.1155365/ASSET/40F39960-4BD7-4824-BD46->

A045D294E2BA/ASSETS/GRAPHIC/320_1458_F2.JPEG

- Chazdon, R. L., Brancalion, P. H. S., Laestadius, L., Bennett-Curry, A., Buckingham, K., Kumar, C., Moll-Rocek, J., Vieira, I. C. G., & Wilson, S. J. (2016). When is a forest a forest? Forest concepts and definitions in the era of forest and landscape restoration. *Ambio*, 45(5), 538–550. <https://doi.org/10.1007/S13280-016-0772-Y/FIGURES/3>
- Collen, B., Whitton, F., Dyer, E. E., Baillie, J. E. M., Cumberlidge, N., Darwall, W. R. T., Pollock, C., Richman, N. I., Soulsby, A. M., & Böhm, M. (2014). Global patterns of freshwater species diversity, threat and endemism. *Global Ecology and Biogeography*, 23(1), 40–51. <https://doi.org/10.1111/GEB.12096/SUPPINFO>
- Curtis, P. G., Slay, C. M., Harris, N. L., Tyukavina, A., & Hansen, M. C. (2018). Classifying drivers of global forest loss. *Science*, 361(6407), 1108–1111.
- Davis, E. W. (1983). Experiences with growing vanilla (*Vanilla planifolia*). *Acta Horticulturae*, 132, 23–30. <https://doi.org/10.17660/ActaHortic.1983.132.2>
- Deininger, K., Carletto, C., Savastano, S., & Muwonge, J. (2012). Can diaries help in improving agricultural production statistics? Evidence from Uganda. *Journal of Development Economics*, 98(1), 42–50. <https://doi.org/10.1016/j.jdevec.2011.05.007>
- Díaz-Bautista, M., Antonieta, M., Quintero, S., Espinoza-Pérez, J., Jair Barrales-Cureño, H., Herrera-Cabrera, B. E., Antonieta Sandoval-Quintero, M., Juárez-Bernabe, Y., & Reyes, C. (2019). Floristic biodiversity in *Vanilla planifolia* agroecosystems in the Totonacapan region of Mexico. *Biocell*, 1, 440–452. <https://www.researchgate.net/publication/337290391>
- Egeskog, A., Barretto, A., Berndes, G., Freitas, F., Holmén, M., Sparovek, G., & Torén, J. (2016). Actions and opinions of Brazilian farmers who shift to sugarcane an interview-based assessment with discussion of implications for land-use change. *Land Use Policy*, 57, 594–604. <https://doi.org/10.1016/J.LANDUSEPOL.2016.06.022>
- FAO. (2020). *FAOSTAT Vanilla Production quantity*. <http://www.fao.org/faostat/en/#data/QC>
- Fischer, J., Riechers, M., Loos, J., Martin-Lopez, B., & Temperton, V. M. (2021). Making the UN Decade on Ecosystem Restoration a Social-Ecological Endeavour. *Trends in Ecology & Evolution*, 36(1), 20–28. <https://doi.org/10.1016/J.TREE.2020.08.018>
- Fisher, B. L., & Smith, M. A. (2008). A Revision of Malagasy Species of *Anochetus* Mayr and *Odontomachus* Latreille (Hymenoptera: Formicidae). *PLOS ONE*, 3(5), e1787. <https://doi.org/10.1371/JOURNAL.PONE.0001787>
- Fulgence, T. R., Martin, D. A., Randriamanantena, R., Botra, R., Befidimanana, E., Osen, K., Wurz, A., Kreft, H., Andrianarimisa, A., & Ratsavina, F. M. (2021). Differential responses of amphibians and reptiles to land-use change in the biodiversity hotspot of north-eastern Madagascar. *Animal Conservation*. <https://doi.org/10.1111/acv.12760>
- Fulgence, Thio Rosin, Andreas Martin, D., Botra, R., Befidimanana, E., Osen, K., Wurz, A., Kreft, H., & Mihaja Ratsavina, F. (2021). Differential responses of amphibians and reptiles to land-use change in the biodiversity hotspot of north-eastern Madagascar. *Centre of the SAVA Region (CURSA)*, 2021.03.18.435920. <https://doi.org/10.1101/2021.03.18.435920>
- Gann, G. D., McDonald, T., Walder, B., Aronson, J., Nelson, C. R., Jonson, J., Hallett, J. G., Eisenberg, C., Guariguata, M. R., Liu, J., Hua, F., Echeverría, C., Gonzales, E., Shaw, N., Decler, K., & Dixon, K. W. (2019). International principles and standards for the practice of ecological restoration. Second edition. *Restoration Ecology*, 27(S1), S1–S46. <https://doi.org/10.1111/REC.13035>
- Geiger, F., Bengtsson, J., Berendse, F., Weisser, W. W., Emmerson, M., Morales, M. B., Ceryngier, P., Liira, J., Tschardtke, T., Winqvist, C., Eggers, S., Bommarco, R., Pärt, T., Bretagnolle, V., Plantegenest, M., Clement, L. W., Dennis, C., Palmer, C., Oñate, J. J., ... Inchausti, P. (2010). Persistent negative effects of pesticides on biodiversity and biological control potential on European farmland. *Basic and Applied Ecology*, 11(2),

- 97–105. <https://doi.org/10.1016/j.baae.2009.12.001>
- Glaw, F., & Vences, M. (2007). A field guide to the amphibians and reptiles of Madagascar. *English*, 496.
- Gregory, L. E., Gaskins, M. H., & Colberg, C. (1967). Parthenocarpic Pod Development by *Vanilla planifolia* Andrews Induced with Growth-Regulating Chemicals. *Economic Botany*, 21(4), 351–357.
- Hänke, H. (2019). *Fairtrade's Living Income Reference Price for Vanilla from Uganda and Madagascar*. <https://www.fairtrade.net/library/living-income-reference-prices-for-vanilla>
- Hänke, H., Barkmann, J., Blum, L., Franke, Y., Martin, D. A., Niens, J., Osen, K., Uruena, V., Witherspoon, S. A., & Wurz, A. (2018). Socio-economic, land use and value chain perspectives on vanilla farming in the SAVA Region (north-eastern Madagascar): The Diversity Turn Baseline Study (DTBS). *Diskussionsbeitrag No. 1806, July 2019*.
- Hansen, M. C., Potapov, P. V., Moore, R., Hancher, M., Turubanova, S. A., Tyukavina, A., Thau, D., Stehman, S. V., Goetz, S. J., Loveland, T. R., Kommareddy, A., Egorov, A., Chini, L., Justice, C. O., & Townshend, J. R. G. (2013). High-resolution global maps of 21st-century forest cover change. *Science*, 342(6160), 850–853. https://doi.org/10.1126/SCIENCE.1244693/SUPPL_FILE/HANSEN.SM.PDF
- Harper, G. J., Steininger, M. K., Tucker, C. J., Juhn, D., & Hawkins, F. (2007). Fifty years of deforestation and forest fragmentation in Madagascar. *Environmental Conservation*, 34(4), 325–333. <https://doi.org/10.1017/S0376892907004262>
- Havkin-Frenkel, D., & Belanger, F. C. (2010). *Handbook of Vanilla Science and Technology* (D. Havkin-Frenkel & F. C. Belanger (eds.)). Wiley-Blackwell. <http://doi.wiley.com/10.1002/9781444329353>
- Hernández Hernández, Juan Lubinsky, P. (2010). Vanilla Production in Mexico in *Vanilla*. CRC Press.
- Hölscher, D., Ludwig, B., Möller, R. F., & Fölster, H. (1997). Dynamic of soil chemical parameters in shifting agriculture in the Eastern Amazon. *Agriculture, Ecosystems & Environment*, 66(2), 153–163. [https://doi.org/10.1016/S0167-8809\(97\)00077-7](https://doi.org/10.1016/S0167-8809(97)00077-7)
- Inoue, Y., Qi, J., Olioso, A., Kiyono, Y., Horie, T., Asai, H., Saito, K., Ochiai, Y., Shiraiwa, T., & Douangsavanh, L. (2007). Traceability of slash-and-burn land-use history using optical satellite sensor imagery: a basis for chronosequential assessment of ecosystem carbon stock in Laos. [Http://Dx.Doi.Org/10.1080/01431160701656323](http://Dx.Doi.Org/10.1080/01431160701656323), 28(24), 5641–5647. <https://doi.org/10.1080/01431160701656323>
- Kremen, C. (2015). Reframing the land-sparing/land-sharing debate for biodiversity conservation. *Annals of the New York Academy of Sciences*, 1355(1), 52–76. <https://doi.org/10.1111/NYAS.12845>
- Kunz, S., Hänke, H., & Schlecht, E. (2020). Income diversification through animal husbandry for smallholder vanilla farmers in Madagascar. *Journal of Agriculture and Rural Development in the Tropics and Subtropics (JARTS)*, 121(1), 63–75. <https://doi.org/10.17170/KOBRA-202004061143>
- Laney, R. M. (2002). Disaggregating Induced Intensification for Land-Change Analysis: A Case Study from Madagascar. *Annals of the Association of American Geographers*, 92(4), 702–726. <https://doi.org/10.1111/1467-8306.00312>
- Martin, D. A., Andriafanomezantsoa, R., Dröge, S., Osen, K., Rakotomalala, E., Wurz, A., Andrianarimisa, A., & Kreft, H. (2021). Bird diversity and endemism along a land-use gradient in Madagascar: The conservation value of vanilla agroforests. *Biotropica*, 53(1), 179–190. <https://doi.org/10.1111/btp.12859>
- Martin, D. A., Andrianisaina, F., Fulgence, T. R., Osen, K., Rakotomalala, A. A. N. A., Raveloaritiana, E., Sozafy, M. R., Wurz, A., Andriafanomezantsoa, R., Andriamaniraka, H., Andrianarimisa, A., Barkmann, J., Dröge, S., Grass, I., Guerrero-Ramirez, N., Hänke, H., Hölscher, D., Rakouth, B., Ranarijaona, H. L. T., ... Kreft, H.

- (2022). Land-use trajectories for sustainable land system transformations: Identifying leverage points in a global biodiversity hotspot. *Proceedings of the National Academy of Sciences*, 119(7), e2107747119. <https://doi.org/10.1073/PNAS.2107747119>
- Martin, D. A., Osen, K., Grass, I., Hölscher, D., Tscharntke, T., Wurz, A., & Kreft, H. (2020). Land-use history determines ecosystem services and conservation value in tropical agroforestry. *Conservation Letters*.
- Martin, D. A., Wurz, A., Osen, K., Grass, I., Hölscher, D., Rabemanantsoa, T., Tscharntke, T., & Kreft, H. (2021). Shade-Tree Rehabilitation in Vanilla Agroforests is Yield Neutral and May Translate into Landscape-Scale Canopy Cover Gains. *Ecosystems*, 24(5), 1253–1267. <https://doi.org/10.1007/s10021-020-00586-5>
- Nair, K. P. (2021). Vanilla. *Minor Spices and Condiments*, 193–226. https://doi.org/10.1007/978-3-030-82246-0_19
- Odoux, E., & Grisoni, M. (2010). *Medicinal and Aromatic Plants - Industrial Profiles: Vanilla*.
- Osen, K., Soazafy, M. R., Martin, D. A., Wurz, A., März, A., Ranarijaona, H. L. T., & Hölscher, D. (2021). Land-use history determines stand structure and tree diversity in vanilla agroforests of northeastern Madagascar. *Applied Vegetation Science*, 24(1), e12563. <https://onlinelibrary.wiley.com/doi/10.1111/avsc.12563>
- Osewold, J., Korol, Y., Osen, K., Soazafy, M. R., Rabemanantsoa, T., Martin, D. A., Wurz, A., Hölscher, D., Martin Biodiversity, D. A., & Wurz Agroecology, A. (2022). Support trees in vanilla agroforests of Madagascar: diversity, composition and origin. *Agroforestry Systems 2022*, 1–14. <https://doi.org/10.1007/S10457-022-00733-Y>
- Parish, S. (2012). A Review of Non-Chemical Weed Control Techniques. <Http://Dx.Doi.Org/10.1080/01448765.1990.9754540>, 7(2), 177–1137. <https://doi.org/10.1080/01448765.1990.9754540>
- Purseglove, J. W., Brown, E. G., Green, C. I., & Robbins, S. R. J. (1981). *Spices (Spices, Va)*. Longman, London.
- Rakotoarison, A., Scherz, M., Glaw, F., & ... J. K. (2017). Describing the smaller majority: Integrative taxonomy reveals twenty-six new species of tiny microhylid frogs (genus *Stumpffia*) from Madagascar. *Vertebrate Zoology*, 67(3), 271–398. <https://www.researchgate.net/publication/320923469>
- Rakotomalala, A. A. N. A., Wurz, A., Grass, I., Martin, D. A., Osen, K., Schwab, D., Soazafy, M. R., Tscharntke, T., & Raveloson Ravaomanarivo, L. H. (2021). Tropical land use drives endemic versus exotic ant communities in a global biodiversity hotspot. *Biodiversity and Conservation*, 30(14), 4417–4434. <https://doi.org/10.1007/S10531-021-02314-4/FIGURES/5>
- Rakotonirina, J. C., & Fisher, B. L. (2014). Revision of the Malagasy ponerine ants of the genus *Leptogenys* Roger (Hymenoptera: Formicidae) . *Zootaxa*, 3836(1), 1–163–1–163. <https://doi.org/10.11646/ZOOTAXA.3836.1.1>
- Ratsoavina, F. M., Raselimanana, A. P., Scherz, M. D., Rakotoarison, A., Razafindraibe, J. H., Glaw, F., & Vences, M. (2019). Finaritra! A splendid new leaf-tailed gecko (*Uroplatus*) species from Marojejy National Park in north-eastern Madagascar. *Zootaxa*, 4545(4), 563–577. <https://doi.org/10.11646/ZOOTAXA.4545.4.7>
- Santos, P. Z. F., Crouzeilles, R., & Sansevero, J. B. B. (2019). Can agroforestry systems enhance biodiversity and ecosystem service provision in agricultural landscapes? A meta-analysis for the Brazilian Atlantic Forest. *Forest Ecology and Management*, 433, 140–145. <https://doi.org/10.1016/j.foreco.2018.10.064>
- Schroth, G., Izac, A. M. N., Vasconcelos, H. L., Gascon, C., da Fonseca, G. A., & Harvey, C. A. (2004). *Agroforestry and Biodiversity Conservation in Tropical Landscapes* ((Eds.) (ed.)). Island Press.
- Schwab, D., Wurz, A., Osen, K., Grass, I., & Tscharntke, T. (2020). Decreasing predation

- rates and shifting predator compositions along a land-use gradient in Madagascar's vanilla landscapes. *Journal of Applied Ecology*.
- Styger, E., Rakotondramasy, H. M., Pfeffer, M. J., Fernandes, E. C. M., & Bates, D. M. (2007). Influence of slash-and-burn farming practices on fallow succession and land degradation in the rainforest region of Madagascar. *Agriculture, Ecosystems and Environment*, 119(3–4), 257–269.
- Tamburini, G., Bommarco, R., Wanger, T. C., Kremen, C., van der Heijden, M. G. A., Liebman, M., & Hallin, S. (2020). Agricultural diversification promotes multiple ecosystem services without compromising yield. *Science Advances*, 6(45). https://doi.org/10.1126/SCIADV.ABA1715/SUPPL_FILE/ABA1715_SM.PDF
- Tews, J., Brose, U., Grimm, V., Tielbörger, K., Wichmann, M. C., Schwager, M., & Jeltsch, F. (2004). Animal species diversity driven by habitat heterogeneity/diversity: The importance of keystone structures. In *Journal of Biogeography* (Vol. 31, Issue 1, pp. 79–92). Blackwell Publishing Ltd. <https://doi.org/10.1046/j.0305-0270.2003.00994.x>
- Tittensor, D. P., Mora, C., Jetz, W., Lotze, H. K., Ricard, D., Vanden Berghe, E., & Worm, B. (2010). Global patterns and predictors of marine biodiversity across taxa. *Nature*, 466. <https://doi.org/10.1038/nature09329>
- Tschora, H., & Cherubini, F. (2020). Co-benefits and trade-offs of agroforestry for climate change mitigation and other sustainability goals in West Africa. *Global Ecology and Conservation*, 22. <https://doi.org/10.1016/j.gecco.2020.e00919>
- United Nations. (2020). *About the UN Decade | UN Decade on Restoration*. <https://www.decadeonrestoration.org/about-un-decade>
- Van Elsas, J. D., Chiurazzi, M., Mallon, C. A., Elhottova, D., Krištufek, V., & Salles, J. F. (2012). Microbial diversity determines the invasion of soil by a bacterial pathogen. *Proceedings of the National Academy of Sciences of the United States of America*, 109(4), 1159–1164. <https://doi.org/10.1073/PNAS.1109326109/-/DCSUPPLEMENTAL>
- Vieilledent, G., Grinand, C., Rakotomalala, F. A., Ranaivosoa, R., Rakotoarijaona, J. R., Allnutt, T. F., & Achard, F. (2018). Combining global tree cover loss data with historical national forest cover maps to look at six decades of deforestation and forest fragmentation in Madagascar. *Biological Conservation*, 222(June), 189–197.
- Westerkamp, C., & Gottsberger, G. (2001). Pollinator diversity is mandatory for crop diversity. *Acta Horticulturae*. <https://doi.org/10.17660/ActaHortic.2001.561.47>
- Wiseman, V., Conteh, L., & Matovu, F. (2005). Using diaries to collect data in resource-poor settings: questions on design and implementation. *Health Policy and Planning*, 20(6), 394–404. <https://doi.org/10.1093/heapol/czi042>
- Wurz, A., Grass, I., & Tschardt, T. (2021). Hand pollination of global crops – A systematic review. *Basic and Applied Ecology*, 56, 299–321. <https://doi.org/10.1016/j.baae.2021.08.008>
- Zaehring, J., Eckert, S., Messerli, P., Zaehring, J. G., Eckert, S., & Messerli, P. (2015). Revealing Regional Deforestation Dynamics in North-Eastern Madagascar—Insights from Multi-Temporal Land Cover Change Analysis. *Land*, 4(2), 454–474. <https://doi.org/10.3390/land4020454>
- Zaehring, J. G., Schwilch, G., Andriamihaja, O. R., Ramamonjisoa, B., & Messerli, P. (2017). Remote sensing combined with social-ecological data: The importance of diverse land uses for ecosystem service provision in north-eastern Madagascar. *Ecosystem Services*, 25, 140–152.
- Zuur, A. F., Ieno, E. N., Walker, N. J., Saveliev, A. A., & Smith, G. M. (2009). *Mixed Effects Modelling for Nested Data*. 101–142. https://doi.org/10.1007/978-0-387-87458-6_5

Reviewers' Comments:

Reviewer #2:

Remarks to the Author:

Line 173: Yields cannot decrease species richness – there is an association or a relationship between yields and species richness mediated by other factors (management, land-use history), so please rephrase to reflect this. I believe I mentioned this point in my first review.

Line 181: did you mean to also include the model results for the negative reptile relationship shown? "or negative (fallow-derived) relationship with endemic reptiles"

I'm delighted to see the improved presentation of the statistical results shown in Figure 2 (i.e. the additional in-line estimates and supplementary tables), the new approach to capturing species richness across all taxa (instead of multidiversity) and the species accumulation curves and NMDS plots, as well as the additional work on endemism. Although you explain that the study focus is on within-plot yield-richness responses, adding these further analyses allows understanding of what's going on at broader spatial scales, and provides much greater depth to your study. The clear patterns in species composition in Figure S4, and the rarefaction curves, make it easier to understand the thoroughness of your sampling and robustness of your work and findings, but also add qualitative information about 'what biodiversity' which is critical. I would be pleased to see the species composition results in the main text, but I am aware there are restrictions on the number of figures, and leave it to the authors/editor to decide whether this is worthwhile.

I agree that a structural equation model approach is not necessarily, but at line 173 would like to at least push for a change to the phrasing and interpretation of these modelled relationships to expressly note that the relationship is mediated by management and land use history, and is not causal.

Line 935: you could add a couple of words here to explain the 75% canopy cover threshold used to define forest, following the text in response 328

Thank you for working to clarify the use of the terms 'restoration' and 'degradation'.

Great to see the forest fragments now included separately from the old growth forest as reference (non-productive) systems, as you had the data already good to make it work for you!

Great to see plots of vanilla yield responses! The explanation of management-related variables, yields and richness is much improved.

Please check these lines – there is some repetition and contradiction (I don't think anything kept in 90% alcohol could then be released afterwards)

"To retrieve a DNA sample, we collected muscle or toe clips as tissue samples of individuals. Until release, we kept them in a ventilated bag to retain moisture, conserved in 90% of alcohol. We stored DNA samples at the Evolutionary Biology laboratory at TU Braunschweig. We also took photos of specimens that we did not identify to species level (ventral, back and flank view). Until release, we kept them in a ventilated bag to retain moisture. We released all specimens after completing the full-time-standardized search." (LINE 1057-1061)

Kindly include this very useful bit of methodological detail in the main text "fishing line covered with Vaseline was used to hang the bait traps, to prevent ant intrusion" – this will help future researchers lose valuable samples.

Thank you for addressing all my points in detail. The addition of this (perhaps laborious-seeming) detail improves interpretation and robustness greatly, and also aids the use of the work to inform future studies. Congratulations on an important and detailed study!

Reviewer #3:

Remarks to the Author:

I have read the authors response to my comments on biodiversity sampling and the corresponding text changes. I am very satisfied with the replies and text modifications. I think the authors made a great job in adding more information in the methods and I appreciate their effort to publish online data with more information. Thank you!

REVIEWERS' COMMENTS

Reviewer #2 (Remarks to the Author):

Line 173: Yields cannot decrease species richness – there is an association or a relationship between yields and species richness mediated by other factors (management, land-use history), so please rephrase to reflect this. I believe I mentioned this point in my first review.

1. RESPONSE: Thanks for this comment! We changed the wording from “decreased” to “associated with” (LINE 110) and also added the following sentence “Notably, the analysed relationship of yield with biodiversity is likely mediated by underlying variables such as management and land-use history. (LINE 112). In addition, we start the paragraph on the “Effects of environmental and management variables on biodiversity” with: “To understand the effect of underlying yield-related management variables on biodiversity, we analyzed the relationship of species richness with four management and six environmental variables.” (LINE 180)

Line 181: did you mean to also include the model results for the negative reptile relationship shown? “or negative (fallow-derived) relationship with endemic reptiles”

2. RESPONSE: We now include also the model results of the negative endemic reptile relationship with yield (LINE 120).

I’m delighted to see the improved presentation of the statistical results shown in Figure 2 (i.e. the additional in-line estimates and supplementary tables), the new approach to capturing species richness across all taxa (instead of multidiversity) and the species accumulation curves and NMDS plots, as well as the additional work on endemism. Although you explain that the study focus is on within-plot yield-richness responses, adding these further analyses allows understanding of what’s going on at broader spatial scales, and provides much greater depth to your study. The clear patterns in species composition in Figure S4, and the rarefaction curves, make it easier to understand the thoroughness of your sampling and robustness of your work and findings, but also add qualitative information about ‘what biodiversity’ which is critical. I would be pleased to see the species composition results in the main text, but I am aware there are restrictions on the number of figures, and leave it to the authors/editor to decide whether this is worthwhile.

3. RESPONSE: We are grateful to hear the positive response to our revision. We are happy that these additional analyses are helpful. Indeed, as also argued by the reviewer, our focus lies on local trade-offs of (alpha-)diversity and agricultural productivity. While this additional analysis on beta-diversity (NMDS ordination) thus adds important additional information, they are rather supporting the main arguments of our manuscript. We, therefore, decided to retain them in the supplementary materials. However, in case the editor thinks that they should be included in the main text, we are of course happy to do so.

I agree that a structural equation model approach is not necessarily, but at line 173 would like to at least push for a change to the phrasing and interpretation of these modelled relationships to expressly note that the relationship is mediated by management and land use history, and is not causal.

4. RESPONSE: We added, “Notably, the analyzed relationship of yield with biodiversity is likely mediated by underlying variables such as management and land-use history”. (LINE 112)

Line 935: you could add a couple of words here to explain the 75% canopy cover threshold used to define forest, following the text in response 328.

5. *RESPONSE*: Thanks for this remark. We now include the following sentence: “To reliably identify forest, tree cover maps and satellite imagery with a tree cover threshold of 75% were combined³⁰“. (LINE 560)

Thank you for working to clarify the use of the terms ‘restoration’ and ‘degradation’.

Great to see the forest fragments now included separately from the old growth forest as reference (non-productive) systems, as you had the data already good to make it work for you!

Great to see plots of vanilla yield responses! The explanation of management-related variables, yields and richness is much improved.

6. *RESPONSE*: We are happy to hear your positive feedback on our last revision!

Please check these lines – there is some repetition and contradiction (I don’t think anything kept in 90% alcohol could then be released afterwards)

“To retrieve a DNA sample, we collected muscle or toe clips as tissue samples of individuals. Until release, we kept them in a ventilated bag to retain moisture, conserved in 90% of alcohol. We stored DNA samples at the Evolutionary Biology laboratory at TU Braunschweig. We also took photos of specimens that we did not identify to species level (ventral, back and flank view). Until release, we kept them in a ventilated bag to retain moisture. We released all specimens after completing the full-time-standardized search.” (LINE 1057-1061)

7. *RESPONSE*: Sorry for this repetition in the last response letter. The paragraph in the main manuscript is correct: “To retrieve a DNA sample, we collected muscle or toe clips as tissue samples, conserved in 90% of alcohol. We stored DNA samples at the Evolutionary Biology laboratory at TU Braunschweig. We also took photos of specimens that we did not identify to species level (ventral, back, and flank view). Until release, we kept them in a ventilated bag to retain moisture. We released all specimens after completing the full-time-standardized search.” (LINE 663-668)

Kindly include this very useful bit of methodological detail in the main text “fishing line covered with Vaseline was used to hang the bait traps, to prevent ant intrusion” – this will help future researchers lose valuable samples.

8. *RESPONSE*: That’s a great idea! We added “We used fish lines covered with vaseline creme to hang the fruit traps. The Vaseline prevented ants to intrude on the fruit traps. In addition, we avoided any contact of branches on the fruit traps” in the main text (LINE 678-680)

Thank you for addressing all my points in detail. The addition of this (perhaps laborious-seeming) detail improves interpretation and robustness greatly, and also aids the use of the work to inform future studies. Congratulations on an important and detailed study!

9. *RESPONSE*: We express our gratitude for your time investment in this detailed and constructive review. We agree that it has greatly improved the manuscript.

Reviewer #3 (Remarks to the Author):

I have read the authors response to my comments on biodiversity sampling and the corresponding text changes. I am very satisfied with the replies and text modifications. I think the authors made a great job in adding more information in the methods and I appreciate their effort to publish online data with more information. Thank you!

10. *RESPONSE*: Thanks for this appreciation! We thank reviewer 3 for his/her review which helped to improve the method section of our manuscript greatly.